# NExT-Mol: 3D Diffusion Meets 1D Language Modeling for 3D Molecule Generation

**Zhiyuan Liu**[1,*] **Yanchen Luo**[2*], **Han Huang**[3], **Enzhi Zhang**[4], **Sihang Li**[2],
**Junfeng Fang**[2], **Yaorui Shi**[2], **Xiang Wang**[2†], **Kenji Kawaguchi**[1], **Tat-Seng Chua**[1]
[1] National University of Singapore, [2] University of Science and Technology of China,
[3] Chinese University of Hong Kong, [4] Hokkaido University
zhiyuan@nus.edu.sg, luoyanchen@mail.ustc.edu.cn

## ABSTRACT

3D molecule generation is crucial for drug discovery and material design. While prior efforts focus on 3D diffusion models for their benefits in modeling continuous 3D conformers, they overlook the advantages of 1D SELFIES-based Language Models (LMs), which can generate 100% valid molecules and leverage the billion-scale 1D molecule datasets. To combine these advantages for 3D molecule generation, we propose a foundation model – NExT-Mol: 3D Diffusion Meets 1D Language Modeling for 3D Molecule Generation. NExT-Mol uses an extensively pretrained molecule LM for 1D molecule generation, and subsequently predicts the generated molecule's 3D conformers with a 3D diffusion model. We enhance NExT-Mol's performance by scaling up the LM's model size, refining the diffusion neural architecture, and applying 1D to 3D transfer learning. Notably, our 1D molecule LM significantly outperforms baselines in distributional similarity while ensuring validity, and our 3D diffusion model achieves leading performances in conformer prediction. Given these improvements in 1D and 3D modeling, NExT-Mol achieves a 26% relative improvement in 3D FCD for *de novo* 3D generation on GEOM-DRUGS, and a 13% average relative gain for conditional 3D generation on QM9-2014. Our codes and pretrained checkpoints are available at https://github.com/acharkq/NExT-Mol.

## 1 INTRODUCTION

Molecule discovery is crucial for designing new drugs and materials. To efficiently navigate the astronomical chemical space of molecules, generative deep learning methods have been extensively explored. While promising progress has been made in generating 2D molecular graphs (Jin et al., 2018; Vignac et al., 2023a), recent research has shifted toward 3D molecule generation due to its broader application scope. For example, understanding the 3D molecular geometry is crucial for structure-based drug design (Zhang et al., 2023), prediction of molecular quantum chemical properties (Zhou et al., 2023), and molecular dynamic simulation (Hansson et al., 2002).

3D molecule generation aims to predict 3D molecular conformers along with their 2D graphs (Hoogeboom et al., 2022). These generated 3D molecules are typically evaluated based on their molecular validity and stability, ensuring adherence to the chemical valency rules. Recent advancements in 3D diffusion models (Vignac et al., 2023b; Hua et al., 2023; Huang et al., 2024) have improved these metrics by better modeling continuous 3D conformers, yet they still occasionally generate invalid molecules. This validity issue hinders distribution learning of valid molecular structures, like pharmacophoric functional groups. For improvement, we draw inspiration from 1D molecule generation (Fang et al., 2024b; Polykovskiy et al., 2020) studies, which reliably ensure 100% validity. By representing 2D molecular graphs as linear strings of SELFIES (Krenn et al., 2020), these approaches typically leverage 1D language models (LMs) for 2D molecule generation. Due to SELFIES' inherent robustness, the generated molecules are guaranteed to be 100% valid. Inspired by these studies, a natural solution for improving 3D molecule generation is to incorporate a 1D SELFIES-based LM into a 3D diffusion model (Jing et al., 2022), thus leveraging the chemical

---

*Equal contribution.   † Correspondence to Xiang Wang. *xiangwang1223@gmail.com*.

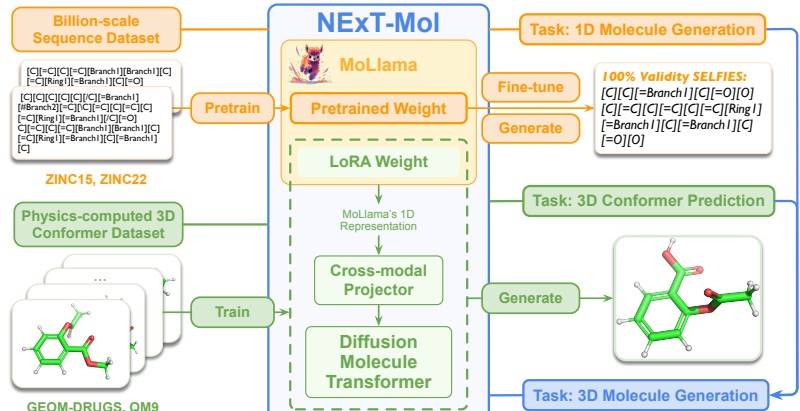

Figure 1: Overview of our NExT-Mol foundation model for 3D molecule generation. NExT-Mol consists of three key components: (1) MoLlama, a large LM for generating 1D molecule sequences; (2) DMT, a diffusion model to predict 3D conformers from the 1D sequences; and (3) NExT-Mol leverages transfer learning to enhance DMT's 3D prediction with MoLlama's 1D representations.

validity of 1D representations while improving 3D conformer prediction. To our best knowledge, few prior research has thoroughly explored this incorporation for 3D molecule generation.

To bridge the research gap above, we explore a two-step solution for 3D molecule generation: initially generating a 1D molecule (a subset of a 3D molecule) using an LM and subsequently predicting its 3D conformer with a diffusion model. Here we focus on three key strategies — scaling up 1D molecular LMs, refining the architecture of 3D diffusion models, and utilizing transfer learning between 1D and 3D modeling — to resolve the following three challenges faced by prior studies:

- **The Development of An Effective 1D Molecular LM.** This can be done by training an autoregressive transformer LM (Vaswani et al., 2017) on a large SELFIES corpus. However, existing studies have the following limitations: some use non-autoregressive pretraining, rendering them unsuitable for *de novo* generation (Fang et al., 2024b; Irwin et al., 2022; Born & Manica, 2023; Yüksel et al., 2023); some do not have 100% validity (Bagal et al., 2021); and others are constrained by small model sizes and employ non-transformer architectures, limiting their scalability (Polykovskiy et al., 2020; Eckmann et al., 2022; Arús-Pous et al., 2019; Jin et al., 2018).

- **The Design of A Powerful 3D Diffusion Model.** This is to accurately generate the 3D conformers for the 1D molecules generated by the 1D molecular LM in the earlier step. Existing works can be improved by adopting scalable architectures (Jing et al., 2022; Corso et al., 2024; Xu et al., 2022; Ganea et al., 2021) or leveraging the full information of 2D molecular graphs (Wang et al., 2024).

- **Transfer Learning between 1D Molecule Sequences and 3D Conformers.** It has the potential to offer a significant improvement to 3D conformer prediction, given the greater availability of 1D sequences compared to high-accuracy 3D conformers, which are typically derived by expensive physics-based computations. For example, ZINC22 (Tingle et al., 2023) now includes over 54.9 billion 1D sequences and GEOM (Axelrod & Gomez-Bombarelli, 2022) holds only 37 million 3D conformers. Although this 1D to 3D transfer learning was successfully applied to 3D protein structure prediction (Lin et al., 2023; Wu et al., 2022), similar methods remain mostly unexplored for small molecules, indicating a significant research opportunity.

To address the challenges above, we propose a foundation model – **NExT-Mol**: 3D Diffusion Meets 1D Language Modeling for 3D Molecule Generation, as illustrated in Figure 1. NExT-Mol consists of three key components: (1) To achieve effective autoregressive 1D molecule generation, we pretrain a Molecular Llama LM (**MoLlama**) (Touvron et al., 2023; Zhang et al., 2024) on a large collection of 1.8B SELFIES sequences. This extensive pretraining empowers MoLlama to effectively capture the desired 1D/2D molecular patterns (*e.g.,* scaffolds and fragments) in downstream datasets, laying a strong foundation for the subsequent 3D conformer prediction. (2) To achieve high-accuracy 3D conformer prediction, we introduce a novel diffusion model – Diffusion Molecular Transformer (**DMT**). DMT combines the power of a scalable neural architecture (Wang et al., 2024) and retains the full information of 2D molecular graphs by incorporating the Relational Multi-Head Self-Attention (Huang et al., 2024) that extends the standard self-attention by incorporating pair in-

formation describing atomic interactions. We show that DMT achieves leading performance for 3D conformer prediction, surpassing prior works by 1.1% COV-R on GEOM-DRUGS. Further, it accurately reveals the 3D structures of MoLlama-generated 1D molecules, providing a 26% relative gain in 3D FCD and significant improvements in geometric similarity and stability on GEOM-DRUGS. (3) We show that transfer learning between 1D molecular sequences and 3D conformers improves conformer prediction by 1.3% COV-R on GEOM-DRUGS. This improvement is driven by transferring MoLlama's pretrained 1D representations, which encode rich molecular knowledge, to DMT for better molecular representation. The 1D-to-3D modality gap in transfer learning is bridged by our proposed cross-modal projector and the corresponding training strategy (Liu et al., 2024a).

Collectively, our NExT-Mol foundation model is a versatile multi-task learner, and demonstrates leading performances for *de novo* 3D molecule generation, conditional 3D molecule generation, and 3D conformer prediction on the GEOM-DRUGS, GEOM-QM9 (Axelrod & Gomez-Bombarelli, 2022) and QM9-2014 (Ramakrishnan et al., 2014) datasets. The strong performance highlights NExT-Mol's effectiveness and its potential impact as a foundation model in the field. We further present extensive ablation studies to demonstrate the significance of each component of NExT-Mol.

## 2 RELATED WORKS

A complete molecule includes atoms, bonds, and the 3D coordinates of atoms (*i.e.,* 3D conformer). However, due to the expensive computation for obtaining high-accuracy 3D conformers (Axelrod & Gomez-Bombarelli, 2022), many studies focus on generating atoms and bonds without 3D conformers, representing molecules as 1D sequences or 2D graphs. Here we begin by reviewing 1D and 2D molecule generation, then discuss 3D molecule generation and 3D conformer prediction.

**1D and 2D Molecule Generation** aims to generate the atoms and bonds of a molecule. 1D generation works are mostly based on LMs. However, they usually apply non-autoregressive pretraining such as span-prediction (Irwin et al., 2022; Fang et al., 2024b; Born & Manica, 2023), making them unsuitable for *de novo* generation. Other works use non-transformer architecture (Arús-Pous et al., 2019; Polykovskiy et al., 2020; Flam-Shepherd et al., 2022; Gómez-Bombarelli et al., 2018; Eckmann et al., 2022; Popova et al., 2018), which are unsuitable for scale-up (Vaswani et al., 2017). 2D molecule generation works typically decompose molecular graphs as functional fragments (or atoms), and train models to recurrently generate or edit these fragments (Jin et al., 2018; Xie et al., 2021; Luo et al., 2021; Shi et al., 2020; Sun et al., 2022; Liu et al., 2018; You et al., 2018; Popova et al., 2019; Jin et al., 2019). However, due to their non-transformer architectures and domain-specialized training methods, these 2D generation models also face challenges with scalability and transfer learning. We refer readers to (Du et al., 2022) for a comprehensive survey in this area.

**3D Molecule Generation** is dominated by diffusion models (Hoogeboom et al., 2022; Bao et al., 2023; Huang et al., 2023a; 2024; 2023b; Vignac et al., 2023b; Hua et al., 2023). While autoregressive methods have been explored (Gebauer et al., 2019; 2022; Luo & Ji, 2022; Simm et al., 2020), they underperform diffusion models, potentially due to their inability to model bonds and the error accumulation when autoregressively generating 3D coordinates. Diffusion models typically employ 3D equivariant neural networks (Satorras et al., 2021) to denoise the variables of atoms, bonds, and 3D coordinates within a single diffusion process. However, they predict molecules without validity constraints and are limited by insufficient 3D data. To address these issues, we aim to integrate the two advantages of 1D SELFIES sequences – 100% validity and the more abundant dataset (Sterling & Irwin, 2015; Tingle et al., 2023) – into 3D molecule generation for improvement.

**3D Conformer Prediction** is to predict the 3D conformer given the atoms and bonds of a molecule (Xu et al., 2022; Ganea et al., 2021; Zhou et al., 2023; Jing et al., 2022; Corso et al., 2024). The current state-of-the-art approach scales up a diffusion model using a general-purpose transformer architecture (Wang et al., 2024), but it overlooks the chemical bond information and uses a lossy representation of molecular structures. We address these issues by introducing the DMT architecture that maintains scalability and retains the full information of 2D molecular graphs.

## 3 3D DIFFUSION MEETS 1D LM FOR 3D MOLECULE GENERATION

**NExT-Mol for 3D Molecule Generation.** NExT-Mol is a foundation model that generates 3D molecules with a two-step method: initially generating the 1D molecule sequence (a subset of a 3D

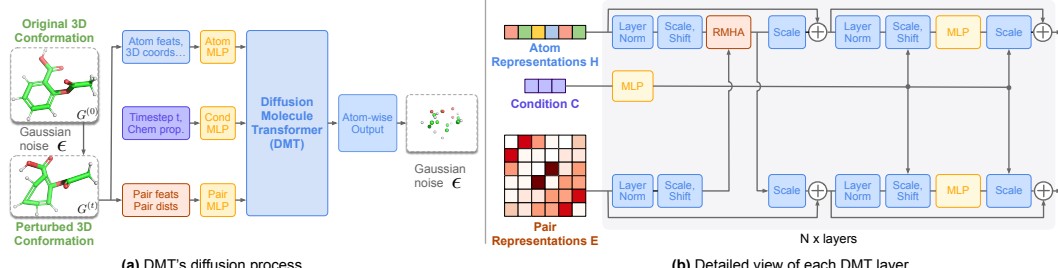

**(a)** DMT's diffusion process.                    **(b)** Detailed view of each DMT layer.

Figure 2: Overview of DMT's neural architecture. **(a)** DMT is a diffusion model learning to denoise random Gaussian perturbations $\boldsymbol{\epsilon}$ applied on the 3D coordinates of atoms. **(b)** DMT relies on the RMHA module to iteratively update atom representations $\mathbf{H}$ and pair representations $\mathbf{E}$.

molecule) using the MoLlama LM and subsequently predicting its 3D conformer using the DMT diffusion model. Here we begin by introducing the MoLlama for 1D molecule generation and then proceed to DMT. Finally, we detail the transfer learning method to incorporate MoLlama's 1D representation to enhance DMT's 3D conformer prediction. Appendix C includes implementation details.

### 3.1 1D MOLECULE GENERATION WITH MOLECULAR LLAMA LM

**Data Preparation.** Following (Irwin et al., 2022), we collect 1.8 billion molecules from the ZINC-15 database (Sterling & Irwin, 2015), significantly more than the 100 million molecules used in previous studies (Irwin et al., 2022; Fang et al., 2024b). We preprocess the molecules to transform them into SELFIES and perform data filtering to avoid overlap with the downstream datasets. The resulting dataset contains 90 billion SELFIES tokens.

**Pretraining MoLlama.** Our MoLlama is a 960M parameter LM with the popular decoder-only Llama-2 (Touvron et al., 2023) architecture. We pretrain it from scratch for 1D molecule generation with the next-token prediction objective. The pretraining takes 555K global steps, processing 145 billion tokens, which amounts to approximately 1.6 passes through the pretraining dataset.

**Randomized SELFIES Augmentation.** We use randomized SELFIES as data augmentations during fine-tuning MoLlama for 1D molecule generation. A molecule can have multiple valid SELFIES, because they are generated by traversing the 2D molecular graph in different orders. Randomized SELFIES are generated by traversing in random orders. This approach improves sample diversity and mitigates overfitting compared to using the canonical traversal order (Arús-Pous et al., 2019). The intuition is that the atoms in a molecule are inherently unordered, therefore an ideal LM should generate different orderings of the same molecule with equal likelihood.

### 3.2 3D CONFORMER PREDICTION WITH DIFFUSION MOLECULAR TRANSFORMER

Here we elaborate on the three key components of our proposed DMT: (1) the diffusion process governing the training and inference; (2) the neural architecture; and (3) the rotation augmentation.

**Diffusion Process.** A molecule $G = (\mathbf{x}, \mathbf{h}, \mathbf{e})$ is represented by its 3D coordinates $\mathbf{x} \in \mathbb{R}^{N \times 3}$, atom features $\mathbf{h} \in \mathbb{R}^{N \times d_1}$ (*e.g.,* atom types), and pair features $\mathbf{e} \in \mathbb{R}^{N \times N \times d_2}$ (*e.g.,* chemical bonds), where $N$ is the number of atoms and $d_1$ and $d_2$ are the feature dimensions. For 3D conformer prediction, we use a continuous-time diffusion model (Kingma et al., 2021) that denoises a molecule's 3D coordinates $\mathbf{x}$ based on its atom and pair features. As Figure 2a shows, in the forward diffusion process, noises are gradually applied to the original 3D coordinates $\mathbf{x}^{(0)} = \mathbf{x}$ such that $q(\mathbf{x}^{(t)}|\mathbf{x}^{(0)}) = \mathcal{N}(\mathbf{x}^{(t)}; \sqrt{\bar{\alpha}^{(t)}}\mathbf{x}^{(0)}, (1-\bar{\alpha}^{(t)})\mathbf{I})$, where $t \in (0, 1]$ is the diffusion's time-step, and $\bar{\alpha}^{(t)}$ is a hyperparameter controlling the noise scale at the $t$ step. Based on the reparameterization trick (Ho et al., 2020), we can sample $\mathbf{x}^{(t)} = \sqrt{\bar{\alpha}^{(t)}}\mathbf{x}^{(0)} + \sqrt{1 - \bar{\alpha}^{(t)}}\boldsymbol{\epsilon}^{(t)}$, where $\boldsymbol{\epsilon}^{(t)} \sim \mathcal{N}(\mathbf{0}, \mathbf{I})$. Given the perturbed coordinates $\mathbf{x}^{(t)}$, DMT is trained to predict the noise $\boldsymbol{\epsilon}^{(t)}$ by minimizing the MSE loss $\mathcal{L} = \|\boldsymbol{\epsilon}^{(t)} - \text{DMT}(G^{(t)}, t)\|_2^2$, where $G^{(t)} = (\mathbf{x}^{(t)}, \mathbf{h}, \mathbf{e})$. After training, DMT can be employed for 3D conformer prediction through ancestral sampling (Ho et al., 2020).

**Neural Architecture.** As Figure 2b illustrates, DMT adopts Relational Multi-Head Self-Attention (**RMHA**) (Huang et al., 2024) and adaptive layernorm (adaLN) (Perez et al., 2018; Peebles & Xie, 2023). adaLN replaces the learnable scale and shift parameters in standard layernorm (Ba, 2016)

(a) Cross-modal projector bridges MoLlama and DMT.    (b) Three training stages for 1D to 3D transfer learning.

Figure 3: Transfer learning between MoLlama's 1D representations and DMT's 3D prediction. **(a)** A cross-modal projector bridges the gap between MoLlama and DMT. Grey H atoms have no corresponding SELFIES tokens, and are replaced by a learnable token. **(b)** Transfer learning's three training stages. Snowflake ❄ denotes frozen parameters while flame 🔥 denotes trainable ones.

with adaptive ones that are generated from the condition embedding $\mathbf{C}$, which combines the time-step and optionally a desired chemical property. For simplicity, we omit adaLNs in discussion below.

The philosophy behind DMT's neural architecture generally follows the "bitter lesson" recently revealed by MCF (Wang et al., 2024) that large scalable models outperform domain-specific inductive biases. Notably, MCF shows that it is unnecessary to have an architecture of built-in 3D equivariance for conformer prediction. However, MCF is limited to employing a lossy representation of 2D molecular structures and overlooks bond information, by relying on the top-k eigenvectors of the graph Laplacian (Maskey et al., 2022) to represent 2D molecular graphs. To address this issue, DMT retains the full information of 2D molecular graphs in its atom representation $\mathbf{H} \in \mathbb{R}^{N \times d}$ and pair representation $\mathbf{E} \in \mathbb{R}^{N \times N \times d}$, and then applies RMHA to learn and distinguish the 2D graph structures. Specifically, the atom representations $\mathbf{H}$ are initialized by concatenating the atom features $\mathbf{h}$ and the perturbed 3D coordinates $\mathbf{x}^{(t)}$, the pair representations $\mathbf{E}$ are initialized by concatenating the pair features $\mathbf{e}$ and the distances between each atom pair. $\mathbf{H}$ and $\mathbf{E}$ are then iteratively refined by RMHA. The single-head RMHA is defined below with the multi-head version in Appendix C.2:

$$[\mathbf{Q}; \mathbf{K}; \mathbf{V}] = [\mathbf{W}_q; \mathbf{W}_k; \mathbf{W}_v]\mathbf{H}^\top, \quad (1) \qquad [\mathbf{Q}^E; \mathbf{V}^E] = \tanh([\mathbf{W}_{eq}; \mathbf{W}_{ev}]\mathbf{E}^\top), \quad (2)$$

$$a_{i,j} = \mathrm{softmax}_j\big(\frac{(\mathbf{Q}^E_{i,j} \odot \mathbf{Q}_i)\mathbf{K}_j^\top}{\sqrt{d}}\big), \quad (3) \qquad \mathbf{O}_i = \sum_{j=1}^{N} a_{i,j}(\mathbf{V}^E_{i,j} \odot \mathbf{V}_j), \quad (4)$$

where $\odot$ denotes element-wise product; $\mathrm{softmax}_j$ denotes softmax along the $j$ dimension; linear projectors $\mathbf{W}_q$, $\mathbf{W}_k$, and $\mathbf{W}_v$ generate queries, keys, and values for atom representations, $\mathbf{W}_{eq}$ and $\mathbf{W}_{ev}$ generate queries and values for pair representations; $\mathbf{O}_i$ is RMHA's output for the $i$-th atom; and $\mathbf{Q}^E_{i,j}, \mathbf{V}^E_{i,j} \in \mathbb{R}^d$ are the query and value for the atom pair representation $(i, j)$.

RMHA uses the pair-level query $\mathbf{Q}^E_{ij}$ and key $\mathbf{V}^E_{ij}$ of $\mathbf{E}$ to modify the atom-level query $\mathbf{Q}_i$ and value $\mathbf{V}_j$ through element-wise multiplication ($\odot$), enabling RMHA to fully incorporate pair representations. Specifically, the pair $\mathbf{E}$ affects attention scores via $(\mathbf{Q}^E_{ij} \odot \mathbf{Q}_i)\mathbf{K}_j^\top$, and affects the aggregated attention values via $\mathbf{V}^E_{ij} \odot \mathbf{V}_j$. In this way, the output $\mathbf{O}$ is adaptively informed by the structural and interaction information in $\mathbf{E}$. After RMHA, $\mathbf{O}_i$ is passed to an MLP to update the atom representation $\mathbf{H}_i$, and the linear combination of $\mathbf{O}_i$ and $\mathbf{O}_j$ is used to update the pair representation $\mathbf{E}_{i,j}$. As Figure 2b illustrates, residual connections and adaLNs are included for improved performance.

**Random Rotation Augmentation.** Following AlphaFold3 (Abramson et al., 2024), we apply the same random rotation augmentation on both the input 3D coordinates ($\mathbf{x}^{(t)}$) and the target 3D coordinates ($\boldsymbol{\epsilon}^{(t)}$) to help DMT obtain equivariance to rotated inputs by learning. While (Wang et al., 2024) report decreased performance given random rotations, DMT benefits from it, potentially due to the improved neural architecture.

### 3.3 MoLlama Representations Improve DMT's 3D Conformer Prediction

We explore the transfer learning between molecular 1D sequences and 3D conformers. As Figure 3 illustrates, we leverage MoLlama's pretrained representation to improve DMT's 3D conformer prediction. This is achieved by our cross-modal projector and the corresponding training strategy.

**Cross-Modal Projector.** Following Liu et al. (2023a), DMT uses a projector to leverage MoLlama for atom representation, addressing two challenges: (1) MoLlama uses causal self-attention, where each token only perceives preceding tokens, limiting the representation quality; and (2) SELFIES tokens do not map directly to individual atoms. Mitigating the first issue, we feed MoLlama's SELF-IES representations into a single-layer bi-directional self-attention (Vaswani et al., 2017), expanding the receptive field of SELFIES tokens. Further, we program the SELFIES-to-atom mapping using the SELFIES and RDKit software. For atoms corresponding to multiple SELFIES tokens, we obtain its representation by mean pooling; for hydrogen atoms without corresponding SELFIES tokens, we use a learnable token as a replacement. The output of the SELFIES-to-atom mapping is then fed into an MLP and concatenated with DMT's original atom representations for 3D conformer prediction.

**Training Strategy.** As Figure 3b illustrates, to save computation, we fine-tune a pretrained DMT to incorporate MoLlama representations, instead of training a new DMT from scratch using MoLlama representations. Throughout the process, MoLlama uses LoRA tuning (Hu et al., 2021) to save memory. The training strategy consists of three stages. In the first stage, we train a standalone DMT without MoLlama until convergence. In the second stage, we attach MoLlama and the cross-modal projector to the pretrained DMT, keeping the DMT parameters frozen, and train for 10 epochs to warmup the random parameters in the projector and LoRA. This step prevents the gradients from the random parameters from distorting the pretrained DMT parameters (Kumar et al., 2022). In the final stage, we fine-tune the entire integrated model until convergence.

When incorporating MoLlama representations into DMT, we find that canonical SELFIES performs better than randomized SELFIES. This may be because bridging the gap between 1D MoLlama and 3D DMT is challenging, and using the fixed canonical representations leads to faster convergence.

## 4 EXPERIMENT

In this section, we evaluate NExT-Mol's performance on *de novo* 3D molecule generation and conditional 3D molecule generation. Further, we report results of 3D conformer prediction, the critical second step in our two-step generation process. Finally, we present ablation studies to demonstrate the effectiveness of each component of NExT-Mol.

### 4.1 EXPERIMENTAL SETTINGS

**Datasets.** As Table 1 shows, we evaluate on the popular GEOM-DRUGS (Axelrod & Gomez-Bombarelli, 2022), GEOM-QM9 (Axelrod & Gomez-Bombarelli, 2022), and QM9-2014 (Ramakrishnan et al., 2014) datasets. Among them, we focus on GEOM-DRUGS,

Table 1: Datasets for each task.

| Task | Dataset |
|---|---|
| *De novo* 3D Mol Gen | GEOM-DRUGS, QM9-2014 |
| Conditional 3D Mol Gen | QM9-2014 |
| 3D Conformer Pred | GEOM-DRUGS, GEOM-QM9 |

which is the most pharmaceutically relevant and largest one. Due to different tasks incorporating different dataset splits, we separately fine-tune NExT-Mol for each task without sharing weights.

**Baselines.** For *de novo* and conditional 3D molecule genration, we use baselines of CDGS (Huang et al., 2023a), JODO (Huang et al., 2024), MiDi (Vignac et al., 2023b), G-SchNet (Gebauer et al., 2019), G-SphereNet (Luo & Ji, 2022), EDM (Hoogeboom et al., 2022), MDM (Huang et al., 2023b), GeoLDM (Xu et al., 2023), EEGSDE (Bao et al., 2023), EQGAT-diff (Le et al., 2024), MolGPT (Bagal et al., 2021), and MolGen (Fang et al., 2024b). For 3D conformer prediction, we use baselines of OMEGA (Hawkins, 2017), GeoMol (Ganea et al., 2021), GeoDiff (Xu et al., 2022), Torsional Diffusion (Jing et al., 2022), Particle Guidance (Corso et al., 2024), and MCF (Wang et al., 2024). More details on experimental settings are in Appendix D.

**NExT-Mol.** Throughout the section, NExT-Mol fine-tunes the pretrained 960M MoLlama for 1D molecule generation. We have trained two versions of DMT: DMT-B of 55 million parameters and DMT-L of 150 million. For the *de novo* and conditional 3D generation molecule tasks (*cf.* Section 4.2 and Section 4.3), NExT-Mol uses DMT-B. DMT uses 100 sampling steps by default.

### 4.2 *DE NOVO* 3D MOLECULE GENERATION

**Experimental Setting.** Generating a complete 3D molecule involves generating the 2D molecular graph and the corresponding 3D conformer. Therefore, we evaluate both the predicted 2D molecu-

Table 2: Performances for *de novo* 3D molecule generation. * denotes our reproduced results using their source codes. Other baseline results are borrowed from (Huang et al., 2024). 2D-Metric evaluates the directly predicted 2D molecular graphs, whereas the 3D-Metric evaluates the predicted 3D coordinates or the 2D molecular graphs reconstructed from the 3D coordinates.

(a) Performances on the GEOM-DRUGS dataset.

| 2D-Metric | FCD↓ | AtomStable | MolStable | V&C | V&U | V&U&N | SNN | Frag | Scaf |
|---|---|---|---|---|---|---|---|---|---|
| Train | 0.251 | 1.000 | 1.000 | 1.000 | 1.000 | 0.000 | 0.585 | 0.999 | 0.584 |
| MolGPT* | 0.888 | 0.979 | 0.977 | 0.957 | 0.955 | 0.918 | 0.520 | 0.991 | 0.539 |
| MolGen* | 0.655 | **1.000** | 0.995 | **1.000** | 0.993 | 0.759 | 0.513 | 0.993 | 0.549 |
| CDGS | 22.051 | 0.991 | 0.706 | 0.285 | 0.285 | 0.285 | 0.262 | 0.789 | 0.022 |
| JODO | 2.523 | **1.000** | 0.981 | 0.874 | 0.905 | 0.902 | 0.417 | 0.993 | 0.483 |
| MiDi* | 7.054 | 0.968 | 0.818 | 0.633 | 0.654 | 0.652 | 0.392 | 0.951 | 0.196 |
| EQGAT-diff* | 6.310 | 0.999 | 0.998 | 0.959 | 0.993 | 0.702 | 0.368 | 0.986 | 0.147 |
| NExT-Mol, ours | **0.334** | **1.000** | **0.999** | **1.000** | **0.999** | **0.945** | **0.529** | **0.999** | **0.552** |

| 3D-Metric | FCD↓ | AtomStable | Bond length↓ | Bond angle↓ | Dihedral angle↓ |
|---|---|---|---|---|---|
| Train | 13.73 | 0.861 | 1.56E-04 | 1.81E-04 | 1.56E-04 |
| EDM | 31.29 | 0.831 | 4.29E-01 | 4.96E-01 | 1.46E-02 |
| JODO | 19.99 | 0.845 | 8.49E-02 | 1.15E-02 | 6.68E-04 |
| MiDi* | 23.14 | 0.750 | 1.17E-01 | 9.57E-02 | 4.46E-03 |
| EQGAT-diff* | 25.89 | 0.846 | 1.23E-01 | 5.29E-02 | 2.17E-03 |
| NExT-Mol, ours | **14.69** | **0.848** | **2.05E-02** | **8.18E-03** | **2.31E-04** |

(b) Performances on the QM9-2014 dataset.

| 2D-Metric | FCD↓ | AtomStable | MolStable | V&C | V&U | V&U&N | SNN | Frag | Scaf |
|---|---|---|---|---|---|---|---|---|---|
| Train | 0.063 | 0.999 | 0.988 | 0.989 | 0.989 | 0.000 | 0.490 | 0.992 | 0.946 |
| MolGPT* | 0.461 | 0.982 | 0.976 | 0.977 | 0.937 | 0.763 | 0.523 | 0.958 | 0.923 |
| MolGen* | 0.085 | **1.000** | 0.988 | **1.000** | 0.955 | 0.479 | 0.500 | 0.988 | 0.934 |
| CDGS | 0.798 | 0.997 | 0.951 | 0.951 | 0.936 | **0.860*** | 0.493 | 0.973 | 0.784 |
| JODO | 0.138 | 0.999 | 0.988 | 0.990 | 0.960 | 0.780* | 0.522 | 0.986 | 0.934 |
| MiDi* | 0.187 | 0.998 | 0.976 | 0.980 | 0.954 | 0.769 | 0.501 | 0.979 | 0.882 |
| EQGAT-diff* | 2.157 | **1.000** | 0.972 | **1.000** | 0.996 | 0.695 | 0.479 | 0.949 | 0.707 |
| NExT-Mol, ours | **0.070** | **1.000** | **0.989** | **1.000** | 0.967 | 0.802 | **0.530** | **0.992** | **0.945** |

| 3D-Metric | FCD↓ | AtomStable | Bond length↓ | Bond angle↓ | Dihedral angle↓ |
|---|---|---|---|---|---|
| Train | 0.877 | 0.994 | 5.44E-04 | 4.65E-04 | 1.78E-04 |
| G-SchNet | 2.386 | 0.957 | 3.62E-01 | 7.27E-02 | 4.20E-03 |
| G-SphereNet | 6.659 | 0.672 | 1.51E-01 | 3.54E-01 | 1.29E-02 |
| EDM | 1.285 | 0.986 | 1.30E-01 | 1.82E-02 | 6.64E-04 |
| MDM | 4.861 | 0.992 | 2.74E-01 | 6.60E-02 | 2.39E-02 |
| JODO | 0.885 | 0.992 | 1.48E-01 | 1.21E-02 | 6.29E-04 |
| MiDi* | 1.100 | 0.983 | 8.96E-01 | 2.08E-02 | 8.14E-04 |
| EQGAT-diff* | 1.519 | 0.988 | 4.09E-01 | 1.91E-02 | 1.14E-03 |
| NExT-Mol, ours | **0.879** | **0.993** | **1.15E-01** | **7.32E-03** | **1.95E-04** |

Table 3: Performance of conditional 3D molecule generation on the QM9-2014 dataset. We report MAE ↓ between the desired properties and the predicted properties of the generated samples. Baseline results are from (Huang et al., 2024). We **bold** the best performance.

| Method | $\mu$ (D) | $\alpha$ (Bohr$^3$) | $C_v$ $\left(\frac{\text{cal}}{\text{mol}}\text{K}\right)$ | $\varepsilon_{\text{HOMO}}$ (meV) | $\varepsilon_{\text{LUMO}}$ (meV) | $\Delta\varepsilon$ (meV) |
|---|---|---|---|---|---|---|
| L-Bound | 0.043 | 0.09 | 0.040 | 39 | 36 | 65 |
| EDM | 1.123 | 2.78 | 1.065 | 371 | 601 | 671 |
| EEGSDE | 0.777 | 2.50 | 0.941 | 302 | 447 | 487 |
| GeoLDM | 1.108 | 2.37 | 1.025 | 340 | 522 | 587 |
| JODO | 0.628 | 1.42 | 0.581 | 226 | 256 | 335 |
| NExT-Mol, ours | **0.507** | **1.16** | **0.512** | **205** | **235** | **297** |
| relative improv. | 19.3% | 18.3% | 11.9% | 9.3% | 8.2% | 11.3% |

lar graphs (*i.e.,* 2D-Metric), and the predicted 3D coordinates (*i.e.,* 3D-Metric), following (Hoogeboom et al., 2022; Huang et al., 2024). 2D-Metrics can be roughly grouped into three types: (1) stability and validity: atom stability, molecule stability, and validity & completeness (V&C); (2)

Table 4: 3D conformer prediction results. Baseline results are from (Jing et al., 2022; Corso et al., 2024; Wang et al., 2024). * denotes reproduction using their codes. -R←Recall and -P←Precision.

(a) Performances on the GEOM-DRUGS dataset. TD *w/* PG denotes torsional diffusion with particle guidance.

| Method | Model Size | COV-R (%)↑ Mean | COV-R (%)↑ Median | AMR-R ↓ Mean | AMR-R ↓ Median | COV-P (%)↑ Mean | COV-P (%)↑ Median | AMR-P ↓ Mean | AMR-P ↓ Median |
|---|---|---|---|---|---|---|---|---|---|
| **Model size ≤ 100M** | | | | | | | | | |
| OMEGA | - | 53.4 | 54.6 | 0.841 | 0.762 | 40.5 | 33.3 | 0.946 | 0.854 |
| GeoMol | 0.3M | 44.6 | 41.4 | 0.875 | 0.834 | 43.0 | 36.4 | 0.928 | 0.841 |
| GeoDiff | 1.6M | 42.1 | 37.8 | 0.835 | 0.809 | 24.9 | 14.5 | 1.136 | 1.090 |
| Torsional Diffusion | 1.6M | 72.7 | 80.0 | 0.582 | 0.565 | 55.2 | 56.9 | 0.778 | 0.729 |
| TD *w/* PG | 1.6M | 77.0 | 82.6 | 0.543 | 0.520 | 68.9 | 78.1 | 0.656 | 0.594 |
| TD *w/ PG** | 1.6M | 73.8 | 79.3 | 0.566 | 0.539 | 65.2 | 70.8 | 0.680 | 0.615 |
| MCF-S | 13M | 79.4 | 87.5 | 0.512 | 0.492 | 57.4 | 57.6 | 0.761 | 0.715 |
| MCF-B | 64M | 84.0 | 91.5 | 0.427 | 0.402 | 64.0 | 66.2 | 0.667 | 0.605 |
| DMT-B, ours | 55M | 85.4 | **92.2** | 0.401 | 0.375 | 65.2 | 67.8 | 0.642 | 0.577 |
| DMT-B, PC samp. | 55M | **85.5** | 91.2 | **0.396** | **0.370** | **67.6** | **71.5** | **0.623** | **0.546** |
| **Model size > 100M** | | | | | | | | | |
| MCF-L | 242M | 84.7 | 92.2 | 0.390 | **0.247** | 66.8 | 71.3 | 0.618 | 0.530 |
| DMT-L, ours | 150M | **85.8** | **92.3** | **0.375** | 0.346 | **67.9** | **72.5** | **0.598** | **0.527** |

(b) Performances on the GEOM-QM9 dataset.

| Method | Model size | COV-R (%)↑ Mean | COV-R (%)↑ Median | AMR-R ↓ Mean | AMR-R ↓ Median | COV-P (%)↑ Mean | COV-P (%)↑ Median | AMR-P ↓ Mean | AMR-P ↓ Median |
|---|---|---|---|---|---|---|---|---|---|
| OMEGA | - | 85.5 | **100.0** | 0.177 | 0.126 | 82.9 | **100.0** | 0.224 | 0.186 |
| GeoMol | 0.3M | 91.5 | **100.0** | 0.225 | 0.193 | 86.7 | **100.0** | 0.270 | 0.241 |
| GeoDiff | 1.6M | 76.5 | **100.0** | 0.297 | 0.229 | 50.0 | 33.5 | 0.524 | 0.510 |
| Torsoinal Diffusion | 1.6M | 92.8 | **100.0** | 0.178 | 0.147 | 92.7 | **100.0** | 0.221 | 0.195 |
| MCF-B | 64M | 95.0 | **100.0** | 0.103 | 0.044 | 93.7 | **100.0** | 0.119 | 0.055 |
| DMT-B, ours | 55M | **95.2** | **100.0** | **0.090** | **0.036** | **93.8** | **100.0** | **0.108** | **0.049** |

diversity: validity & uniqueness (V&U), and validity & uniqueness & novelty (V&U&N); and (3) distribution similarity between the generated molecules and the test set: similarity to nearest neighbor (SNN), fragment similarity (Frag), scaffold similarity (Scaf), and Fréchet ChemNet Distance (FCD) (Polykovskiy et al., 2020). For 3D-Metrics, we follow (Hoogeboom et al., 2022) to evaluate the predicted 3D molecules by assessing atom stability, and FCD of the 2D molecular graphs reconstructed from predicted 3D coordinates. Additionally, 3D-Metrics includes the maximum mean discrepancy (MMD) (Gretton et al., 2012) for bond lengths, bond angles, and dihedral angles to evaluate geometric similarity to the test set. Training set performance is also reported for reference. The experimental results are presented in Table 2. We can observe that:

**Obs. 1: NExT-Mol Demonstrates Leading Performances for 3D Molecule Generation.** It achieves the best performance across all metrics on GEOM-DRUGS, and achieves the best performance in 13 out of 14 metrics on QM9-2014. Although CDGS shows a higher novelty score on QM9-2014, it significantly underperforms NExT-Mol for other metrics. This observation shows that NExT-Mol is highly effective at generating chemically valid and diverse 3D molecular structures. Its strong performance on both large (*i.e.,* GEOM-DRUGS) and small (*i.e.,* QM9-2014) molecules highlights its robustness and potential as a foundation model for various tasks.

**Obs. 2: NExT-Mol is Powerful in Capturing 1D/2D Molecular Characteristics**, including SNN, Frag, Scaf, and FCD. Notably, it improves the FCD from 0.655 to 0.334 on GEOM-DRUGS, acheiving a 49% relative improvement. This good performance is attributed to MoLlama's extensive pre-training, which lays a strong foundation for the subsequent 3D conformer prediction.

### 4.3 CONDITIONAL 3D MOLECULE GENERATION WITH QUANTUM CHEMICAL PROPERTIES

**Adatping NExT-Mol for Conditional Generation.** We employ NExT-Mol for conditional 3D molecule generation targeting quantum chemistry properties. To adapt NExT-Mol to incorporate numerical conditions, the desired property values are encoded into vector embeddings using MLPs. These embeddings are prepended to the SELFIES sequences during MoLlama fine-tuning, serving

Table 5: Incorporating MoLlama's 1D representations to improve DMT's 3D conformer prediction.

| Dataset | Method | COV-R (%)↑ | | AMR-R ↓ | | COV-P (%)↑ | | AMR-P ↓ | |
|---|---|---|---|---|---|---|---|---|---|
| | | Mean | Median | Mean | Median | Mean | Median | Mean | Median |
| GEOM-DRUGS | DMT-B | 85.4 | **92.2** | 0.401 | 0.375 | 65.2 | 67.8 | 0.642 | 0.577 |
| | +MoLlama | **86.1** | 92.1 | **0.383** | **0.367** | **66.2** | **68.6** | **0.626** | **0.566** |
| | DMT-L | 85.8 | 92.3 | 0.375 | 0.346 | 67.9 | **72.5** | 0.598 | 0.527 |
| | +MoLLama | **87.1** | **93.0** | **0.360** | **0.334** | **68.1** | 71.8 | **0.595** | **0.525** |

Table 6: 3D conformer prediction performance on GEOM-DRUGS's test subsets, split by scaffold frequency in the training set. 68 low-quality samples are filtered following (Jing et al., 2022).

| Test subset | #Mol | Method | AMR-R | AMR-P |
|---|---|---|---|---|
| unseen scaffold | 348 | DMT-B | 0.450 | 0.785 |
| | | +MoLlama | **0.422** | **0.755** |
| scaf. freq. ≥1 | 584 | DMT-B | 0.364 | 0.549 |
| | | +MoLlama | **0.359** | **0.548** |
| scaf. freq. ≥10 | 285 | DMT-B | 0.348 | 0.515 |
| | | +MoLlama | **0.347** | **0.513** |

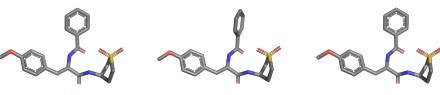

(a) Case 1. L to R: GT, DMT, DMT+MoLlama.

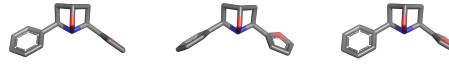

(b) Case 2. L to R: GT, DMT, DMT+MoLlama.

Figure 4: Visualization of 3D conformers. We select the predicted conformers with the least RMSD to the ground truth (GT).

as a soft-prompt to condition its output (Li & Liang, 2021), and are also fed into the DMT through the condition MLP module (*cf.* Figure 2). See Appendix D.4 for details of this methodology.

**Remark.** Quantum chemical properties (*e.g.,* HOMO-LUMO gap) often vary across a molecule's different 3D conformers. As a result, the 1D molecules generated by MoLlama alone cannot achieve errors lower than the average across a molecule's different conformers. To address this, we condition DMT on the desired property value when predicting the 3D conformer, enabling DMT to find the conformer that best matches the target property.

**Experimental Settings.** Following (Hoogeboom et al., 2022; Huang et al., 2024), we focus on six properties of heat capacity $C_v$, dipole moment $\mu$, polarizability $\alpha$, highest occupied molecular orbital energy $\epsilon_{\text{HOMO}}$, lowest unoccupied molecular orbital energy $\epsilon_{\text{LUMO}}$, and HOMO-LUMO gap $\Delta\epsilon$. For evaluation, we report the mean absolute error (MAE) between the desired property values and the predicted values of the generated molecules, using a property classifier network $\phi_c$ (Hoogeboom et al., 2022). QM9-2014's training set is split into two halves: $D_a$ and $D_b$, each containing 50k molecules. $\phi_c$ is trained on $D_a$ and NExT-Mol is trained on $D_b$. We report $\phi_c$'s performance on $D_b$ as the performance's lower-bound (L-Bound). Table 3 shows the results.

**Obs. 3: NExT-Mol Outperforms Baselines for Conditional 3D Molecule Generation.** NExT-Mol's improvements are consistent and significant, with an average relative gain of 13% on MAE, demonstrating its ability to effectively capture quantum chemical properties. This good performance is partially attributed to DMT, which finds the 3D conformer that best aligns the desired property.

### 4.4 3D MOLECULAR CONFORMER PREDICTION

**Experimental Setting.** Our setting follows (Jing et al., 2022). Evaluation metrics include Average Minimum RMSD (AMR), which measures the distance between a predicted conformer and a ground truth, and Coverage (COV), which measures the proportion of predicted conformers that are sufficiently close to a ground truth. Due to a 2D molecule can have multiple ground truth and predicted conformers, we report both precision (comparing a prediction to its most similar ground truth) and recall (comparing a ground truth to its most similar prediction) for AMR and Coverage. For DMT-B, we report performance with the predictor-corrector sampler (PC samp.; Song et al. (2021)), comparing with the particle guidance sampler (TD w/ PG).

**Obs. 4: DMT Demonstrates Leading Performance for 3D Conformer Prediction.** Table 4 compares DMT and baselines for 3D conformer prediction. We can observe that DMT-B outperforms MCF-B, and DMT-L surpasses MCF-L, even though DMT-L is only 60% of the size of MCF-L. This improvement demonstrates that DMT can better utilize 2D molecular graph structures than MCF. Further, DMT-L improves upon DMT-B, demonstrating DMT's scalability. Both the improvements

Table 7: Enhancing 3D molecule generation with MoLlama representations on GEOM-DRUGS.

| Method | 3D Pred. | FCD↓ | AtomStable | Bond length↓ | Bond angle↓ | Dihedral angle↓ |
|---|---|---|---|---|---|---|
| NExT-Mol | DMT-B | 14.69 | 0.848 | 2.05E-02 | 8.18E-03 | 2.31E-04 |
|  | +MoLLama | **14.32** | **0.852** | **1.48E-02** | **8.08E-03** | **1.81E-04** |

Table 8: Ablating randomized SELFIES augmentations for 1D molecule generation on QM9-2014.

| 2D metrics | FCD↓ | AtomStable | MolStable | V&C | V&U | V&U&N | SNN | Frag | Scaf |
|---|---|---|---|---|---|---|---|---|---|
| MoLlama | **0.070** | **1.000** | **0.989** | **1.000** | **0.967** | **0.802** | **0.530** | **0.992** | **0.945** |
| *w/o* randomized aug. | 0.074 | **1.000** | 0.988 | **1.000** | 0.948 | 0.395 | 0.491 | 0.989 | 0.939 |

above are attributed to DMT's meticulously designed architecture, combining the power of scalability while effectively leveraging the full information of 2D molecular graphs.

**Obs. 5: MoLlama's 1D Representation Improves DMT's 3D Conformer Prediction.** As Table 5 shows, MoLlama enhances DMT on GEOM-DRUGS. Table 12 shows integrating MoLlama into DMT also improves performance on GEOM-QM9. This observation demonstrates the potential to leverage the abundant 1D molecule sequences to improve 3D generation and design tasks, mitigating their data scarcity issue. Further, this observation highlights MoLlama's value to generate expressive molecule representations for 3D tasks, beyond its 1D molecule generation ability. Although MoLlama is pretrained only on 1D molecules, we hypothesize that large-scale pretraining helps it develop chemical heuristics useful for 3D prediction.

## 4.5    ANALYSIS AND ABLATION STUDIES

**MoLlama's 1D Representation Improves 3D Prediction for Unseen Scaffolds.** Scaffold split is widely used to evaluate a molecular model's generalization ability to unseen structures (Hu et al., 2020). We divide GEOM-DRUGS's test set into subsets based on the test molecule's scaffold frequency in the training set. As Table 6 shows, DMT-B's performance drops significantly for molecules with unseen scaffolds: AMR-R and AMR-P increase by 0.086 and 0.236, respectively, compared to molecules with scaffold frequency $\geq 1$. However, incorporating MoLlama mitigates this issue, reducing AMR-R and AMR-P by 0.028 and 0.030, respectively. This improvement stems from MoLlama's exposure to diverse scaffolds during pretraining on a large molecular dataset, enabling better generalization for transfer learning. Figure 4 highlights cases where MoLlama significantly enhances conformer prediction, particularly by improving torsion angle predictions.

**Enhancing 3D Molecule Generation with MoLlama Representations.** NExT-Mol uses DMT-B without MoLlama for conformer prediction by default for *de novo* 3D molecule generation. Here we show that enhancing DMT-B with MoLlama further improves its performance on 3D-metrics. Table 7 shows significant gains in geometric measures (*i.e.,* bond lengths, angles, and dihedral angles), highlighting MoLlama's ability to enhance DMT's 3D geometry prediction.

**Random SELFIES Augmentation.** As Table 8 shows, using randomized SELFIES augmentation significantly improves the novelty (*i.e.,* V&U&N) of the generated samples. It also improves other metrics, like SNN and FCD, highlighting its importance for 1D molecule generation.

## 5    CONCLUSION AND FUTURE WORKS

In this work, we presented NExT-Mol, a foundation model for 3D molecule generation that integrated the strengths of 1D SELFIES-based LMs and 3D diffusion models. NExT-Mol demonstrated leading performances in *de novo* 3D molecule generation, 3D conformer prediction, and conditional 3D molecule generation. These good performances are attributed to our focus on incorporating chemical inductive biases without compromising model scalability, and they highlight NExT-Mol's promising potential as a foundation model in the field. Additionally, NExT-Mol showed that transfer learning between 1D molecule sequences and 3D conformers can significantly improve 3D conformer prediction performance, underscoring the value of leveraging the abundant 1D molecular data to enhance 3D prediction tasks. Looking ahead, we plan to extend NExT-Mol to process multiple molecular inputs, aiming to tackle structure-based molecule design and modeling interactions between small molecules and proteins or RNAs, with real-world applications in drug discovery.

## 6 ETHICS STATEMENT

Our research advances 3D molecule generation with the NExT-Mol model, aiming to enhance generative deep learning methods for molecular design. This work is primarily technical and foundational, with applications in drug discovery and materials science. We have carefully considered potential societal impacts and do not foresee any direct, immediate, or negative consequences. We are committed to the ethical dissemination of our findings and encourage their responsible use.

## 7 REPRODUCIBILITY STATEMENT

All the results in this work are reproducible. We provide all the necessary code to replicate our results in an anonymous GitHub repository https://github.com/acharkq/NExT-Mol. The repository includes environment configurations, run scripts, and other relevant materials.

We discuss the experimental settings for various tasks in Section 4, including details on parameters such as sampling steps. Additionally, detailed experimental settings are provided in Appendix D.

## ACKNOWLEDGEMENT

This research is supported by the National Natural Science Foundation of China (92270114). This material is based upon work supported by the Air Force Office of Scientific Research under award number FA2386-24-1-4011, and this research is partially supported by the Singapore Ministry of Education Academic Research Fund Tier 1 (Award No: T1 251RES2207). This research is supported by NExT Research Center.

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

## A LIMITATIONS AND MORE FUTURE WORKS

NExT-Mol has several limitations that have not been addressed due to our limited computational resources and other technical challenges. We outline these limitations below:

**Explore Generalization of 3D Conformer Prediction to Unseen Scaffolds.** Table 6 shows that DMT-B's performance drop significantly on test molecules with unseen scaffolds in the training set. While our proposed transfer learning using MoLlama's pretrained 1D representations can mitigate this issue, there is still room for improvement. Future work could explore advanced generalization techniques and the integration of chemical inductive biases to enhance performance on unseen scaffolds. Additionally, developing a more comprehensive evaluation benchmark with a stricter scaffold split would provide deeper insights into model generalization. We leave these for future research.

**Explore Randomized SELFIES Data Augmentation in Pretraining.** Although randomized SELFIES augmentation shows promising results when fine-tuning MoLlama for 1D molecule generation, we do not use this augmentation technique during pretraining due to our limited computational resources. We believe applying this technique in pretraining could lead to different outcomes. We leave this exploration for future work.

**Explore Pretrained Molecular Large LM with Bi-directional Self-Attention.** MoLlama uses causal self-attention, where each token can only attend to previous tokens. While this approach is a good fit for 1D molecule generation, it constrains MoLlama's potential for molecule representation learning. To mitigate this issue, we have attached a bi-directional self-attention layer after MoLlama (*cf.* Figure 3). However, a more natural solution would be to use a molecular LM with built-in bi-directional self-attention. Due to resource constraints, we do not pursue this, and existing works are often limited in scale (Irwin et al., 2022; Zheng & Tomiura, 2024). We hope this work draws more attention to this area and encourages the development of more foundation models for biochemistry.

**Explore NExT-Mol for Struture-based Molecule Generation.** We do not explore NExT-Mol for structure-based molecule generation (Zhang et al., 2023) due to the limited scope of this work. However, NExT-Mol could be extended for this task by conditioning the generation process on the structural embeddings of target pockets, potentially using techniques like cross-attention, adaptive layer normalization, or soft-prompting (Li & Liang, 2021). We leave this for future works.

**Limited Exploration on Diffusion Guidance.** Our DMT model utilizes i.i.d. sampling, without exploring advanced sampling method like classifier guidance (Dhariwal & Nichol, 2021) and particle guidance (Corso et al., 2024). However, particle guidance demonstrates that a well-tuned guidance method can improve the conformer prediction by 10% precision. This is because the 3D molecular conformational space is large, and a guidance method with appropriate chemical inductive bias can improve the sampling efficiency. We leave this exploration as a future work.

**Computational Cost when Incorporating MoLlama for 3D Conformer Prediction.** Incorporating MoLlama, a large LM with 960M parameters, increases training time. For example, training DMT-B alone (55M parameters) takes 52 seconds per epoch on an A100 GPU, while DMT-B *with* MoLlama takes 210 seconds. We mitigated this problem by using a pretrained DMT-B, instead of training it from scratch, to reduce the training epochs when incorporating MoLlama. Yet, we will need improvement when transferring 1D representations from a large LM.

**Quadratic Memory Complexity of DMT's Pair Representation.** This pair representation incurs an additional $O(N^2)$ GPU memory cost than the standard transformer, compared to the standard transformer's $O(N)$ memory complexity when using FlashAttention, where N is the node number of molecular graphs. While we encountered no memory issues on the GEOM-DRUGS dataset (molecules with hundreds of nodes), this could be a bottleneck for molecules with thousands of nodes. Potential solutions include smaller batch sizes and model parallelism.

**More Future Works.** In future, we plan to build multi-modal foundation models (Liu et al., 2024c; Li et al., 2024a) with NExT-Mol as the essential backbone for 3D molecule generation (Luo et al., 2023), in order to support tasks like chemical reaction prediction (Liu et al., 2024b; Shi et al., 2023) and drug-drug interaction prediction (Fang et al., 2024a). This is to support the broad application of LLMs for scientific discovery (Cai et al., 2024; Shi et al., 2024) and multi-modal LLMs (Li et al., 2024b). We will also explore model editing techniques (Fang et al., 2025) to locate the influential parameters and update its knowledge according to the application requirement, and explore causal

interventional methods (Li et al., 2021) to find the rationale part (Li et al., 2023a;b;c; 2022b;c) of 2D molecules that influence 3D conformer prediction.

## B  MORE EXPERIMENTAL RESULTS

### B.1  ABLATION STUDY

**Ablating MoLlama Pretraining.** As Table 9 shows, pretraining significantly improves MoLlama's performances on the 1D distribution similarity metrics of SNN, Scaf and FCD, but slightly decreases novelty score (V&U&N). This may be because the model without pretraining prefers a more random sampling, increasing the novelty but reducing the similarity to the desired molecule distribution. Pretraining does not significantly influence stability and validity measures, because they are mostly guaranteed by the SELFIES representations.

Table 9: Ablation study for the MoLlama pretraining for 1D molecule generation on the GEOM-DRUGS dataset.

| Method | FCD↓ | AtomStable | MolStable | V&C | V&U | V&U&N | SNN | Frag | Scaf |
|---|---|---|---|---|---|---|---|---|---|
| MoLlama | **0.334** | **1.000** | **0.999** | **1.000** | **0.999** | 0.945 | **0.529** | **0.999** | **0.552** |
| *w/o* pretraining | 0.586 | **1.000** | 0.995 | **1.000** | **0.999** | **0.974** | 0.495 | **0.999** | 0.534 |

Table 10: Ablating random rotation augmentation for 3D conformer prediction on GEOM-QM9.

| Method | COV-R (%)↑ | | AMR-R ↓ | | COV-P (%)↑ | | AMR-P ↓ | |
|---|---|---|---|---|---|---|---|---|
| | Mean | Median | Mean | Median | Mean | Median | Mean | Median |
| DMT-B | **95.2** | **100.0** | **0.090** | **0.036** | **93.8** | **100.0** | **0.108** | **0.049** |
| *w/o* rand rot aug. | **95.2** | **100.0** | 0.095 | 0.040 | 93.3 | **100.0** | 0.113 | 0.053 |

**Random Rotation Augmentation.** Table 10 shows that DMT benefits from random rotation augmentations. Unlike MCF (Wang et al., 2024), which relies on fixed canonical rotations, this is a key improvement because real data may be out-of-distribution and do not follow canonical rotations.

### B.2  MOLECULE PROPERTY PREDICTION RESULTS FOR MOLLAMA

**Experimental Settings.** To evaluate MoLlama's capabilities beyond 1D molecule generation, we apply it to molecular property prediction tasks (Liu et al., 2023b; Li et al., 2022a), highlighting the quality of its molecular representations. Following the setup in (Rollins et al., 2024), we fine-tune MoLlama on four MoleculeNet (Wu et al., 2018) datasets: FreeSolv, ESOL, Lipo, and QM7. We adopt the same experimental settings and dataset splits as (Rollins et al., 2024), reporting mean performance and standard deviation over 10 random seeds. Following Section 3.3, we attach a single-layer bi-directional self-attention layer after MoLlama to improve its encoding ability. After that, we apply a linear layer on the mean embedding of all molecule tokens for property prediction. For each run, MoLlama is trained for 100 epochs, with test performance selected based on the validation dataset. We use a fixed learning rate of 1e-4 with the AdamW optimizer, and fine-tune MoLlama using LoRA (Hu et al., 2021) (LoRA $r = 8$ and $\alpha = 32$) applied to all linear layers of the model. Following Section 3.1, we apply the random SELFIES augmentation during training. During inference, we use the average prediction values of 20 differently augmented SELFIES as the final prediction.

**Observation.** As shown in Table 11, MoLlama significantly outperforms baseline methods, achieving relative improvements of 6.5%, 4.7%, 3.5%, and 16.9% on the FreeSolv, ESOL, Lipo, and QM7 datasets, respectively. Notably, our baselines include LM-based, GNN-based, and pretrained GNN-based methods, and MoLlama's better performance demonstrates its advantages derived from the extensive pretraining.

Table 11: Molecule property regression results on four MoleculeNet datasets (Wu et al., 2018). Baseline results are from (Rollins et al., 2024). Lower↓ is better.

| Method | FreeSolv (RMSE) | ESOL (RMSE) | Lipo (RMSE) | QM7 (MAE) |
|---|---|---|---|---|
| **Supervised Learning Methods** | | | | |
| RF (Wang et al., 2022) | 2.03±0.22 | 1.07±0.19 | 0.88±0.04 | 122.7±4.2 |
| SVM (Wang et al., 2022) | 3.14±0.00 | 1.50±0.00 | 0.82±0.00 | 156.9±0.0 |
| **Supervised GNN-based Methods** | | | | |
| GCN (Kipf & Welling, 2017) | 2.87±0.14 | 1.43±0.05 | 0.85±0.08 | 122.9±2.2 |
| GATv2 (Brody et al., 2022) | 3.14±0.00 | 1.41±0.00 | 0.89±0.00 | 113.3±0.0 |
| GIN (Xu et al., 2019) | 2.76±0.18 | 1.45±0.02 | 0.85±0.07 | 124.8±0.7 |
| SchNet (Schütt et al., 2018) | 3.22±0.76 | 1.05±0.06 | 0.91±0.10 | 74.2±6.0 |
| 3D Infomax (Stärk et al., 2022) | 2.23±0.26 | 0.95±0.04 | 0.74±0.01 | - |
| MGCN (Lu et al., 2019) | 3.35±0.01 | 1.27±0.15 | 1.11±0.04 | 77.6±4.7 |
| D-MPNN (Yang et al., 2019) | 2.18±0.91 | 0.98±0.26 | 0.65±0.05 | 105.8±13.2 |
| **Pretrained GNN-based Methods** | | | | |
| Pretrain-GNN (Hu et al., 2020) | 2.83±0.12 | 1.22±0.02 | 0.74±0.00 | 110.2±6.4 |
| MolCLR (Wang et al., 2022) | 2.20±0.20 | 1.11±0.01 | 0.65±0.08 | 87.2±2.0 |
| **LM-based Methods** | | | | |
| ChemBERTa-2 (Ahmad et al., 2022) | 2.047±0.00 | 0.889±0.00 | 0.798±0.00 | 172.8±0.00 |
| MolPROP (Rollins et al., 2024) | 1.70±0.09 | 0.777±0.02 | 0.733±0.02 | 151.8±10.0 |
| MoLlama, ours | **1.59±0.04** | **0.740±0.01** | **0.627±0.01** | **63.5±1.6** |

Table 12: Incorporating MoLlama's 1D representations to improve DMT's 3D conformer prediction.

| Dataset | Method | COV-R (%)↑ | | AMR-R ↓ | | COV-P (%)↑ | | AMR-P ↓ | |
|---|---|---|---|---|---|---|---|---|---|
| | | Mean | Median | Mean | Median | Mean | Median | Mean | Median |
| GEOM-QM9 | DMT-B | 95.2 | **100.0** | 0.090 | **0.036** | 93.8 | **100.0** | 0.108 | 0.049 |
| | +MoLlama | **95.6** | **100.0** | **0.083** | **0.036** | **94.2** | **100.0** | **0.097** | **0.044** |

## B.3 3D MOLECULAR CONFORMER PREDICTION

Table 12 presents the results of integrating MoLlama's pretrained 1D representations into DMT-B for 3D conformer prediction, using the same experimental setup as Table 5. The results demonstrate that MoLlama's pretrained representations can enhance DMT-B's performance.

## B.4 INFLUENCE OF HYPERPARAMETERS

**More Results on the Predictor-Corrector Sampler.** Table 4 reports DMT-B's performance with the predictor-corrector sampler with the hyperparameter of snr=0.3. To provide a more comprehensive analysis, Table 13 presents results for additional hyperparameter settings.

**Different Noise Schedules at Inference Time.** We test DMT-B's robustness to different noise schedulers at inference, using two representative options: the linear (Ho et al., 2020) and polynomial (Hoogeboom et al., 2022) schedulers. The original noise scheduler, based on the cosine function, follows (Nichol & Dhariwal, 2021). In this study, we use the existing DMT-B checkpoint without retraining the model with these new schedulers, so the results are suboptimal.

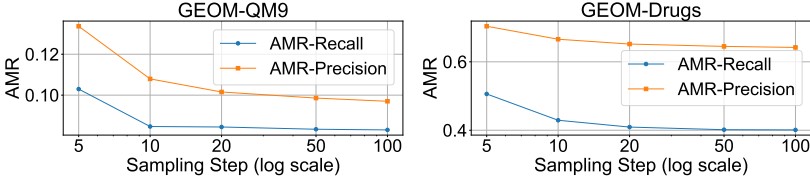

Figure 5: Effect of sampling steps on AMR↓ for 3D conformer prediction using DMT-B.

**Observation.** As shown in Table 14, the polynomial scheduler achieves performance close to the cosine scheduler, likely because their curve shapes are similar. However, the linear scheduler results

Table 13: Performances of 3D conformer prediction on the GEOM-DRUGS dataset.

| | COV-R (%)↑ | | AMR-R ↓ | | COV-P (%)↑ | | AMR-P ↓ | |
|---|---|---|---|---|---|---|---|---|
| Model | Mean | Median | Mean | Median | Mean | Median | Mean | Median |
| DMT-B, PC samp., snr=0.2 | 85.3 | 91.5 | 0.398 | 0.372 | 66.5 | 69.2 | 0.633 | 0.560 |
| DMT-B, PC samp., snr=0.3 | 85.5 | 91.2 | 0.396 | 0.370 | 67.6 | 71.5 | 0.623 | 0.546 |
| DMT-B, PC samp., snr=0.4 | 73.8 | 79.9 | 0.535 | 0.501 | 68.0 | 72.1 | 0.621 | 0.548 |

Table 14: DMT-B's 3D conformer prediction performances on the GEOM-DRUGS dataset when using different noise schedulers at inference time.

| | COV-R (%) ↑ | | AMR-R ↓ | | COV-P (%) ↑ | | AMR-P ↓ | |
|---|---|---|---|---|---|---|---|---|
| Noise schedule | Mean | Median | Mean | Median | Mean | Median | Mean | Median |
| linear | 62.7 | 62.7 | 0.648 | 0.634 | 60.3 | 60.6 | 0.726 | 0.624 |
| cosine, original | 85.4 | 92.2 | 0.401 | 0.375 | 65.2 | 67.8 | 0.642 | 0.577 |
| polynomial | 84.9 | 91.7 | 0.454 | 0.421 | 64.5 | 66.2 | 0.685 | 0.619 |

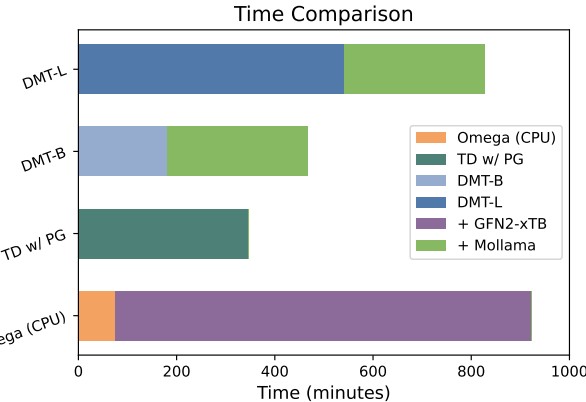

Figure 6: Comparison of conformer generation time on the test set of the GEOM-Drugs dataset using various methods.

in a significant performance drop, suggesting that retraining DMT-B with the linear scheduler is necessary to achieve better results.

**Sampling Steps.** We evaluate 3D conformer prediction performance given different sampling steps.

**Observation.** As shown in Figure 5, we observe an improving trend in AMR for both recall and precision as the sampling steps increase from 5 to 100. The most significant improvements occur between 5 and 20 steps, with diminishing returns beyond 50 steps. This indicates that our model can half the inference cost by trading off a small amount of performance.

**The Influence of Batch Size to 3D Conformer Prediction.** We evaluate the performance of DMT-B with different batch sizes. The original batch size of 256 was chosen to maximize GPU utilization. To assess the impact of batch size, we tested two variations: (1) reducing the batch size to 128, and (2) increasing it to 512 using gradient accumulation.

**Observation.** As shown in Table 15, the performance with a 512 batch size is slightly worse than the original model. This is likely due to underfitting caused by fewer training steps. We keep the number of training epochs the same as the original experiment (256 batch size), therefore the larger batch size results in fewer gradient updates, leading to reduced model performance. Other than this observation, using the 128 batch size does not lead to significant difference than the original model.

Table 15: DMT-B's 3D conformer prediction performances on the GEOM-DRUGS dataset when using different batch sizes.

| Batch size | COV-R (%) ↑ | | AMR-R ↓ | | COV-P (%) ↑ | | AMR-P ↓ | |
|---|---|---|---|---|---|---|---|---|
| | Mean | Median | Mean | Median | Mean | Median | Mean | Median |
| 128 | 85.5 | 92.4 | 0.395 | 0.366 | 65.1 | 68.0 | 0.644 | 0.575 |
| 256, original | 85.4 | 92.2 | 0.401 | 0.375 | 65.2 | 67.8 | 0.642 | 0.577 |
| 512 | 85.1 | 92.0 | 0.410 | 0.377 | 64.9 | 67.7 | 0.645 | 0.582 |

Table 16: 3D Molecule stability performances. * denotes our reproduced results.

(a) GEOM-DRGUS dataset.

| 3D-Metric | MolStable |
|---|---|
| Train | 0.028 |
| EDM | 0.002 |
| JODO | 0.010 |
| MiDi* | 0.003 |
| EQGAT | 0.025 |
| NExT-Mol, ours | **0.027** |

(b) QM9-2014 dataset.

| 3D-Metric | MolStable |
|---|---|
| Train | 0.953 |
| G-SchNet | 0.681 |
| G-SphereNet | 0.134 |
| EDM | 0.817 |
| MDM | 0.896 |
| JODO | 0.934 |
| MiDi* | 0.842 |
| EQGAT | 0.889 |
| NExT-Mol, ours | **0.946** |

## B.5 COMPUTATIONAL TIME COMPARISON

We conducted a time comparison between our model and representative baselines for conformer generation on the test set of the GEOM-Drugs dataset, which includes 1000 molecules. The baselines include the OpenEye Omega (OpenEye, Cadence Molecular Sciences), TD w/ PG (Corso et al., 2024), and xTB[1]. The results are shown in Figure 6.

These experiments were performed on a platform with an 8-core Intel Xeon Processor@2.90GHz CPU and an NVIDIA A100 GPU and the time is measured in minutes and seconds. Please note that the Omega and xTB are run on the CPU only, while DMT and Mollama are run on the GPU. So the results may vary depending on the hardware.

## B.6 3D MOLECULAR STABILITY PERFORMANCE

We do not report the 3D molecule stability metric (Hoogeboom et al., 2022) in the main part of this work, because this metric presents a significant limitation on the GEOM-DRUGS dataset, showing only 2.8% for the ground truth training set. We present the results here for backup purposes.

## B.7 MORE VISUALIZATIONS

**Visualization of Random Samples.** Visualizations of complete molecules sampled from NExT-Mol on GEOM-Drugs and QM9 are shown in Figure 8 and Figure 9, respectively. These samples are randomly selected to illustrate the diversity and effectiveness of our model. The visualization includes 1D SELFIES sequences, 2D molecular graphs, and 3D conformers highlighting the spatial arrangement of atoms within the molecules. Notably, in the complex GEOM-Drugs dataset, NExT-Mol demonstrates its robustness by consistently generating molecules without disconnected components and effectively preserving the stable geometric planes of aromatic ring structures. These visualizations not only demonstrate the fidelity of the molecules generated by NExT-Mol with 1D SELFIES sequences along with 3D spatial coordinates, but also emphasize the ability of our model to produce stable and chemically valid conformers accommodating a wide range of molecular weights.

**Visualization of 3D Conformer Prediction.** To gain more insights on how transfer learning using MoLlama's 1D representations can improve 3D conformer prediction, we present more visu-

---

[1]https://xtb-docs.readthedocs.io/en/latest/

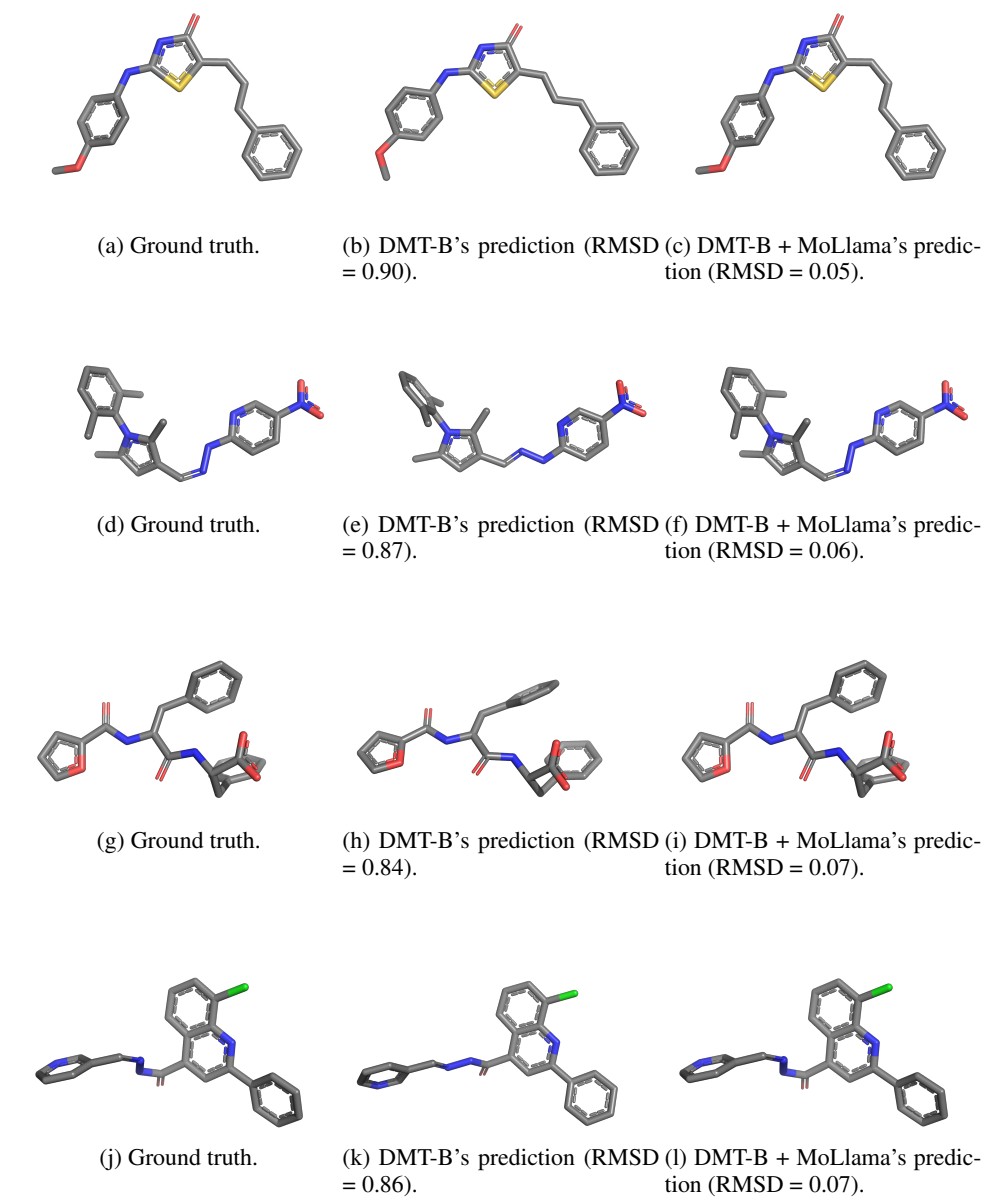

(a) Ground truth.

(b) DMT-B's prediction (RMSD = 0.90).

(c) DMT-B + MoLlama's prediction (RMSD = 0.05).

(d) Ground truth.

(e) DMT-B's prediction (RMSD = 0.87).

(f) DMT-B + MoLlama's prediction (RMSD = 0.06).

(g) Ground truth.

(h) DMT-B's prediction (RMSD = 0.84).

(i) DMT-B + MoLlama's prediction (RMSD = 0.07).

(j) Ground truth.

(k) DMT-B's prediction (RMSD = 0.86).

(l) DMT-B + MoLlama's prediction (RMSD = 0.07).

Figure 7: Visualization of 3D conformers. From left-to-right, we have the ground truth conformer, the conformer predicted by DMT-B, and the conformer predicted by DMT-B+MoLlama. For each model, we select the predicted conformer with the least RMSD to the ground truth.

alizations of 3D conformer prediction in Figure 7. The samples are selected from the test set of GEOM-DRUGS with unseen scaffolds in the training set.

## C FURTHER DETAILS ON METHODOLOGY

### C.1 1D MOLECULE GENERATION WITH MOLECULAR LLAMA LM

**Data Preparation.** Following (Irwin et al., 2022), we collect 1.8 billion molecules from the ZINC-15 database (Sterling & Irwin, 2015), significantly more than the 100 million molecules used in previous studies (Irwin et al., 2022; Fang et al., 2024b). We keep only molecules with molecular

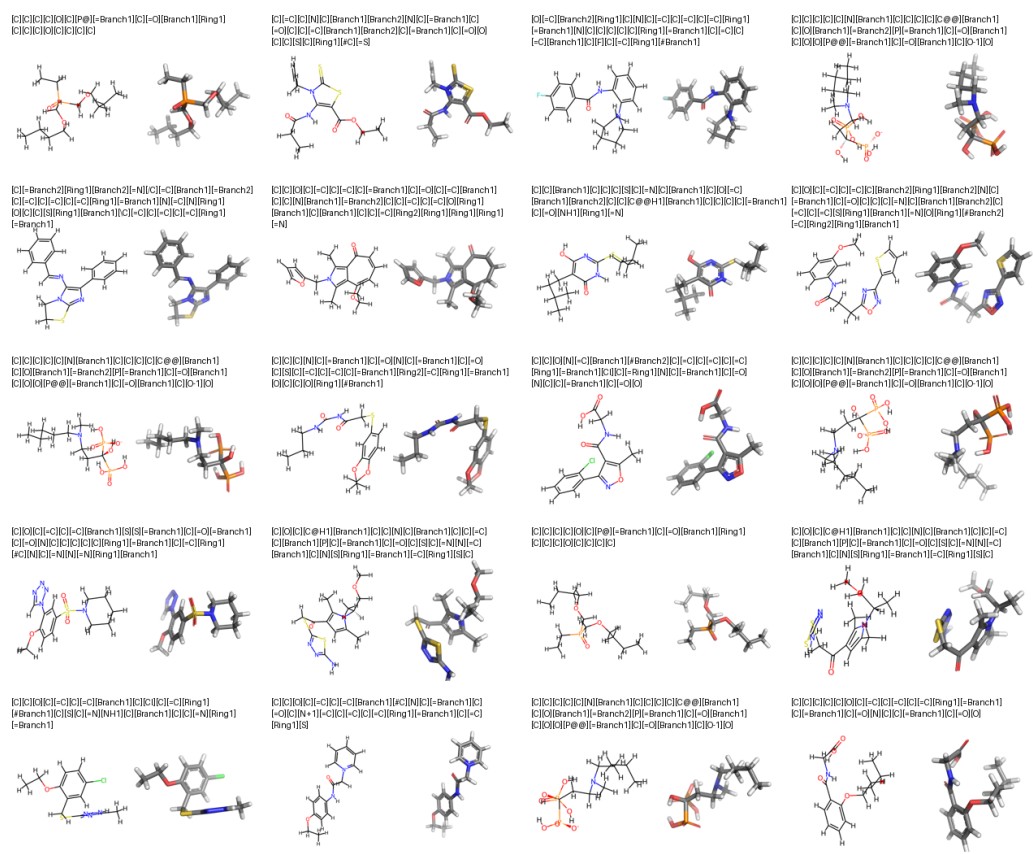

Figure 8: Visualization of random samples generated by NExT-Mol trained on GEOM-DRUGS.

Table 17: Hyperparameter for pretraining MoLlama.

| | | | |
|---|---|---|---|
| hidden size | 2048 | hidden act | silu |
| intermediate size | 5632 | batch size | 512 |
| max position embeddings | 512 | warmup steps | 2000 |
| num attention heads | 32 | min lr | 4.00E-05 |
| num hidden layers | 22 | init lr | 4.00E-04 |
| num key value heads | 4 | weight decay | 1.00E-01 |
| n query groups | 4 | grad clip | 1.0 |

weight≤500 Daltons and LogP≤5 (Flynn, 1980), and transform them into SELFIES (Krenn et al., 2020) sequences. After canonicalizing the SELFIES and removing hydrogen atoms, the dataset contains 90 billion tokens. We further filter the molecules in the valid and test sets of the GEOM-QM9 and GEOM-DRUGS datasets (Axelrod & Gomez-Bombarelli, 2022) and randomly sampled 1% of the remaining data as the validation set.

**Randomized SELFIES Augmentation Details.** In order to generate randomized SELFIES, we first generate the randomized SMILES (Weininger, 1988), and transform the SMILES into SELF-IES. We follow (Arús-Pous et al., 2019) for the implementation details of random SMILES, and use a restricted random sampling of SMILES. Similarly, we also generate canonical SELFIES by transforming canonical SMILES.

**Pretraining Details.** We train MoLlama from scratch for 1D molecule generation using a next-token prediction objective. The code and hyperparameters are based on (Zhang et al., 2024), utilizing Flash-Attention (Dao, 2024) and FSDP (Zhao et al., 2023) for faster training. We use a max context length of 512, concatenating multiple SELFIES sequences into the same context, with any overflow

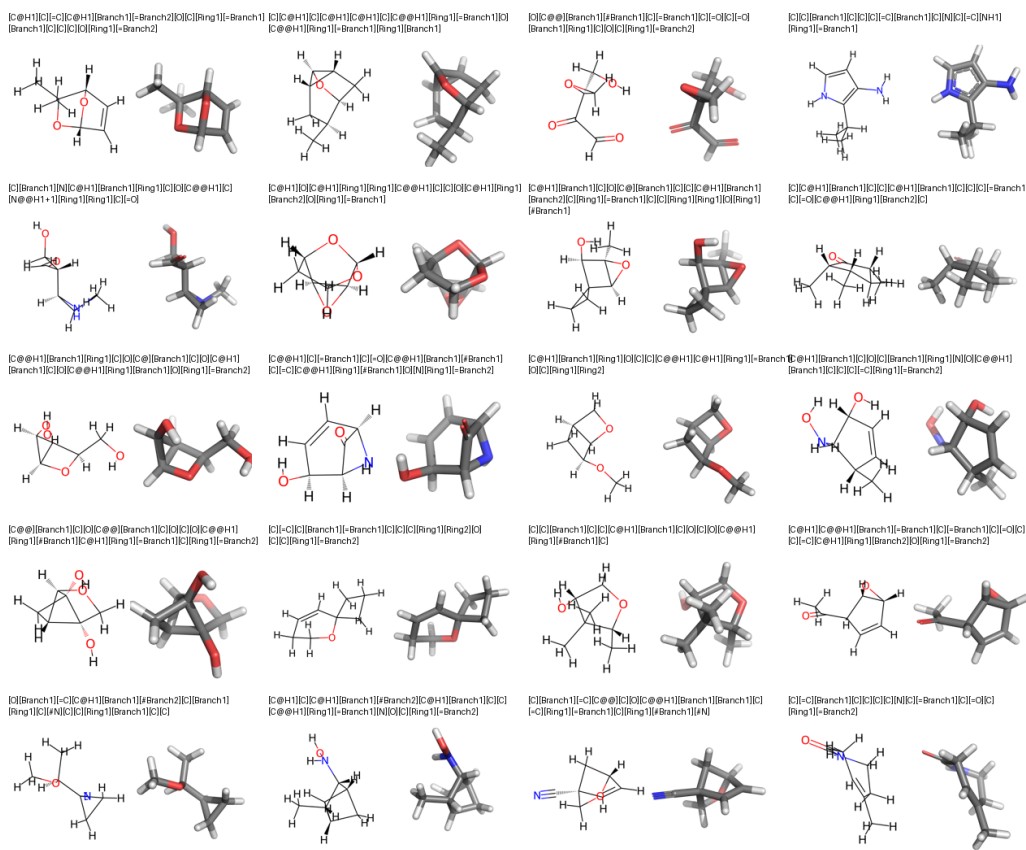

Figure 9: Visualization of random samples generated by NExT-Mol trained on QM9-2014.

trimmed and used in the next context. We use the AdamW optimizer and a scheduler with linear warmup and cosine decay. The key parameters are included in Table 17. We train the model for 555k global steps. The training was done on 4 NVIDIA A100-40G GPUs and took approximately two weeks. The training log is shown in Figure 10.

**On the Advantages of Acheiving 100% Validity beyond Validity Itself.** We employ the 1D SELF-IES representation for LM training. Here we elaborate on the other advantages beyond 100% validity, which are also crucial for real-world applications:

- **Improving validity could improve other 2D metrics, like SNN, Frag, and Scaf.** These metrics measure the distributional similarity of 2D molecular structures of valid molecules. If a model still generate invalid molecules, it is likely the model does not capture the true target distribution, which contain only valid molecules. 100% validity helps the model learn from and sample from the valid molecular structures, which is essential for molecule generation tasks. This is demonstrated by our improved FCD, SNN, Frag, and Scaf metrics in Table 2.

- **Improving validity could improve 3D geometry learning.** The improved validity also leads to better learning of 3D molecular geometry, because it grounds 3D structure prediction on valid 2D structures. Other joint 2D and 3D prediction methods (Huang et al., 2024; Vignac et al., 2023b) can easily encounter invalid 2D structures when sampling 3D structures, therefore leads to worse 3D structure prediction. This is demonstrated by NExT-Mol's significant improvements in geometry similarity metrics (*e.g.,* bond angle and bond length) in Table 2.

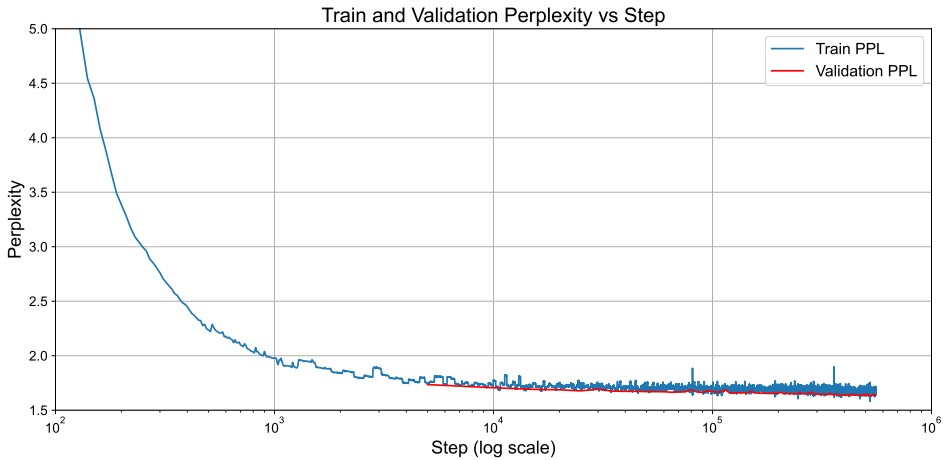

Figure 10: Visualization of MoLlama's training and validation PPL log during pretraining.

## C.2 3D CONFORMER PREDICTION WITH DIFFUSION MOLECULAR TRANSFORMER

**Diffusion Process.** Here we elaborate on the details of our diffusion process. Following (Nichol & Dhariwal, 2021; Huang et al., 2024), we use the cosine scheduler controlling the noise scale for the diffusion process:

$$\bar{\alpha}_t = \frac{f(t)}{f(0)}, \quad f(t) = \cos\left(\frac{t+s}{1+s} \cdot \frac{\pi}{2}\right), \tag{5}$$

where $t \in (0,1]$ is the time step, and $s$ is a hyperparameter empirically set to 0.008, following (Nichol & Dhariwal, 2021).

Our pseudo codes for training and sampling are shown in Algorithm 1 and Algorithm 2 below. Following (Ho et al., 2020), we have the following hyperparameters used in the pseudo-codes for training and sampling:

$$\alpha^{(t)} = \bar{\alpha}^{(t)}/\bar{\alpha}^{(t-1)}, \quad \sigma^{(t)} = \sqrt{1 - \alpha^{(t)}}. \tag{6}$$

---

**Algorithm 1** Training

---

1: $t \sim \mathcal{U}(0,1]$     {Sample a time step}
2: $G^{(0)} = (\mathbf{x}^{(0)}, \mathbf{h}, \mathbf{e}) \sim$ Training Set     {Sample a 3D molecule}
3: $\mathbf{x}^{(0)} \leftarrow \mathbf{x}^{(0)} - \bar{\mathbf{x}}^{(0)}$     {Centering molecule coordinates}
4: $\mathbf{x}^{(0)} \leftarrow \mathbf{x}^{(0)}R$, where $R \in SO(3)$ is randomly sampled     {Random rotation augmentation}
5: $\boldsymbol{\epsilon}^{(t)} \sim \mathcal{N}(\mathbf{0}|\mathbf{I})$
6: $\mathbf{x}^{(t)} = \sqrt{\bar{\alpha}^{(t)}}\mathbf{x}^{(0)} + \sqrt{1 - \bar{\alpha}^{(t)}}\boldsymbol{\epsilon}^{(t)}$     {Forward diffusion}
7: $G^{(t)} \leftarrow (\mathbf{x}^{(t)}, \mathbf{h}, \mathbf{e})$
8: Minimize loss $\mathcal{L} = ||\boldsymbol{\epsilon}^{(t)} - \text{DMT}(G^{(t)}, t)||_2^2$

---

**RMHA.** Here we define the multi-head version of RMHA. Similar to the single-head version, we first generate the queries, keys, and values for atom representation $\mathbf{H}$, and generate the queries and values for pair representation $\mathbf{E}$:

$$[\mathbf{Q}; \mathbf{K}; \mathbf{V}] = [\mathbf{W}_q; \mathbf{W}_k; \mathbf{W}_v]\mathbf{H}^\top, \quad (7) \qquad [\mathbf{Q}^E; \mathbf{V}^E] = \tanh([\mathbf{W}_{eq}; \mathbf{W}_{ev}]\mathbf{E}^\top), \quad (8)$$

---

**Algorithm 2** Sampling 3D Conformers

---

**Require:** time steps $\{t_i\}_{i=1}^M$, a 2D molecular graph $G_{2D} \leftarrow (\mathbf{h}, \mathbf{e})$

1: $\mathbf{x}^{(t_1)} \sim \mathcal{N}(\mathbf{0}, \mathbf{I})$          {Set the initial noise conformer}
2: **for** $i \leftarrow 1$ to $M$ **do**
3:     $t \leftarrow t_{i-1}$, $s \leftarrow t_i$          {Set time step}
4:     $G^{(t)} \leftarrow (\mathbf{x}^{(t)}, \mathbf{h}, \mathbf{e})$
5:     $\mathbf{z} \sim \mathcal{N}(\mathbf{0}, \mathbf{I})$ if $i < M$ else $\mathbf{z} = \mathbf{0}$
6:     $\mathbf{x}^{(s)} = \frac{1}{\sqrt{\alpha_t}} \left( \mathbf{x}^{(t)} - \frac{1 - \alpha^{(t)}}{\sqrt{1 - \bar{\alpha}^{(t)}}} \text{DMT}(G^{(t)}, t) \right) + \sigma^{(t)} \mathbf{z}$          {Update conformer}
7: **end for**
8: **return** $\mathbf{x}^{(M)}$

---

Subsequently, we define the Relational-Attention (R-Attention) module, which is the combination of Equation 3 and Equation 4:

$$\mathbf{O} = \text{R-Attention}(\mathbf{Q}, \mathbf{K}, \mathbf{V}, \mathbf{Q}^E, \mathbf{V}^E), \tag{9}$$

$$\text{where } \mathbf{O}_i = \sum_{j=1}^N a_{i,j}(\mathbf{V}_{i,j}^E \odot \mathbf{V}_j), \tag{10}$$

$$a_{i,j} = \text{softmax}_j\left(\frac{(\mathbf{Q}_{i,j}^E \odot \mathbf{Q}_i)\mathbf{K}_j^\top}{\sqrt{d}}\right). \tag{11}$$

After this, the muli-head version of RMHA can be written as:

$$\text{RMHA}(\mathbf{Q}, \mathbf{K}, \mathbf{V}, \mathbf{Q}^E, \mathbf{V}^E) = \text{Concat}(\mathbf{O}^1, ..., \mathbf{O}^h)\mathbf{W}_o \tag{12}$$

$$\text{where } \mathbf{O}^f = \text{R-Attention}(\mathbf{W}_{qf}\mathbf{Q}, \mathbf{W}_{kf}\mathbf{K}, \mathbf{W}_{vf}\mathbf{V}, \mathbf{W}_{eqf}\mathbf{Q}^E, \mathbf{W}_{evf}\mathbf{V}^E), \tag{13}$$

where $h$ is the number of head; $f \in [1, h]$; $\mathbf{W}_o$ is the linear projector combining outputs of different heads; and $\mathbf{W}_{qf}$, $\mathbf{W}_{kf}$, and $\mathbf{W}_{vf}$ are linear projectors for the $f$-th head of atom representations; and $\mathbf{W}_{eqf}$ and $\mathbf{W}_{eqf}$ are linear projectors for the $f$-th head of the pair representation.

### C.3 MoLlama Representations Improve DMT's 3D Conformer Prediction

**Details of SELFIES-to-Atom Mapping.** The mapping process is not straightforward with existing software, so we have to manually code a significant portion. For details on the full implementation, please refer to our code. In brief, the SELFIES software provides a mapping between SELFIES and SMILES tokens, and RDKit gives the atom order when generating SMILES. We manually convert this atom order into a mapping between SMILES and atom indices, then combine the SELFIES-to-SMILES and SMILES-to-atom mappings into the SELFIES-to-atom mapping. Additionally, we handle missing hydrogen atoms in both SMILES and SELFIES during the mapping process.

**Rationale behind Transfer Learning between 1D Molecule Sequences and 3D Conformers.** The final goal of this transfer learning is to leverage the billion-scale 1D/2D molecule dataset to improve the 3D conformer prediction performance, which is constrained by limited 3D data. For clarity, we decompose the rationale into the following chain of arguments:

- **3D conformers are theoretically governed by 2D molecular graphs under quantum mechanics (QM).** 3D molecular properties and structures are fundamentally rooted in QM. Using (approximated) QM-based methods, like DFT, we can accurately predict 3D conformers from 2D molecular graphs, though at high computational cost. This establishes the critical role of 2D representations in determining 3D structures.

- **3D conformer prediction relies on high quality 2D molecule representations.** Deep learning models predict 3D conformers from 2D graphs, and their performance is heavily influenced by the quality of 2D molecular representations. Transfer learning can enhance 2D molecular representations, as demonstrated by prior works (Hu et al., 2020; Liu et al., 2022; Hou et al., 2022).

- **1D molecular representations can be converted to 2D molecular representations, and contribute to 3D prediction.** 1D molecule sequences encode the same information as 2D molecular graphs, and the 1D to 2D transformation can be achieved by deterministic toolkit, like RDkit. Leveraging RDkit and our proposed cross-modal projector (*cf.* Section 3.3), we can transform 1D molecular representations to 2D molecular representations, and therefore contribute to the 3D prediction. We have demonstrated this improvement in Table 5, where using the pretrained 1D representations improve 3D conformer prediction.

- **1D pretraining scales more effectively than 2D.** Given the billion-scale 1D/2D molecule dataset, we mostly prioritize the scalability when selecting the pretraining method. After literature review, we find that 1D LM-based pretraining methods, like Llama (Touvron et al., 2023) and BERT (Devlin et al., 2019), are extensively demonstrated for scalability and effectiveness. Therefore, we opt to 1D pretraining instead of 2D pretraining.

# D EXPERIMENTAL DETAILS

## D.1 BASELINES

Here we present a brief introduction for the baselines used in our experiments. We categorize baselines by their benchmarks.

***De Novo* and Conditional 3D Molecule Generation.**

- G-SchNet (Gebauer et al., 2019): G-SchNet autoregressively generates 3D molecules by considering molecular symmetries through the SchNet (Schütt et al., 2018)

- G-SphereNet (Luo & Ji, 2022): G-SphereNet autoregressively generates 3D molecules, in which each step determines the atom type, bond length, angle, and torsion angles.

- EDM (Hoogeboom et al., 2022): EDM pioneers the diffusion methods for 3D molecue generation. It constructs a diffusion model with the EGNN (Satorras et al., 2021) architecture and the VDM diffusion process (Kingma et al., 2021).

- MDM (Huang et al., 2023b): MDM is a diffusion model for 3D molecule generation. Through a specialized edge construction module, it leverages both global interatomic interactions and local interatomic interactions for 3D modeling.

- CDGS (Huang et al., 2023a) and JODO (Huang et al., 2024): CDGS is a diffusion model for 2D molecular graph generation. It models discrete vairables (*e.g.,* atom types and bond types) using one-hot encoding and applies a continous diffusion for generative modeling. JODO extends CDGS by studying joint 2D and 3D molecule generation. It features a RMHA module for enhanced relational molecular graph modeling.

- MiDi (Vignac et al., 2023b): MiDi is a joint 2D and 3D diffusion model for 3D molecule generation. It leverages two diffusion processes of discrete diffusion (Vignac et al., 2023a) and continous diffusion (Hoogeboom et al., 2022) for the corresponding data types in a molecule.

- EQGAT-diff (Le et al., 2024): EQGAT-diff modified the EQGAT (Le et al., 2022) architecture for joint 2D and 3D molecular generation. EQGAT is based on the Tensor Field Networks (Thomas et al., 2018) to achieve 3D rotational and translational equivariance.

- GeoLDM (Xu et al., 2023): GeoLDM explores the idea of latent diffusion model (Rombach et al., 2022) for 3D molecule generation.

- EEGSDE (Bao et al., 2023): EEGSDE explores conditional 3D molecule generation with diffusion guidance by an energy function.

- MolGPT (Bagal et al., 2021): MolGPT is a decoder-only molecule LM pretrained on 1D SMILES sequences.

- MolGen (Fang et al., 2024b): MolGen is an encoder-decoder molecular LM pretrained on 1D SELFIES sequences. Following (Raffel et al., 2020), it is pretrained and evaluated using a span-corruption objective.

**3D Conformer Prediction.**

Table 18: Hyperparameters of the DMT-B and DMT-L models.

| | DMT-B | DMT-L |
|---|---|---|
| n layers | 10 | 12 |
| atom hidden size | 512 | 768 |
| atom intermediate size | 2048 | 3072 |
| pair hidden size | 128 | 192 |
| pair intermediate size | 512 | 768 |
| n heads | 8 | 8 |
| total params | 55M | 150M |
| optimizer | AdamW | |
| init lr | 1.00E-04 | |
| min lr | 1.00E-05 | |
| warmup lr | 1.00E-06 | |
| warmup steps | 1000 | |
| weight decay | 0.05 | |

- OMEGA (Hawkins, 2017): OpenEye OMEGA is a commercial software that employs a combination of fragment-based methods and torsional sampling, guided by empirical force fields or customized energy functions, to predict 3D conformers.

- GeoMol (Ganea et al., 2021): GeoMol is an SE(3)-invariant model for 3D conformer prediction. In the first step, it predicts the bond angles and bond lengths for all the neighbors of each non-terminal atom. Next, it assembles the local structures together by predicting their torsion angles.

- GeoDiff (Xu et al., 2022): GeoDiff is a diffusion model that leverages a roto-translational equivariant GNN for 3D conformer prediction.

- Torsional Diffusion (Jing et al., 2022): Torsional diffusion is a diffusion model defined on the dihedral angles of 3D molecules. It samples seed conformers using RDkit, and applies diffusion only on the dihedral angles of molecular bonds, while fixing the bond lengths and bond angles.

- Particle Guidance (Corso et al., 2024): Particle guidance is a diffusion guidance method designed to improve the sampling diversity compared to the vanilla i.i.d. sampling. It modifies torsional diffusion's sampling process for 3D conformer prediction, without changing its training process.

- MCF (Wang et al., 2024): MCF explores the power of scaling law for 3D conformer prediction. Instead of following prior works and leveraging a neural architecture with built-in 3D equivariance, it scales up a general-purpose transformer, and demonstrates strong performances.

### D.2 DMT CONFIGURATIONS

**Hyperparameter.** Table 18 shows the key hyperparameters used for training the DMT-B and DMT-L models. Other hyperparameters, like batch size and training epochs, are separately listed for each task in the following sections.

**Features.** We use the same atom features and pair features as (Jing et al., 2022). For the GEOM-DRUGS dataset, the atom feature has 74 dimensions; for the QM9-2014 and GEOM-QM9 datasets, the atom feature has 44 dimensions. The bond feature has 4 dimensions.

### D.3 TASK: *De Novo* MOLECULE GENERATION

For *De Novo* molecule generation, we separately train NExT-Mol for the GEOM-DRUGS and the QM9-2014 datasets. This process involve training both the MoLlama and DMT of NExT-Mol.

**MoLlama Settings.** For QM9-2014, we use a batch size of 512 and train for 100 epochs, while for GEOM-DRUGS, we use a batch size of 256 and train for 20 epochs. For sampling, we employ a sampling temperature of 1.0 and, beam size of 1, and we sample 10,000 molecules for evaluation. We use the AdamW optimizer and a learning rate scheduler with linear warmup and cosine decay. The optimizer hyperparameters are as follows: init_lr=1e-4, min_lr=1e-5, warmup_lr=1e-6, warmup_steps=1000, and weight_decay=0.05.

**DMT Settings.** We use a dropout rate of 0.1 for QM9-2014 and 0.05 for GEOM-DRUGS. Following (Huang et al., 2024), we select only the conformer with the lowest energy for training on the GEOM-DRUGS dataset. For both datasets, we train DMT-B for 1000 epochs. The batch size for QM9-2014 is 2048 and the batch size for GEOM-DRUGS is 256.

**Details on the Evaluation Metrics.** We use the MMD distance when computing the distributional similarity of bond lengths, bond angles, and dihedral angles. Note that, we do not perform Kekulization and Sanitization when computing molecule and atom stability for 2D and 3D molecules. We use canonicalized SMILES for both the generated molecules and the training dataset when computing novelty and uniqueness of molecules. All the baselines are consistently evaluated under the same setting above.

### D.4  TASK: CONDITIONAL MOLECULE GENERATION

**Details for Adapting NExT-Mol for Conditional Generation.** For conditional molecule generation on the QM9-2014 dataset, we modify the NExT-Mol architecture to incorporate property-specific information into both the MoLlama language model and the DMT conformer prediction model. This approach allows us to generate molecules with desired properties in both 1D sequence and 3D structure spaces.

- **Condioning MoLlama.** We implement a condition MLP to encode property information into a soft prompt. This MLP consists of two linear layers with a GELU activation function in between. It transforms a single property value into a 4-token sequence embedding, each token having the same dimensionality as the model's hidden size. The resulting soft prompt is prepended to the input sequence embeddings of SELFIES before being fed into the language model. We adjust the attention mask accordingly to ensure the model attends to these conditional tokens.

- **Condioning DMT.** We use an MLP to process the property value, followed by a linear projection to match the time embedding dimension. This processed condition is then added to the time embedding, allowing the diffusion process to be guided by the desired property throughout the denoising steps.

**MoLlama Setting.** For conditional molecule generation, we train MoLlama with a batch size of 256 for 100 epochs on the QM9-2014 dataset. We use a sampling temperature of 1.0, beam size of 5, and we sample 10,000 molecules for evaluation of each desired property.

**DMT Setting.** For the DMT-B model, we train with a batch size of 512 for 1000 epochs on the QM9-2014 dataset. We employ a dropout rate of 0 with 100 sampling steps for evaluation.

The optimizer and learning rate schedule are consistent with the *de novo* generation task, using AdamW with a linear warmup followed by cosine decay. We train the conditional generation model for six different quantum properties using the same optimization strategy as in the *de novo* generation task. Each model is trained on 4 NVIDIA A100-80GB GPUs.

### D.5  TASK: 3D CONFORMER PREDICTION

**Training Details.** We elaborate the training details for each of the three training stages in Section 3.3.

- **Stage 1: DMT Training.** For GEOM-QM9, we train the DMT-B model for 2000 epochs with a batch size of 2048. For GEOM-DRUGS, we train both the DMT-B and DMT-L models for 3000 epochs with batch size 256. Note that, for each epoch, we randomly sample a 3D conformer for each molecule, but not enumerate all the 3D conformers of that molecule. The resulting models (*i.e.,* DMT-B and DMT-L) are used directly for evaluation in Table 4.

- **Stage 2: Projector Warmup.** For both datasets, we train only the LoRA weights of MoLlama, and the cross-modal projector for 10 epochs. The pretrained weights of DMT and MoLlama are frozen throughout the process.

- **Stage 3: Integrated Fine-tuning.** For both datasets, we train the integrated model for 500 epochs. We train the LoRA weight of MoLlama, the cross-modal pojector, and the DMT model. The pretrained weights of MoLlama are frozen throughout the process.

**Evaluation.** Following (Wang et al., 2024; Jing et al., 2022), we use the dataset split of 243473/30433/1000 for GEOM-DRUGS and 106586/13323/1000 for GEOM-QM9, provided by (Ganea et al., 2021). For a molecule with $K$ ground truth conformers, we generate $2K$ conformers as predictions.

**Evaluation Metrics.** Let $\{C_l^*\}_{l \in [1,L]}$ be the $L$ predicted conformers and let $\{C_k\}_{k \in [1,K]}$ be the $K$ ground truth conformers. The evaluation metrics AMR-R (AMR-Recall) and COV-R (COV-Recall) can be formally defined as follows:

$$\text{COV-R} := \frac{1}{L} |\{l \in [1..L] : \exists k \in [1..K], \text{RMSD}(C_k, C_l^*) < \delta\}|, \tag{14}$$

$$\text{AMR-R} := \frac{1}{L} \sum_{l \in [1..L]} \min_{k \in [1..K]} \text{RMSD}(C_k, C_l^*), \tag{15}$$

where $\delta$ is a threshold that is set to $0.75\mathring{A}$ for GEOM-DRUGS and set to $0.5\mathring{A}$ for GEOM-QM9, following (Wang et al., 2024; Jing et al., 2022). AMR-P (AMR-Precision) and COV-P (COV-Precision) can be similarly defined by swapping the ground truth conformers and predicted conformers.

