# OpenReview forum: "NExT-Mol: 3D Diffusion Meets 1D Language Modeling for 3D Molecule Generation"
_ICLR.cc/2025/Conference — ICLR 2025 Poster_

### Official Review · Reviewer_iSmq · 2024-10-24

**Soundness:** 3
**Presentation:** 3
**Contribution:** 3
**Rating:** 8
**Confidence:** 4

**Summary:**

The paper introduces Next Mol, an innovative model for 3D molecule generation that combines the strengths of 1D molecule generation using SELFIES representations with subsequent conformer prediction. This approach is particularly timely given the scarcity of 3D annotated molecules in existing databases. While billions of molecules are cataloged in databases like ZINC or Enamine, researchers often rely on datasets such as GEOM DRUGS, which contain only about 400,000 unique molecules with approximately 40 million 3D structures.

Key Contributions:
- Novel Methodology: The integration of 1D molecule generation through SELFIES with conformer prediction addresses the limitations of current datasets and methods in 3D molecule generation.
- Performance Improvements: Next Mol demonstrates significant enhancements on benchmarks for both conformer generation and unconditional 3D molecule generation, outperforming existing models.
- Transfer Learning: The study shows that transfer learning between the stages of 1D molecule generation and conformer prediction positively impacts the results, suggesting a valuable strategy for future research.
- Advancing Beyond Equivariance Restrictions: The proposed DMT (Diffusion Molecular Transformer) model pushes the boundaries of 3D molecule generation by moving beyond the equivariant restrictions that have been prevalent in recent years, potentially opening new avenues in molecular modeling.

**Strengths:**

- State-of-the-Art Conformer Generation: The model achieves state-of-the-art performance in conformer generation, demonstrated through extensive comparisons with popular models like GeoDiff, Torsional Diffusion, and MCF, as well as widely used tools such as RDKit and OpenEye-Omega.
- High Topological Metrics: Utilizing 1D generative models significantly improves topological metrics—such as molecular stability, validity, and uniqueness—elevating them to nearly 100%.
- Scalable Transformer Architecture: The Diffusion Molecular Transformer is a scalable model that employs a simple transformer architecture with proven efficiency, making it an excellent base model for numerous other related tasks.

**Weaknesses:**

Weaknesses:

**Lack of Comparison with Other Molecular Language Models:** Although the paper introduces a 1D molecule generation component (MoLama), a 3D conformer generation model (DMT), and a transfer learning technique, it primarily showcases the performance of the conformer generation part and the advantages of transfer learning. However, it lacks a comparison with other molecular language models concerning the quality of the generated SELFIES representations.

**Overemphasis on 100% Validity:** The paper focuses on achieving 100% validity, but in real-world applications, validity filtering is an extremely simple process due to how validity is defined. Consequently, there is no significant practical difference between achieving 90% validity and 100% validity.

**Mischaracterization of Computational Complexity:** The paper states that structures were obtained using computationally intensive geometry optimization with DFT. However, the GEOM dataset was designed using the CREST software for conformer sampling, followed by geometry optimization with GFN2-xTB—a semi-empirical tight-binding method, not DFT. Moreover, compared to some deep learning models, this approach is not computationally intensive; on a reasonable workstation, geometry optimization takes about 0.5 seconds per average GEOM-Drugs structure.

**Missing Performance Metrics for Conformer Generation:** In conformer generation, computational performance is extremely important. OpenEye Omega remains one of the most popular software tools for this task because of its speed. The paper lacks performance metrics related to speed and efficiency, which are necessary for a fair comparison with existing tools.

**Use of Questionable Metrics:** The metric reported by JODO shows a 2.8% 3D molecule stability for the GEOM-Drugs dataset, rendering it practically meaningless. I strongly encourage avoiding the propagation of this metric in new papers. According to the code, the metric is based on a predefined bond length table, and a bond length is considered "good" if it is within 0.05 Å of the table value. However, the optimal distances between atoms are primarily defined by the energy landscape underlying the data—for GEOM-Drugs, it's GFN2-xTB—and depending on atom configurations, deviations in bond lengths can exceed 10%. While I'm uncertain about the validity of the 3D FCP metric, it's not entirely clear that it's completely off when compared with 3D molecule stability. Instead of attempting to report every possible metric to compare with other methods, focus on identifying the most important ones and emphasize the significance of the margins your model is achieving.

**Inconsistencies in Metric Definitions and Reporting:** Providing detailed descriptions of the metrics used in the supplementary material is crucial due to inconsistencies across different papers. For example, MiDi reported the Wasserstein distance for bond angles and bond length distributions for all bonds and angles, whereas this paper (and at least JODO) uses MMD for the most frequent bonds, angles, and torsions. Additionally, the way MiDi computes atom stability and molecule stability differs from the JODO code. Performing kekulization at the beginning can alter the valencies of atoms, leading to different results (e.g., if you manually define H:O:H, where ":" is an aromatic bond, kekulization converts it to the valid water molecule H-O-H). Some models measure atom and molecule stability for raw data, while others use RDKit preprocessing before measuring stability, which can inflate results. It is essential to be consistent in comparisons. While this may not significantly change your results, it could artificially boost JODO's stability results, skewing the comparison. Overall, ensure that the same versions of the metrics are used across all comparisons and that they are well-defined in the supplementary material.

**Omission of EQGAT-Diffusion in Comparisons:** EQGAT-Diffusion is a recent model—published well before the ICLR deadline—that should be included in the comparisons for 3D unconditional molecule generation to provide a more comprehensive evaluation.

**Questions:**

Questions to the Authors:

**Comparison of MoLama with Other Models:** Could you provide a comparison between your 1D molecule generation model (MoLama) and other models like Equiformer on tasks beyond 3D molecule generation, such as molecule classification (or any suitable task where you can compare MoLama and let's say Equiformer)? Including such a comparison would highlight the standalone value of MoLama.

**Practical Advantages of 100% Validity:** Given that validity filtering is straightforward in practice, what are the practical advantages of achieving 100% validity over, for example, 90% validity in real-world applications?

**Inclusion of Performance Metrics:** Considering that computational performance is crucial in conformer generation tasks, could you include speed and efficiency metrics for your model, particularly in comparison with tools like OpenEye Omega? I would also add OpenEye Omega conformer generation + consequent xTB geometry optimization time.

**Analysis of Metrics Used:** Could you analyze the metrics used in your study, especially focusing on 3D molecular stability and the FCP metric? Since the FCP metric relies on a neural network, is it capable of handling your data distribution, and is it up-to-date with recent developments?

**Consistency and Clarity of Metrics:** To ensure consistency, could you provide detailed descriptions of the metrics used in your study and confirm that the same versions are applied across all comparisons? Additionally, please clarify any preprocessing steps that might affect the results.

**Inclusion of EQGAT-Diffusion in Comparisons:** EQGAT-Diffusion is a recent model relevant to 3D unconditional molecule generation. Could you include it in your comparisons to provide a more comprehensive evaluation?

---

> ### Author Response · Authors · 2024-11-21
> **Response to Reviewer iSmq (Part 1/7)**
>
> Thank you so much for the positive ratings and the extensive comments and suggestions about our submission. Here we provide a point-by-point response to address your concerns and incorporate your suggestions. For clarity, we have consolidated similar questions and weaknesses where appropriate, labeling them with corresponding identifiers (e.g., **W1** for Weakness 1 and **Q1** for Question 1) based on your original comments.
>
> > **W1: Lack of Comparison with Other Molecular Language Models:** Although the paper introduces a 1D molecule generation component (MoLama), a 3D conformer generation model (DMT), and a transfer learning technique, it primarily showcases the performance of the conformer generation part and the advantages of transfer learning. However, it lacks a comparison with other molecular language models concerning the quality of the generated SELFIES representations.
> >
>
> **Response:** Thank you for the suggestions. We agree that including the comparison with other molecular language models (LMs) will provide further empirical evidence to support our method. To address your concern, we include MolGPT [1] and MolGen [2] as new baselines for *de novo* 1D/2D molecule generation on the GEOM-DRUGS and QM9-2014 datasets. MolGPT is a decoder-only transformer trained on SMILES sequences, and MolGen is an encoder-decoder transformer trained on SELFIES sequences. For MolGen, we fine-tune its public checkpoint on the two datasets; for MolGPT, we first reproduce its pretrained model using its source code, and then fine-tune it on the two datasets. Note that, we do not directly compare with MolGen’s reported performances, because MolGen employs a different experimental setting to fill-in corrupted spans in SELFIES sequences from the test set, which is unusual. Considering these two baselines cannot generate 3D structures, we report only 2D-metric of them for the *de novo* generation task, and do not include them for conditional 3D molecule generation (Table 5), which requires 3D structures. The results are presented below:

---

> ### Author Response · Authors · 2024-11-21
> **Response to Reviewer iSmq (Part 2/7)**
>
> **Table 2:** Performances for ***de novo*** 3D molecule generation. We highlight the results of new 1D LM baselines in $\textcolor{blue}{\text{blue}}$. * denotes our reproduced baseline results using their source codes.
>
> **(a)** Performances on the GEOM-DRUGS dataset. New baselines are evaluated for 2D-metric only.
>
> | **2D-Metric** | **FCD**$\downarrow$ | **AtomStable** | **MolStable** | **V&C** | **V&U** | **V&U&N** | **SNN** | **Frag** | **Scaf** |
> | --- | --- | --- | --- | --- | --- | --- | --- | --- | --- |
> | Train | 0.251 | 1.000 | 1.000 | 1.000 | 1.000 | 0.000 | 0.585 | 0.999 | 0.584 |
> | $\textcolor{blue}{\text{MolGPT}}*$ | $\textcolor{blue}{0.888}$ | $\textcolor{blue}{0.975}$ | $\textcolor{blue}{0.975}$ | $\textcolor{blue}{0.975}$ | $\textcolor{blue}{0.955}$ | $\textcolor{blue}{0.918}$ | $\textcolor{blue}{0.520}$ | $\textcolor{blue}{0.991}$ | $\textcolor{blue}{0.539}$ |
> | $\textcolor{blue}{\text{MolGen}}*$ | $\textcolor{blue}{0.655}$ | $\textcolor{blue}{\mathbf{1.000}}$ | $\textcolor{blue}{0.995}$ | $\textcolor{blue}{\mathbf{1.000}}$ | $\textcolor{blue}{0.993}$ | $\textcolor{blue}{0.759}$ | $\textcolor{blue}{0.513}$ | $\textcolor{blue}{0.993}$ | $\textcolor{blue}{0.549}$ |
> | CDGS | 22.051 | 0.991 | 0.706 | 0.285 | 0.285 | 0.285 | 0.262 | 0.789 | 0.022 |
> | JODO | 2.523 | **1.000** | 0.981 | 0.874 | 0.905 | 0.902 | 0.417 | 0.993 | 0.483 |
> | MiDi* | 7.054 | 0.968 | 0.822 | 0.633 | 0.654 | 0.652 | 0.392 | 0.951 | 0.196 |
> | EQGAT* | 6.310 | 0.999 | 0.998 | 0.959 | 0.993 | 0.702 | 0.368 | 0.986 | 0.147 |
> | NEXT-Mol, ours | **0.334** | **1.000** | **0.999** | **1.000** | **0.999** | **0.945** | **0.529** | **0.999** | **0.552** |
>
> **(b)** Performances on the QM9-2014 dataset. New baselines are evaluated for 2D-metric only.
>
> | **2D-Metric** | **FCD** $\downarrow$ | **AtomStable** | **MolStable** | **V&C** | **V&U** | **V&U&N** | **SNN** | **Frag** | **Scaf** |
> | --- | --- | --- | --- | --- | --- | --- | --- | --- | --- |
> | Train | 0.063 | 0.999 | 0.988 | 0.989 | 0.989 | 0.000 | 0.490 | 0.992 | 0.946 |
> | $\textcolor{blue}{\text{MolGPT}}$* | $\textcolor{blue}{0.461}$ | $\textcolor{blue}{0.975}$ | $\textcolor{blue}{0.975}$ | $\textcolor{blue}{0.975}$ | $\textcolor{blue}{0.936}$ | $\textcolor{blue}{0.763}$ | $\textcolor{blue}{0.523}$ | $\textcolor{blue}{0.958}$ | $\textcolor{blue}{0.923}$ |
> | $\textcolor{blue}{\text{MolGen}}$* | $\textcolor{blue}{0.085}$ | $\textcolor{blue}{\textbf{1.000}}$ | $\textcolor{blue}{0.988}$ | $\textcolor{blue}{\textbf{1.000}}$ | $\textcolor{blue}{0.955}$ | $\textcolor{blue}{0.479}$ | $\textcolor{blue}{0.500}$ | $\textcolor{blue}{0.988}$ | $\textcolor{blue}{0.934}$ |
> | CDGS | 0.798 | 0.997 | 0.951 | 0.951 | 0.936 | **0.860*** | 0.493 | 0.973 | 0.784 |
> | JODO | 0.138 | 0.999 | 0.988 | 0.990 | 0.960 | 0.780* | 0.522 | 0.986 | 0.934 |
> | MiDi* | 0.187 | 0.998 | 0.976 | 0.980 | 0.954 | 0.769 | 0.501 | 0.979 | 0.882 |
> | EQGAT* | 2.157 | **1.000** | 0.972 | **1.000** | **0.996** | 0.695 | 0.479 | 0.949 | 0.707 |
> | NEXT-Mol, ours | **0.070** | **1.000** | **0.989** | **1.000** | 0.967 | 0.802 | **0.530** | **0.992** | **0.945** |
>
> As shown in Table 2(a) and 2(b) above, NEXT-Mol consistently outperforms the new baselines. Specifically, **NEXT-Mol achieves relative FCD score improvement by 49% over MolGen and 62% over MolGPT on the GEOM-DRUGS datasets.** These significant improvements are attributed to NEXT-Mol’s benefits from the scaling law:  it is pretrained on a large dataset of 1.8 billion molecules, and features a large model size of 960M parameters. Additionally, NEXT-Mol will be released to support open-source research.
>
> We have revised our submission to include these new results.

---

> ### Author Response · Authors · 2024-11-21
> **Response to Reviewer iSmq (Part 3/7)**
>
> > **W2**: **Overemphasis on 100% Validity:** The paper focuses on achieving 100% validity, but in real-world applications, validity filtering is an extremely simple process due to how validity is defined. Consequently, there is no significant practical difference between achieving 90% validity and 100% validity.
> >
> > **Q2:** **Practical Advantages of 100% Validity:** Given that validity filtering is straightforward in practice, what are the practical advantages of achieving 100% validity over, for example, 90% validity in real-world applications?
> >
>
> **Response:** Thank you for your feedback. We agree that validity filtering is an simple process. Here we clarify that achieving 100% validity offers critical advantages beyond increasing validity itself:
>
> 1. **Improving validity could improve other 2D metrics, like SNN, Frag, and Scaf (see Table 2)**. These metrics measure the distributional similarity of 2D molecular structures of valid molecules. If a model still generate invalid molecules, it is likely the model does not capture the true target distribution, which contain only valid molecules. 100% validity helps the model learn from and sample from the valid molecular structures, which is essential for molecule generation tasks.
> 2. **Improving validity could improve 3D geometry learning.** The improved validity also leads to better learning of 3D molecular geometry, because it grounds 3D structure prediction on valid 2D structures. Other joint 2D and 3D prediction methods [3,4] can easily encounter invalid 2D structures when sampling 3D structures, therefore leads to worse 3D structure prediction. This is demonstrated by NEXT-Mol’s significant improvements in geometry similarity metrics (e.g., bond angle and bond length distributions) in Table 2.
>
> These improvements beyond validity are crucial for real-world applications. We appreciate your feedback and have clarify the corresponding paragraph in the introduction and Appendix C.1 in the revised manuscript.

---

> ### Author Response · Authors · 2024-11-21
> **Response to Reviewer iSmq (Part 4/7)**
>
> > **W3: Mischaracterization of Computational Complexity:** The paper states that structures were obtained using computationally intensive geometry optimization with DFT. However, the GEOM dataset was designed using the CREST software for conformer sampling, followed by geometry optimization with GFN2-xTB—a semi-empirical tight-binding method, not DFT. Moreover, compared to some deep learning models, this approach is not computationally intensive; on a reasonable workstation, geometry optimization takes about 0.5 seconds per average GEOM-Drugs structure.
> >
> > **W4: Missing Performance Metrics for Conformer Generation:** In conformer generation, computational performance is extremely important. OpenEye Omega remains one of the most popular software tools for this task because of its speed. The paper lacks performance metrics related to speed and efficiency, which are necessary for a fair comparison with existing tools.
> >
> > **Q3: Inclusion of Performance Metrics:** Considering that computational performance is crucial in conformer generation tasks, could you include speed and efficiency metrics for your model, particularly in comparison with tools like OpenEye Omega? I would also add OpenEye Omega conformer generation + consequent xTB geometry optimization time.
> >
>
> **Response:** We appreciate your attention to the computational complexity of our conformer generation process and the comparison with other conformer optimization toolkits. We would like to take a further discussion on these charming topics:
>
> - **The ground truth conformer sampling process (RDKit -> xTB -> CREST) in GEOM:** For a molecule in GEOM-DRUGS, the initial conformers are generated from SMILES strings using RDKit, followed by geometry optimization with GFN2-xTB. The lowest energy conformer is then selected as input to CREST for further optimization.
> - **Computational efficiency of GEOM-Drugs conformer generation:** Although the GFN2-xTB method is fast enough to be used in long metadynamics runs, the whole progress (RDKit -> xTB -> CREST) may take much more time. Particularly, the CREST program takes 89 core hours on Knights Landing Nodes or 8.2 core hours on Cascake Lake/Skay lake node [5] on average for each molecule in GEOM-DRUGS. Therefore, given a desktop of 8 core CPUs, it takes 1 -- 10 hours for conformer sampling, based on the CPU's performance and the complexity of the molecule.
> - **Time comparison with OpenEye Omega:** We conducted a time comparison between our model and OpenEye Omega for conformer generation on the test set of the GEOM-Drugs dataset, which includes 1000 molecules. The results are shown in the Table below.
>
> **Table:** Comparison of conformer generation time on the test set of the GEOM-Drugs dataset using various methods.
>
> | Method | Time |
> | --- | --- |
> | TD w/ PG | 347'08'' |
> | Omega | 74'39'' |
> | Omega + xTB | 74'39'' + 848'45'' |
> | DMT-L | 541'34'' |
> | DMT-L+Mollama | 827'44'' |
> | DMT-B | 181'22'' |
> | DMT-B+Mollama | 467'32'' |
>
> These experiments were performed on a platform with an 8-core Intel Xeon Processor@2.90GHz CPU and an NVIDIA A100 GPU and the time is measured in minutes and seconds. Please note that the Omega and GFN2-xTB are only run on the CPU only, while DMT and Mollama are run on the GPU. So the results may vary depending on the hardware.
>
> We have revised our Appendix B.4 to incorporate these results.

---

> ### Author Response · Authors · 2024-11-21
> **Response to Reviewer iSmq (Part 5/7)**
>
> > **W5: Use of Questionable Metrics:** The metric reported by JODO shows a 2.8% 3D molecule stability for the GEOM-Drugs dataset, rendering it practically meaningless. I strongly encourage avoiding the propagation of this metric in new papers. According to the code, the metric is based on a predefined bond length table, and a bond length is considered "good" if it is within 0.05 Å of the table value. However, the optimal distances between atoms are primarily defined by the energy landscape underlying the data—for GEOM-Drugs, it's GFN2-xTB—and depending on atom configurations, deviations in bond lengths can exceed 10%. While I'm uncertain about the validity of the 3D FCP metric, it's not entirely clear that it's completely off when compared with 3D molecule stability. Instead of attempting to report every possible metric to compare with other methods, focus on identifying the most important ones and emphasize the significance of the margins your model is achieving.
> >
> > **Q4: Analysis of Metrics Used:** Could you analyze the metrics used in your study, especially focusing on 3D molecular stability and the FCP metric? Since the FCP metric relies on a neural network, is it capable of handling your data distribution, and is it up-to-date with recent developments?
> >
>
> **Response:** Thank you for the valuable suggestions and comments.
>
> **On the validity of our evaluation metrics.** Our evaluation metrics follow the previous works [3,4,8,9] in 3D molecule generation. We discuss them below:
>
> - **On the 3D molecule stability metric.** We agree that this metric might be problematic, and we generally agree on your analysis. **Following your suggestion, we remove this metric in our revised main part of the submission**, and move it to the Appendix B.5 as a backup. We find that other metrics, including FCD, atom stability, the distribution similarity of bond length, bond angle, and dihedral angle, can still showcase the improvement of our method.
> - **On the FCD metric.** We agree that FCD is a more important metric for evaluating the quality of generated molecules.
>     - **Introduction to Fréchet ChemNet Distance (FCD).** It is widely used for evaluating the quality of generated molecules. It measures the similarity between the distributions of molecular representations of generated molecules and the ground truth molecules [6]. It is calculated using activations of the penultimate layer of the deep neural network ChemNet trained to predict biological activities of drugs which captures properties of the compounds. ChemNet was trained to predict biological activities, therefore the learned molecule representations contain both chemical and biological information.
>     - **Highlighting FCD.** Thank you for suggesting to highlighting our improvement of FCD. Following you suggestion, we have moved the presentation of FCD to the left most column in every comparison, to highlight its importance. On the GEOM-Drugs dataset, NEXT-Mol outperforms baselines by 5.39 FCD scores, a 26.5% relative improvement.
>     - **Alignment between FCD and our target data distribution.** The ChemNet used in FCD is trained on subsets of ZINC, PubChem, and ChEMBL. However, their filtered datasets are not open-source, therefore making the direct comparison infeasible. Here we draw some empirical insights: GDB-17, a superset of QM9 [10], overlaps with 57% of PubChem’s 2017 release and 60% of ChEMBL’s 2017 release. Similarly, GEOM-DRUGS likely overlaps with ChEMBL and ZINC, given their shared focus on drug-like molecules.
> - **On the distribution similarity of bond length, bond angle, and dihedral angle.** We measure the MMD distribution similarity for these geometry values. In general, these metrics can correctly reflect their description and should be reliable. In our understanding, they are included as surrogate metrics to evaluate the conformer quality, because the ground truth conformer is unavailable.

---

> ### Author Response · Authors · 2024-11-21
> **Response to Reviewer iSmq (Part 6/7)**
>
> > **W6: Inconsistencies in Metric Definitions and Reporting:** Providing detailed descriptions of the metrics used in the supplementary material is crucial due to inconsistencies across different papers. For example, MiDi reported the Wasserstein distance for bond angles and bond length distributions for all bonds and angles, whereas this paper (and at least JODO) uses MMD for the most frequent bonds, angles, and torsions. Additionally, the way MiDi computes atom stability and molecule stability differs from the JODO code. Performing kekulization at the beginning can alter the valencies of atoms, leading to different results (e.g., if you manually define H:O:H, where ":" is an aromatic bond, kekulization converts it to the valid water molecule H-O-H). Some models measure atom and molecule stability for raw data, while others use RDKit preprocessing before measuring stability, which can inflate results. It is essential to be consistent in comparisons. While this may not significantly change your results, it could artificially boost JODO's stability results, skewing the comparison. Overall, ensure that the same versions of the metrics are used across all comparisons and that they are well-defined in the supplementary material.
> >
> > **Q5 Consistency and Clarity of Metrics:** To ensure consistency, could you provide detailed descriptions of the metrics used in your study and confirm that the same versions are applied across all comparisons? Additionally, please clarify any preprocessing steps that might affect the results.
> >
>
> **Response:** Thank you for the question.
>
> - **Distribution similarity measures.** We use the MMD distance, following JODO, when reproducing MiDi and other baselines for consistent comparison.
> - **Regarding Kekulization.**
>     - **3D molecule.** We do not include this step when computing 3D atom and molecule stability. The corresponding code can be found at the get_3D_edm_metric function in the evaluation/eval_functions.py file in our anonymous code. This setting is consistently used for other baselines.
>     - **2D molecule.** Kekulization is used here following JODO. The same evaluation code is used for all the baselines, therefore the comparison should be fair. Due to the time limitation in rebuttal, we are still working on a reproduction without Kekulization for 2D molecules. Thank you for the suggestion. We will publish the results on OpenReview if they become available before the discussion deadline.
> - **RDKit preprocessing.**
>     - We use canonicalized SMILES for both the generated molecules and the training dataset when computing novelty and uniqueness of molecules. This is to ensure the correctness of the reported performance. We consistently use this measure for all the baselines.
>     - Other than canonicalization and Kekulization, no other RDkit preprocessing is used.
>
> We have revised our Appendix D.2 to include these important details of evaluation metrics.
>
> > **W7: Omission of EQGAT-Diffusion in Comparisons:** EQGAT-Diffusion is a recent model—published well before the ICLR deadline—that should be included in the comparisons for 3D unconditional molecule generation to provide a more comprehensive evaluation.
> >
> > **Q6:** **Inclusion of EQGAT-Diffusion in Comparisons:** EQGAT-Diffusion is a recent model relevant to 3D unconditional molecule generation. Could you include it in your comparisons to provide a more comprehensive evaluation?
> >
>
> **Response:** Thank you for pointing to this important related work and baseline model. We have included EQGAT-Diffusion in our comparisons for *de novo* 3D molecule generation to provide a more comprehensive evaluation. The results have been added to the revised manuscript in Table 2. We appreciate your feedback and have revised the manuscript accordingly.

---

> ### Author Response · Authors · 2024-11-21
> **Response to Reviewer iSmq (Part 7/7)**
>
> > **Q1: Comparison of MoLama with Other Models:** Could you provide a comparison between your 1D molecule generation model (MoLama) and other models like Equiformer on tasks beyond 3D molecule generation, such as molecule classification (or any suitable task where you can compare MoLama and let's say Equiformer)? Including such a comparison would highlight the standalone value of MoLama.
> >
>
> **Response:**  Thank you for the suggestion. To evaluate MoLlama's capabilities beyond 1D molecule generation, we apply it to molecular property prediction tasks, highlighting the quality of its molecular representations. Following the setup in [11], we fine-tune MoLlama on four MoleculeNet datasets: FreeSolv, ESOL, Lipo, and QM7. We adopt the same experimental settings and dataset splits as [11], reporting mean performance and standard deviation over 10 random seeds. For each run, MoLlama is trained for 100 epochs, with test performance selected based on the validation dataset. We use a fixed learning rate of 1e-4 with the AdamW optimizer, and fine-tune MoLlama using LoRA (LoRA r=8 and $\alpha=32$) applied to all linear layers of the model. Following Section 3.3, we attach a single-layer bi-directional self-attention layer after MoLlama to improve its encoding ability. After that, we apply a linear layer on the mean embedding of all molecule tokens for property prediction.
>
> **Table 10:** Molecule property regression results on four MoleculeNet datasets. Baseline results are from [11]. Lower $\downarrow$ is better.
>
> | Method | FreeSolv | ESOL | Lipo | QM7 |
> | --- | --- | --- | --- | --- |
> | **GNN-based methods** |  |  |  |  |
> | RF | 2.03±0.22 | 1.07±0.19 | 0.88±0.04 | 122.7±4.2 |
> | SVM | 3.14±0.00 | 1.50±0.00 | 0.82±0.00 | 156.9±0.0 |
> | GCN | 2.87±0.14 | 1.43±0.05 | 0.85±0.08 | 122.9±2.2 |
> | GATv2 | 3.14±0.00 | 1.41±0.00 | 0.89±0.00 | 113.3±0.0 |
> | GIN | 2.76±0.18 | 1.45±0.02 | 0.85±0.07 | 124.8±0.7 |
> | SchNet | 3.22±0.76 | 1.05±0.06 | 0.91±0.10 | 74.2±6.0 |
> | 3D Infomax | 2.23±0.26 | 0.947±0.04 | 0.739±0.01 | - |
> | MGCN | 3.35±0.01 | 1.27±0.15 | 1.11±0.04 | 77.6±4.7 |
> | D-MPNN | 2.18±0.91 | 0.98±0.26 | 0.65±0.05 | 105.8±13.2 |
> | **Pretrained GNN-based  methods** |  |  |  |  |
> | Pretrain-GNN | 2.83±0.12 | 1.22±0.02 | 0.74±0.00 | 110.2±6.4 |
> | MolCLR | 2.20±0.20 | 1.11±0.01 | 0.65±0.08 | 87.2±2.0 |
> | **LM-based methods** |  |  |  |  |
> | ChemBERTa-2 | 2.047±0.00 | 0.889±0.00 | 0.798±0.00 | 172.8±0.00 |
> | MolPROP | 1.70±0.09 | 0.777±0.02 | 0.733±0.02 | 151.8±10.0 |
> | MoLlama, ours | **1.59±0.04** | **0.740±0.01** | **0.627±0.01** | **63.5±1.6** |
>
> **Observation.** As shown in Table 10, MoLlama outperforms baseline methods, achieving relative improvements of 6.5%, 4.7%, 3.5%, and 16.9% on the FreeSolv, ESOL, Lipo, and QM7 datasets, respectively. Notably, our baselines include LM-based, GNN-based, and pretrained GNN-based methods, and MoLlama's better performance demonstrates its advantages derived from the extensive pretraining.
>
> We did not use choose Equiformer as a baseline because Equiformer is designed for 3D molecule property prediction, while MoLlama can only process 1D molecules without 3D. Therefore, we compare MoLlama with baselines that also process on 1D/2D molecules to ensure fairnes.
>
> We have included these results in Appendix B.1 to strengthen our presentation. Thank you for your insightful suggestion.
>
> **Reference:**
>
> [1] MolGPT: Molecular Generation Using a Transformer-Decoder Model. In Journal of Chemical Information and Modeling 2021.
>
> [2] Domain-Agnostic Molecular Generation with Chemical Feedback. In ICLR 2024.
>
> [3] Learning Joint 2D & 3D Diffusion Models for  Complete Molecule Generation. In TNNLS 2024
>
> [4] MiDi: Mixed Graph and 3D Denoising Diffusion for Molecule Generation. In ECML 2023.
>
> [5] GEOM, energy-annotated  molecular conformations for  property prediction and molecular  generation. In Scientific Data 2022.
>
> [6] Fréchet ChemNet Distance: A Metric for Generative Models forMolecules in Drug Discovery. In Journal of Chemical Information and Modeling 2018.
> [7] Molecular Sets (MOSES): A Benchmarking Platform for Molecular Generation Models.
> [8] Equivariant Diffusion for Molecule Generation in 3D. In ICML 2022.
>
> [9] Geometric Latent Diffusion Models for 3D Molecule Generation. In ICML 2023.
>
> [10] Enumeration of 166 Billion Organic Small Molecules in the Chemical Universe Database GDB-17. In Journal of Chemical Information and Modeling 2022.
>
> [11] MolPROP: Molecular Property prediction with multimodal language and graph fusion. In Journal of Cheminformatics 2024.

---

> > ### Comment · Reviewer_iSmq · 2024-11-21
> >
> > Thank you very much for providing comprehensive answers to all the raised questions, including the thorough speed evaluation and the assessment of the standalone MoLama component. The paper clearly demonstrates great performance in both conformer generation and de novo molecule generation. Even though it may not represent a breakthrough new method, the ability to stack and adjust known modules to achieve significant practical performance is as important as developing completely new theories.

---

> > > ### Author Response · Authors · 2024-11-22
> > > **Thank you for increasing the rating and the encouraging feedbacks**
> > >
> > > Thank you for increasing the rating from 6 to 8. Your positive feedback means a great deal to us and validates the effort we put into our work and our rebuttal.
> > >
> > > We deeply appreciate your valuable suggestions, such as including a more detailed comparison of MoLlama and your analysis of the evaluation metrics. These insights were instrumental in helping us refine our submission and gain a better understanding of the task.
> > >
> > > Your constructive and thoughtful comments have been incredibly helpful, and we are truly grateful for your support.

---

> ### Author Response · Authors · 2024-11-23
> **Updated Response for W6**
>
> > **W6:** Additionally, the way MiDi computes atom stability and molecule stability differs from the JODO code. Performing kekulization at the beginning can alter the valencies of atoms, leading to different results (e.g., if you manually define H:O:H, where ":" is an aromatic bond, kekulization converts it to the valid water molecule H-O-H). Some models measure atom and molecule stability for raw data, while others use RDKit preprocessing before measuring stability, which can inflate results. It is essential to be consistent in comparisons. While this may not significantly change your results, it could artificially boost JODO's stability results, skewing the comparison.
> >
>
> **Updated Response:** Thank you for the suggestion. Following your feedback, we removed the kekulization step from the evaluation script. Additionally, we modified the RDKit `MolFromSmiles` function to disable the sanitization step when loading SMILES into RDKit molecule objects. The molecule sanitization is a preprocessing and includes kekulization, disabling it improves the fairness when comparing 1D-based methods and 2D-based methods, which do not use `MolFromSmiles`.
>
> **Observation.** The evaluation results showed minimal changes for *de novo* 3D molecule generation. Only MolGPT exhibited notable differences, which is presented in the Table below:
>
> **Table:** Comparing MolGPT’s performance difference when using the new evaluation script.
>
> | Dataset | 2D-Metric | FCD $\downarrow$ | AtomStable | MolStable | V&C | V&U | V&U&N | SNN | Frag | Scaf |
> | --- | --- | --- | --- | --- | --- | --- | --- | --- | --- | --- |
> | GEOM-DRUGS | MolGPT | 0.888 | 0.957 | 0.957 | 0.957 | 0.955 | 0.918 | 0.520 | 0.991 | 0.539 |
> |  | MolGPT, new evaluation | 0.888 | $\color{blue}0.979$ | $\color{blue}0.977$ | 0.957 | 0.955 | 0.918 | 0.520 | 0.991 | 0.539 |
> | QM92014 | MolGPT | 0.461 | 0.975 | 0.975 | 0.975 | 0.936 | 0.760 | 0.523 | 0.958 | 0.923 |
> |  | MolGPT, new evaluation | 0.461 | $\color{blue}0.982$ | $\color{blue}0.976$ | $\color{blue}0.977$ | $\color{blue}0.937$ | 0.760 | 0.523 | 0.958 | 0.923 |
> - We can observe that, on the GEOM-DRUGS dataset, MolGPT’s atom stability increased from 0.957 to 0.979, and molecule stability increased from 0.957 to 0.977.
> - This change occurs because the sanitization step in RDKit imposes stricter constraints when loading molecules, leading to more SMILES-to-RDKit transformation failures. Disabling sanitization allows more molecules generated by MolGPT to be loaded successfully, improving its stability metrics.
> - Other models were minimally affected because they already generate nearly 100% stable molecules. For these models, the removal of sanitization and kekulization steps does not alter results significantly.
>
> **Why Removing Kekulization Has Little Effect?** We believe this is because most models already learn to distinguish aromatic bonds from other bonds effectively during training. As a result, their performance remains consistent regardless of the kekulization step.
>
> We have revised the manuscript to include this latest evaluation results. Other than MolGPT, MiDi’s performance also has a slight difference. The newly edited part are highlighted in $\text{\color{red}red}$. Thank you again for your valuable suggestions.

---

### Official Review · Reviewer_pyyj · 2024-11-02

**Soundness:** 2
**Presentation:** 2
**Contribution:** 2
**Rating:** 3
**Confidence:** 4

**Summary:**

The paper introduces NEXT-Mol, a model for 3D molecule generation that combines 1D Language Models with 3D diffusion models. NEXT-Mol first uses a pre-trained LM to generate 1D molecular sequences, ensuring chemical validity. It then predicts these molecules' 3D shapes with a refined diffusion model. Enhancements in model scaling, architecture, and transfer learning between 1D and 3D representations improve 3D predictions. NEXT-Mol is claimed to generate stable, accurate 3D conformers.

**Strengths:**

100% Validity: Ensures generated molecules are chemically valid by using a 1D SELFIES-based LM.
●  	Improved 3D Accuracy: Enhanced 3D conformer prediction through a refined diffusion model.
●  	Transfer Learning: Leverages 1D representations to boost 3D conformer prediction accuracy.
●  	Scalability: Scales well with large molecular datasets for robust molecule generation.
●  	Versatility: Performs well across tasks like de novo 3D generation, conformer prediction, and conditional molecule generation.

**Weaknesses:**

●  	Focus on Core Objective: In the title the paper is presented as a generative model for molecules. The model is benchmarked as a conformer generating model. Currently, benchmarks appear misaligned from the title; consider established benchmarks like CheckPose or DrugPose for 3D generation.
●  	1D-to-3D Transformation Claim: The claim that converting a 1D sequence to 3D adds value is questionable, as the graph already provides all necessary information.
●  	Not accurate statement on Rotation Augmentation: The statement, “Following AlphaFold3 (Abramson et al., 2024), we apply random rotation augmentation on 3D conformers to help DMT obtain equivariance to rotated inputs by learning. While (Wang et al., 2024) report decreased performance given random rotations, DMT benefits from it, potentially due to the improved neural architecture,” is unclear. The authors imply that rotation can be achieved without using equivariant networks (those maintaining symmetry under rotation). However, AlphaFold claimed that it is not necessary to have equivariant networks, so it is essential to clarify how DMT benefits from rotation augmentation and to distinguish it from learned equivariance.
●  	Limited ML Novelty: The model presents minimal innovation from an ML perspective, as it mainly combines existing components—LLaMA and diffusion models. This combination, particularly in transferring 1D information to 3D, offers limited novelty and benefit for the conformer generation part.

**Questions:**

●  	What is the rationale for combining 1D and 3D generation sequentially, and what benefits does this approach offer?
●  	Could you clarify how transferring a 1D representation contributes to the overall model performance?

---

> ### Author Response · Authors · 2024-11-21
> **Response to Reviewer pyyj (Part 1/5)**
>
> Thank you for your valuable comments and insightful feedback on our submission. Below, we offer a detailed, point-by-point response to address your concerns and integrate your suggestions. For clarity, we have grouped similar questions and identified weaknesses where applicable, assigning corresponding labels (e.g., W1 for Weakness 1 and Q1 for Question 1) based on your original comments.
>
> > **W1:** Focus on Core Objective: In the title the paper is presented as a generative model for molecules. The model is benchmarked as a conformer generating model. Currently, benchmarks appear misaligned from the title; consider established benchmarks like CheckPose or DrugPose for 3D generation.
> >
>
> **Response:** Thank you for the feedback. For response, we (1) clarify that **we have already employed 3D generation benchmarks of *de novo*  3D molecule generation and conditional 3D molecule generation**,  (2) **explain the rationale of using 3D conformer prediction as an additional benchmark**, and (3) justify that our benchmarks and contents align with our title.
>
> - **We have already included 3D molecule generation benchmarks.** Our method are evaluated on two-well established benchmarks:
>     - *De novo* 3D generation (Section 4.2): this task complements virtual screening by providing a candidate molecule space of manageable size [5], enabling the exploration only on the mostly relevant molecules.
>     - Conditional 3D molecule generation (Section 4.4): This task targets at quantum chemical properties (*e.g.*, HOMO and LUMO), which are useful for designing new material [6].
>
>     Both are well-established benchmarks in 3D molecular generation used by many previous works [1,2,3,4]  and demonstrate the practical value of our approach.
>
> - **Why not PoseCheck or DrugPose?** 3D molecule generation spans many tasks, such as antibody generation, de novo 3D molecule generation, conditional 3D molecule generation targeting quantum chemical properties, and structure based molecule generation (e.g., PoseCheck and DrugPose). In this work, we focus on de novo 3D molecule generation and conditional 3D molecule generation targeting quantum chemical properties. The benchmarks of PoseCheck and DrugPos involve different evaluation metrics and different baselines, making them outside the scope of this work. They can be studied as future works, as we mentioned in the Conclusion section.
> - **Content alignment with the title.** We are willing to revise the title to “*NEXT-Mol: 3D Diffusion Meets 1D Language Modeling for 3D Molecule Generation and Conformer Prediction*”, if you agree that this new title can better reflect our used benchmarks on both 3D molecule generation and 3D conformer prediction. Our existing title “*NEXT-Mol: 3D Diffusion Meets 1D Language Modeling for 3D Molecule Generation*”  can accurately reflect our used benchmarks of *de novo* 3D molecule generation and conditional 3D molecule generation, similar to the following recent studies:
>     - Equivariant Diffusion for Molecule Generation in 3D. In ICML 2022.
>     - GeoLDM: Geometric Latent Diffusion Models for 3D Molecule Generation. In ICML 2023.
>     - Learning Joint 2D & 3D Diffusion Models for Complete Molecule Generation. In TNNLS 2024.
>     - MiDi: Mixed Graph and 3D Denoising Diffusion  for Molecule Generation. In ECML 2023.
>     - Navigating the Design Space of Equivariant Diffusion-Based Generative Models for De Novo 3D Molecule Generation. In ICLR 2024.
>
>     All the works above use the same benchmarks (de novo 3D molecule generation or conditional 3D molecule generation) as ours, and share the keywords  of “molecule generation” and “3D”.
>
> - **Rationale of using 3D conformer prediction as an additional benchmark.** The performance of 3D conformer prediction is critical to our final de novo 3D molecule generation performance. This is because our method is a two-step solution for 3D molecule generation: initially generating a 1D molecule (a subset of a 3D molecule) using an LM and subsequently predicting its 3D conformer with a diffusion model. Thus, we include 3D conformer prediction as an additional benchmark to evaluate the effectiveness of our second step.

---

> ### Author Response · Authors · 2024-11-21
> **Response to Reviewer pyyj (Part 2/5)**
>
> > **W2:** 1D-to-3D Transformation Claim: The claim that converting a 1D sequence to 3D adds value is questionable, as the graph already provides all necessary information.
> >
> > **Q2:** Could you clarify how transferring a 1D representation contributes to the overall model performance?
> >
>
> **Response:** Thank you for the feedback. We agree that 1D sequences and 2D molecular graphs convey the same information. However, the key here is transfer learning: **the 2D molecular graphs do not have the knowledge learned by the 1D molecule LM from a billion-scale dataset**, but this knowledge can be encoded in 1D representations generated by the 1D LM. Our transfer learning is to use these knowledge learned from a billion-scale dataset to help 3D conformer prediction, which is constrained by limited data. Below, we elaborate the rationale behind this transfer learning.
>
> **Rationale behind transfer learning between 1D molecule sequences and 3D conformers.** The final goal of this transfer learning is to leverage the billion-scale 1D/2D molecule dataset to improve the 3D conformer prediction performance, which is constrained by limited 3D data. For clarity, we decompose the rationale into the following chain of arguments:
>
> 1. **3D conformers are theoretically governed by 2D molecular graphs under quantum mechanics (QM).** 3D molecular properties and structures are fundamentally rooted in QM. Using (approximated) QM-based methods, like DFT, we can accurately predict 3D conformers from 2D molecular graphs, though at high computational cost. This establishes the critical role of 2D representations in determining 3D structures.
> 2. **3D conformer prediction relies on high quality 2D molecule representations.** Deep learning models predict 3D conformers from 2D graphs, and their performance is heavily influenced by the quality of 2D molecular representations. Transfer learning can enhance 2D molecular representations, as demonstrated by prior works [7,8,9].
> 3. **1D molecular representations can be converted to 2D molecular representations, and contribute to 3D prediction.** 1D molecule sequences encode the same information as 2D molecular graphs, and the 1D to 2D transformation can be achieved by deterministic toolkit, like RDkit. Leveraging RDkit and our proposed cross-modal projector, which is detailed in Section 3.3 and Appendix C.3, we can transform 1D molecular representations to 2D molecular representations, and therefore contribute to the 3D prediction. We have demonstrated this improvement in Table 4, where using the pretrained 1D representations improve 3D conformer prediction.
> 4. **1D pretraining scales more effectively than 2D.** Given the billion-scale 1D/2D molecule dataset, we mostly prioritize the scalability when selecting the pretraining method. After literature review, we find that 1D LM-based pretraining methods, like Llama [10] and BERT [11], are extensively demonstrated for scalability and effectiveness. Therefore, we opt to 1D pretraining instead of 2D pretraining.

---

> ### Author Response · Authors · 2024-11-21
> **Response to Reviewer pyyj (Part 3/5)**
>
> > **W3:** Not accurate statement on Rotation Augmentation: The statement, “Following AlphaFold3 (Abramson et al., 2024), we apply random rotation augmentation on 3D conformers to help DMT obtain equivariance to rotated inputs by learning. While (Wang et al., 2024) report decreased performance given random rotations, DMT benefits from it, potentially due to the improved neural architecture,” is unclear. **The authors imply that rotation can be achieved without using equivariant networks (those maintaining symmetry under rotation). However, AlphaFold claimed that it is not necessary to have equivariant networks, so it is essential to clarify how DMT benefits from rotation augmentation and to distinguish it from learned equivariance.**
> >
>
> **Response:** Thank you for your question regarding equivariance. In this work, we adopt random rotation augmentation on 3D conformers to help DMT learn equivariance. We would like to clarify the following points:
>
> 1. **DMT uses learned equivariance instead of using built-in equivariant networks**: Our DMT model obtains equivariance to rotations by applying random rotation augmentation: we apply the same random rotation on the input 3D coordinates and the prediction target 3D coordinates. Therefore our model **learns to** output 3D coordinates that change equivariantly with the input. We **do not use build-in equivariant networks**, such as EGNN [13], which can achieve equivariance without any training.
> 2. **Our rotation augmentation is consistent with AlphaFold3**: We clarify that **AlphaFold3 can also learn equivariance** to rotations because it uses random rotation augmentation (see Algorithm 19 in their Appendix). We clarify that, the claim in their paper “we find that no invariance or equivariance with respect to global rotations and translation of the molecule are required in the architecture”, means that they do not use built-in equivariant architecture, like EGNN [13].
> 3. **Clarification on the benefits of rotation augmentation**: As shown in Table 7 below, the performance of DMT-B is improved with random rotation augmentation, which proves the effectiveness of this technique.
>
> **Table 7:** Ablating random rotation augmentation for 3D conformer prediction on GEOM-QM9.
>
> |  | COV-R (%) $\uparrow$ |  | AMR-R $\downarrow$ |  | COV-P (%) $\uparrow$ |  | AMR-P $\downarrow$ |  |
> | --- | --- | --- | --- | --- | --- | --- | --- | --- |
> | Method | Mean | Median | Mean | Median | Mean | Median | Mean | Median |
> | DMT-B | **95.2** | **100.0** | **0.090** | **0.036** | **93.8** | **100.0** | **0.108** | **0.049** |
> | *w/o* rand rot aug. | **95.2** | **100.0** | 0.089 | 0.04 | 93.3 | 100.0 | 0.113 | 0.053 |
>
> Thank you for the feedback. We have revised our description about rotation augmentation in Section 3.2 to improve clarity.

---

> ### Author Response · Authors · 2024-11-21
> **Response to Reviewer pyyj (Part 4/5)**
>
> > **W4:** Limited ML Novelty: The model presents minimal innovation from an ML perspective, as it mainly combines existing components—LLaMA and diffusion models. This combination, particularly in transferring 1D information to 3D, offers limited novelty and benefit for the conformer generation part.
> >
>
> **Response:** We respectfully disagree and argue that our model introduces significant ML novelties. Below, we outline the key novel contributions of our method:
>
> 1. **Novel Diffusion Molecular Transformer (DMT) Architecture**:
>     - Our DMT introduces a novel architecture that preserves complete 2D molecular graph information through Relational Multi-Head Attention (RMHA). RMHA extends standard self-attention by incorporating pairwise information that describes atomic interactions.
>     - DMT achieves superior performance while using fewer parameters than previous approaches (55M vs. MCF's 64M). Table 2 demonstrates significant improvements: FCD score improved from 0.655 to 0.334 on GEOM-DRUGS, with substantial gains in bond length, angle, and torsion MMD metrics
> 2. **Innovative 1D-to-3D Transfer**:
>     - Our cross-modal projector and training strategy effectively leverages MolLama's 1D representations to enhance DMT's 3D predictions. This approach uniquely addresses 3D data scarcity by utilizing abundant 1D molecular sequences. Table 4 shows consistent performance on the 3D conformer prediction task on both GEOM-DRUGS and GEOM-QM9 datasets after incorporating MolLama representations. Table 6 shows consistent improvement on the 3D molecule generation task.
> 3. **Large 1D Molecule LM:**
>     - To our knowledge, our MoLlama, which has 960M parameters and being pretrained on 1.8B molecules, is the largest SELFIES-based autoregresive LM for 1D molecule generation. It presents a significant contribution to the community.
> 4. **Versatile Multi-task Learning**:
>     - Our NEXT-Mol is a versatile multi-task learner and uniquely excels in all three tasks: de novo 3D generation, conformer prediction, and conditional 3D generation, as demonstrated in our comprehensive evaluation, while previous works typically specialize in only one of these tasks.
>
> **Other Reviewers’ Opinion:** we quote other reviewers’ comments regarding the novelty of our approach.
>
> - Reviewer ZnjA: “Innovative Approach: The combination of a 1D language model with a 3D diffusion model is an original solution to ensuring chemical validity while efficiently generating 3D conformers.”
> - Reviewer  iSmq: “Novel Methodology: The integration of 1D molecule generation through SELFIES with conformer prediction addresses the limitations of current datasets and methods in 3D molecule generation.”
>
> **Summary:** Our work combines novel architectural components, innovative 1D-to-3D transfer learning, and large-scale pretraining to tackle critical challenges in 3D molecular generation. The model’s versatility, scalability, and demonstrated improvements across diverse tasks highlight its significant ML contributions. We hope this clarifies the novelty of our approach.

---

> > ### Comment · Reviewer_pyyj · 2024-11-26
> >
> > Could you please provide concrete examples where transfer learning demonstrably improves 3D generation?
> >
> > My concern is that the 1D generation aspect is not novel, as it has been explored previously and is associated with well-documented failure points. While the 3D generation shows some improvement based on evaluation metrics, I am not convinced  that new information is being learned by the model. This is because the conformer generation step already leverages the graph representation as input, which already contains the required information for the prediction.
> >
> > Could you clarify which specific failure modes of graph-based models are being addressed or mitigated by your approach?

---

> ### Author Response · Authors · 2024-11-21
> **Response to Reviewer pyyj (Part 5/5)**
>
> > **Q1:** What is the rationale for combining 1D and 3D generation sequentially, and what benefits does this approach offer?
> >
>
> **Response:** Unlike previous works, NEXT-Mol decomposes 3D molecule generation into two steps: first generates 1D molecular sequences using MolLama (a large language model), and then conduct by 3D conformer prediction using a 3D diffusion model. The main motivation is to leverage the following two advantages of 1D generation:
>
> 1. **1D generation guarantees 100% molecular validity**: Achieving 100% validity offers critical advantages beyond increasing validity itself:
>     - **Improving validity could improve other 2D metrics, like SNN, Frag, and Scaf (see Table 2)**. These metrics measure the distributional similarity of 2D molecular structures of valid molecules. If a model still generate invalid molecules, it is likely the model does not capture the true target distribution, which contain only valid molecules. 100% validity helps the model learn from and sample from the valid molecular structures, which is essential for molecule generation tasks.
>     - **The improved validity also leads to better learning of 3D molecular geometry,** because it grounds 3D structure prediction on valid 2D structures. Other joint 2D and 3D prediction methods [15,16] can easily encounter invalid 2D structures when sampling 3D structures, therefore leads to worse 3D structure prediction.
> 2. **1D generation can be effectively large-scale pretrained**:
>     - Our large-scale pretraining leverages 1.8B molecules of 90B SELFIES tokens. It enables better learning of 1D/2D molecular patterns, compared to other methods that are constrained to the GEOM-DRUGS and QM9-2014 datasets, which are limited in size.
>     - The pretrained 1D LM can be used to improve 3D conformer prediction by transfer learning. Our proposed cross-modal projector effectively transfers MolLama's molecular knowledge to guide 3D prediction. Table 4 shows consistent improvements across multiple 3D conformer prediction metrics when incorporating MolLama representations
>
> This sequential pipeline effectively combines the strengths of SELFIES language modeling and 3D diffusion for accurate molecular generation.
>
> **Reference:**
>
> [1] Equivariant Diffusion for Molecule Generation in 3D. In ICML 2022
>
> [2] Learning Joint 2D & 3D Diffusion Models for  Complete Molecule Generation. In TNNLS 2024.
>
> [3] Equivariant Energy-Guided SDE for Inverse Molecular Design. In ICLR 2023
>
> [4] NAVIGATING THE DESIGN SPACE OF EQUIVARIANT DIFFUSION-BASED GENERATIVE MODELS FOR DE NOVO 3D MOLECULE GENERATION. In ICLR 2024
>
> [5] GuacaMol: Benchmarking Models for De Novo Molecular Design
>
> [6] Recent progress on exciplex-emitting OLEDs. In Journal of Information Display 2019.
>
> [7] STRATEGIES FOR PRE-TRAINING GRAPH NEURAL NETWORKS. In ICLR 2020
>
> [8] PRE-TRAINING MOLECULAR GRAPH REPRESENTATION WITH 3D GEOMETRY. In ICLR 2022.
>
> [9] GraphMAE: Self-Supervised Masked Graph Autoencoders. In KDD 2022.
>
> [10] Llama 2: Open Foundation and Fine-Tuned Chat Models. In Arxiv 2023.
>
> [11] BERT: Pre-training of Deep Bidirectional Transformers for Language Understanding. In NAACL 2019.
>
> [12] Swallowing the bitter pill: Simplified scalable conformer generation. In ICML 2024.
>
> [13] E(n) Equivariant Graph Neural Networks. In ICML 2021.
> [14] Accurate structure prediction of biomolecular interactions with AlphaFold3. In Nature 2024.
>
> [15] Learning Joint 2D & 3D Diffusion Models for  Complete Molecule Generation. In TNNLS 2024.
>
> [16] MiDi: Mixed Graph and 3D Denoising Diffusion  for Molecule Generation. In ECML 2023.

---

> ### Author Response · Authors · 2024-11-24
> **Follow-up Discussion**
>
> Dear Reviewer pyyj,
>
> Thank you for your thoughtful feedbacks on our submission, especially for advising us to 1) **clarifying the reasoning of choosing the existing 3D molecule generation benchmarks**, 2) **clarifying the content alignment with the title**, and 3) **clarifying the rationale behind 1D-to-3D transfer learning**. These valuable suggestions have improved the clarity and quality of our work. We hope that these improvements will be taken into consideration.
>
> If our response has resolved your concerns on our paper, we will greatly appreciate it if you could re-evaluate our paper for a higher rating. We are also willing and ready to engage in discussions, if you have any further questions.

---

> ### Author Response · Authors · 2024-11-27
> **Response to Followup Discussion (Part 1/2)**
>
> > **Followup Question.** Could you please provide concrete examples where transfer learning demonstrably improves 3D generation?
> >
> >
> > My concern is that the 1D generation aspect is not novel, as it has been explored previously and is associated with well-documented failure points. While the 3D generation shows some improvement based on evaluation metrics, I am not convinced that new information is being learned by the model. This is because the conformer generation step already leverages the graph representation as input, which already contains the required information for the prediction.
> >
> > Could you clarify which specific failure modes of graph-based models are being addressed or mitigated by your approach?
> >
>
> **Response:** Thank you for the follow-up discussions. Here we elaborate on the advantages of transfer learning from 1D to 3D generation in two aspects:
>
> **Argument 1: Pretrained 1D representations improves 3D conformer prediction.** We understand this to be the reviewer’s primary concern. Our ablation study in Table 5 demonstrates improvements in conformer prediction evaluation metrics, validating this claim. To further support our argument, we provide additional analysis and concrete examples that highlight the benefits of incorporating pretrained 1D representations.
>
> **Table 6:** 3D conformer prediction performance on GEOM-DRUGS’s test subsets, split by scaffold frequency in the training set. 68 low-quality samples are filtered following [1]. We report mean values of AMR-R and AMR-P.
>
> | Test subset | #Mol | Method | AMR-R $\downarrow$ | AMR-P $\downarrow$ |
> | --- | --- | --- | --- | --- |
> | unseen scaffold | 348 | DMT-B | 0.450 | 0.785 |
> |  |  | +MoLlama | **0.422** | **0.755** |
> | scaf. freq.>=1 | 584 | DMT-B | 0.364 | 0.549 |
> |  |  | +MoLlama | **0.359** | **0.548** |
> | scaf. freq.>=10 | 285 | DMT-B | 0.348 | 0.515 |
> |  |  | +MoLlama | **0.347** | **0.513** |
> - **MoLlama's 1D Representation Improves 3D Prediction for Unseen Scaffolds.** A molecule scaffold is the core structure of a molecule. Scaffold split, which ensures no scaffold overlap for molecules between the training and testing sets, is widely used to evaluate a molecular model's generalization ability to unseen structures [2]. To evaluate this, we divide GEOM-DRUGS's test set into subsets based on the test molecule's scaffold frequency in the training set. The results are shown in Table 6, we can observe that:
>     - **DMT’s failure mode.** DMT-B's performance drops significantly for test molecules with unseen scaffolds: AMR-R and AMR-P increase by 0.086 and 0.236, respectively, compared to molecules with scaffold frequency $\geq$ 1.
>         - **Why does this failure happen?** This is because deep learning methods are typically trained to mimic the training data, and often struggle when faced with distinct data distributions than training [2]. For molecular scaffolds absent from GEOM-DRUGS’s training set, DMT lacks sufficient prior knowledge to make accurate predictions. Even though all the bonds and atoms information are already provided in the molecular graph, the ability (1) to represent the 2D molecular structure, and (2) to decode 3D structures from the 2D structures, can still constrain a 3D conformer prediction model’s performance.
>     - **Transfer learning with MoLlama can mitigate this failure.** Incorporating MoLlama can mitigate this issue, reducing AMR-R and AMR-P by 0.028 and 0.030, respectively, for molecules with unseen scaffolds.
>         - **Reason for MoLlama’s Improvement.** This improvement is attributed to MoLlama’s pretraining on a large molecular dataset containing 1.8 billion molecules, compared to the 230k molecules in the GEOM-DRUGS training set. This extensive pretraining exposes MoLlama to a far more diverse range of scaffolds, enabling it to generate molecular representations that generalize more effectively. In contrast, DMT is limited to the scaffolds present in the smaller GEOM-DRUGS training set. The broader scaffold knowledge provided by MoLlama’s pretraining allows for better performance on unseen scaffolds.
>     - **Insights for seen scaffolds.** For test molecules with seen scaffolds, MoLlama’s improvement is minimal. This is because DMT can effectively learn the 3D structure prediction of these molecules from similar examples in the training data.
>     - $\textbf{\textcolor{red}{Visualization of concrete examples}.}$ We have updated Figure 4 in Section 4 and Figure 7 in Appendix B.7 to include visual examples where DMT+MoLlama outperforms DMT, focusing on cases with scaffolds absent from the training set. Notably, the primary improvements are observed in torsion angle predictions, even though our design does not explicitly target torsions. This may be because bond lengths and angles are easier to predict, while torsion angles pose greater challenges.

---

> > ### Author Response · Authors · 2024-11-27
> > **Response to Followup Discussion (Part 2/2)**
> >
> > - We revised our Section 4.5 to include these new results. Thank you very much for the insightful suggestion — it has significantly enhanced the completeness of our evaluation.
> > - Additionally, we revised the Limitations section (Appendix A) to highlight DMT’s generalization challenges with unseen scaffolds as a limitation and to suggest this as a potential direction for future work.
> >
> > **Argument 2: 1D pretraining improves *de novo* 1D generation.** *De novo* 1D molecule generation is the first step in our *de novo* 3D molecule generation pipeline, therefore improving 1D generation will improve the overall 3D generation performance. The advantage of using a pretrained model is clearly demonstrated in Table 2, by comparing NEXT-Mol to MolGPT and MolGen, which are also molecule language models, and in Table 9, by comparing the pretrained MoLlama and its non-pretrained version. These experimental results show that our extensive 1D pretraining makes MoLlama very effective for 1D molecule generation, laying a strong foundation for the subsequent 3D conformer prediction step.
> >
> > **Reference:**
> >
> > [1] Torsional Diffusion for Molecular Conformer Generation. In NeurIPS 2022.
> >
> > [2] Wilds: A Benchmark of in-the-Wild Distribution Shifts. In ICML 2021.
> >
> > [3] Strategies for Pre-training Graph Neural Networks. In ICLR 2020
> >
> > **Additional Updates.** We also make the following changes on our submission to follow your suggestions and improve the clarity:
> >
> > - We moved the experiment of 3D conformer prediction to Section 4.4, and moved the experiment of Conditional 3D Molecule Generation to Section 4.3.
> >     - This update aligns with your W1 suggestion to emphasize molecule generation as our primary focus.
> > - We revised the description on previous 3D conformer prediction models in the introduction.
> >     - This is to clearly reflect MCF’s current open-source status, and more accurately reflect the challenges faced by other methods.
> > - We revised the abstract and the second-to-last paragraph to highlight our model’s improvement in 3D molecule generation tasks.
> >     - This update aligns with your W1 suggestion to emphasize molecule generation as our primary focus.
> > - We moved the ablation study on rotation augmentation to Appendix B.1 to save room for the new analysis on unseen scaffolds.
> > - We have updated Figure 1 and Figure 2 to ensure a consistent figure style.
> >
> > **Further Discussion.** Thank you very much for your feedbacks. If our response has resolved your concerns on our paper, we will greatly appreciate it if you could re-evaluate our paper for a higher rating. We are also willing and ready to engage in discussions, if you have any further questions.

---

> > > ### Author Response · Authors · 2024-12-02
> > > **Follow-Up: Have We Addressed Your Concerns?**
> > >
> > > Dear Reviewer **pyyj**,
> > >
> > > Thank you for your thoughtful review of our submission. We recognize that the reviews are mixed, with two accept ratings (8) and two reject ratings (3).
> > >
> > > As the reviewer discussion window will close in 1 day, we would like to ask if the examples and analysis we provided on unseen scaffolds have addressed your concerns on how 1D representations improve 3D prediction?
> > >
> > > If your concerns are resolved by our response, we kindly invite you to re-evaluate our submission. Should you have any remaining questions or concerns, we would be glad to address them promptly.
> > >
> > > Thank you for your time and consideration.
> > >
> > > Authors

---

### Official Review · Reviewer_ZnjA · 2024-11-02

**Soundness:** 3
**Presentation:** 3
**Contribution:** 4
**Rating:** 8
**Confidence:** 3

**Summary:**

The paper "NEXT-Mol: 3D Diffusion Meets 1D Language Modeling for 3D Molecule Generation" introduces a foundational model for 3D molecule generation that combines the 3D diffusion model with a 1D language model trained on SELFIES representations. By integrating the advantages of both approaches, NEXT-Mol aims to address challenges in chemical validity, scalability, and data scarcity. The model comprises three components: (1) MoLlama, a large language model for generating 1D molecules, (2) DMT, a diffusion model for 3D conformer prediction, and (3) a transfer learning technique that utilizes MoLlama’s 1D representations to improve DMT’s 3D predictions. The experiments show that NEXT-Mol performs well on several datasets and tasks, including 3D molecule generation and conditional molecule generation with specific quantum chemical properties.

**Strengths:**

Innovative Approach: The combination of a 1D language model with a 3D diffusion model is an original solution to ensuring chemical validity while efficiently generating 3D conformers. This cross-modal learning technique enhances the model’s adaptability.
Comprehensive Experiments: The authors provide extensive experimental results across different tasks and datasets, demonstrating NEXT-Mol’s versatility and effectiveness in molecular generation and conformer prediction.
Practical Application Potential: This approach is particularly relevant for pharmaceutical applications where 3D molecular structures are critical for drug discovery and chemical analysis. The model’s strong performance on chemical validity and stability metrics suggests its practicality.
Scalability and Adaptability: The design of NEXT-Mol allows for transfer learning, making it more resource-efficient and adaptable to different datasets or molecule sizes, which is useful in a field with diverse requirements.

**Weaknesses:**

Limited Theoretical Insight: The paper lacks a theoretical explanation of why combining 1D and 3D modeling via transfer learning improves performance. Further theoretical analysis could provide deeper insights into the effectiveness of this architecture and potential limitations.
Absence of Ablation Studies on Model Size and Hyperparameters: While the paper shows promising results with two model sizes, a more detailed examination of how model size or key hyperparameters (e.g., noise schedule, batch size) impact performance would provide more guidance on model tuning.
Limited Exploration of Alternative Architectures: The use of RMHA and the specific structure of DMT are well-motivated but not directly compared to alternative architectures. A comparative study could clarify if these design choices are optimal for all molecular generation tasks.
Lack of Discussion on Model Limitations and Future Extensions: Although the model shows improvements, potential challenges such as memory overhead in larger molecules or limitations in certain chemical property predictions are not thoroughly discussed.

**Questions:**

Could you provide a theoretical justification for transfer learning between 1D molecular sequences and 3D conformers? An in-depth explanation could clarify why this cross-modal transfer is effective.
What are the potential computational trade-offs for using larger models (DMT-L) in terms of scalability and inference speed? Including a computational analysis of DMT-B versus DMT-L could highlight the scalability limits.
Are there specific molecular properties or types of molecules where NEXT-Mol struggles to perform as well? Identifying any limitations or edge cases where the model’s performance drops would clarify its practical scope.
Could you elaborate on why RMHA was chosen over other potential attention mechanisms? A comparison or justification of this design choice could strengthen the architectural motivation.

---

> ### Author Response · Authors · 2024-11-21
> **Response to Reviewer ZnjA (Part 1/4)**
>
> Thank you very much for the positive comments and valuable feedbacks on our submission. Here we provide a point-by-point response to address your concerns and incorporate your suggestions. For clarity, we have consolidated similar questions and weaknesses where appropriate, labeling them with corresponding identifiers (e.g., **W1** for Weakness 1 and **Q1** for Question 1) based on your original comments.
>
> > **W1:** Limited Theoretical Insight: The paper lacks a theoretical explanation of why combining 1D and 3D modeling via transfer learning improves performance. Further theoretical analysis could provide deeper insights into the effectiveness of this architecture and potential limitations.
> >
> > **Q1:** Could you provide a theoretical justification for transfer learning between 1D molecular sequences and 3D conformers? An in-depth explanation could clarify why this cross-modal transfer is effective.
> >
>
> **Response:** Thank you for the suggestion. We agree that a theoretical analysis for this 1D to 3D transfer learning can improve the rationality and presentation of our submission. While a comprehensive theoretical analysis is beyond the scope of this work, we offer an explanation combining theoretical insights and empirical observations.
>
> **The Rationale behind transfer learning between 1D molecule sequences and 3D conformers.** The final goal of this transfer learning is to leverage the billion-scale 1D/2D molecule dataset to improve the 3D conformer prediction performance, which is constrained by limited 3D data. For clarity, we decompose the rationale into the following chain of arguments:
>
> 1. **3D conformers are theoretically governed by 2D molecular graphs under quantum mechanics (QM).** 3D molecular properties and structures are fundamentally rooted in QM. Using (approximated) QM-based methods, like DFT, we can accurately predict 3D conformers from 2D molecular graphs, though at a high computational cost. This establishes the critical role of 2D representations in determining 3D structures.
> 2. **3D conformer prediction relies on high quality 2D molecule representations.** Deep learning models predict 3D conformers from 2D graphs, and their performance is heavily influenced by the quality of 2D molecular representations. Transfer learning can enhance 2D molecular representations, as demonstrated by prior works [1,2,3].
> 3. **1D molecular representations can be converted to 2D molecular representations, and contribute to 3D prediction.** 1D molecule sequences encode the same information as 2D molecular graphs, and the 1D to 2D transformation can be achieved by a deterministic toolkit, like RDkit. Leveraging RDkit and our proposed cross-modal projector, which is detailed in Section 3.3 and Appendix C.3, we can transform 1D molecular representations to 2D molecular representations, and therefore contribute to the 3D prediction. We have demonstrated this improvement in Table 4, where using the pretrained 1D representations improves 3D conformer prediction.
> 4. **1D pretraining scales more effectively than 2D.** Given the billion-scale 1D/2D molecule dataset, we mostly prioritize scalability when selecting the pretraining method. After the literature review, we find that 1D LM-based pretraining methods, like Llama [4] and BERT [5], are extensively demonstrated for scalability and effectiveness. Therefore, we opt to 1D pretraining instead of 2D pretraining.
>
> We have included this discussion in Appendix C.3 for enhanced presentation.
>
> > **W2:** Absence of Ablation Studies on Model Size and Hyperparameters: While the paper shows promising results with two model sizes, a more detailed examination of how model size or key hyperparameters (e.g., noise schedule, batch size) impact performance would provide more guidance on model tuning.
> >
> > **Q2:** What are the potential computational trade-offs for using larger models (DMT-L) in terms of scalability and inference speed? Including a computational analysis of DMT-B versus DMT-L could highlight the scalability limits.
> >
>
> **Response:** Thank you for the suggestion. In response, we discuss the influence of model size, noise schedule, and batch size on the 3D conformer prediction task.
>
> **Computational Trade-offs: DMT-B vs. DMT-L and MoLlama Integration.** We compare the performance and inference time of DMT-B, DMT-L, and their enhanced versions with MoLlama (DMT + MoLlama) using 1000 molecules from the GEOM-DRUGS test set.
>
> **Table:** Comparison of conformer generation time on the test set of the GEOM-DRUGS dataset using various methods.
>
> | Method | Time |
> | --- | --- |
> | DMT-L | 541'34'' |
> | DMT-L+Mollama | 827'44'' |
> | DMT-B | 181'22'' |
> | DMT-B+Mollama | 467'32'' |

---

> ### Author Response · Authors · 2024-11-21
> **Response to Reviewer ZnjA (Part 2/4)**
>
> - **DMT-B vs DMT-L.**
>     - **Performance:** DMT-L delivers better performance (4.7 % improvement on COV-P Median) than DMT-B, as shown in Table 3 in the manuscript.
>     - **Inference Time**: As shown by the Table above, DMT-L takes about 3 times as long as DMT-B for conformer generation on the GEOM-DRUGS dataset.
> - **DMT vs DMT + MoLlama.**
>     - **Performance:** Adding MoLlama improves 3D conformer predictions by ~1% in COV-R for both DMT-B and DMT-L on GEOM-DRUGS (see Table 6 in the manuscript).
>     - **Inference time:** Incorporating MoLlama increases inference time. For example, DMT-B’s inference time increases from under 200 minutes to 467 minutes.
> - **Conclusion.** The choice between DMT-B and DMT-L, as well as the inclusion of MoLlama, depends on task-specific requirements, particularly the trade-off between computational efficiency and 3D conformer prediction accuracy
> - Following your suggestion, we have revised Appendix B.3 to include a comparison of the inference time of our model and some representative baselines.
>
> **Different noise schedules at inference time.** We test DMT-B’s robustness to different noise schedulers at inference, using two representative options: the linear [7] and polynomial [8] schedulers. The original noise scheduler, based on the cosine function, follows [6]. Due to time constraints during the rebuttal, we use the existing DMT-B checkpoint without retraining the model with these new schedulers, so the results are suboptimal.
>
> **Table 11:**  Performance of DMT-B on the GEOM-DRUGS dataset with different noise schedulers at inference.
>
> |  | COV-R (%) $\uparrow$ |  | AMR-R $\downarrow$ |  | COV-P (%) $\uparrow$ |  | AMR-P $\downarrow$ |  |
> | --- | --- | --- | --- | --- | --- | --- | --- | --- |
> | Noise schedule | Mean | Median | Mean | Median | Mean | Median | Mean | Median |
> | linear | 62.7 | 62.7 | 0.648 | 0.634 | 60.3 | 60.6 | 0.726 | 0.624 |
> | cosine, training schedule | 85.4 | 92.2 | 0.401 | 0.375 | 65.2 | 67.8 | 0.642 | 0.577 |
> | polynomial | 84.9 | 91.7 | 0.454 | 0.421 | 64.5 | 66.2 | 0.685 | 0.619 |
>
> As shown in Table 11, the polynomial scheduler achieves performance close to the cosine scheduler, likely because their curve shapes are similar. However, the linear scheduler results in a significant performance drop, suggesting that retraining DMT-B with the linear scheduler is necessary to achieve better results.
>
> **The Influence of Batch Size to 3D Conformer Prediction.** We evaluate the performance of DMT-B with different batch sizes. The original batch size of 256 was chosen to maximize GPU utilization. To assess the impact of batch size, we tested two variations: (1) reducing the batch size to 128, and (2) increasing it to 512 using gradient accumulation.
>
> **Table 12:** DMT-B’s 3D conformer prediction performances on the GEOM-DRUGS dataset when using different batch sizes.
>
> |  | COV-R (%) $\uparrow$ |  | AMR-R $\downarrow$ |  | COV-P (%) $\uparrow$ |  | AMR-P $\downarrow$ |  |
> | --- | --- | --- | --- | --- | --- | --- | --- | --- |
> | Batch size | Mean | Median | Mean | Median | Mean | Median | Mean | Median |
> | 128 | 85.5 | 92.4 | 0.395 | 0.366 | 65.1 | 68.0 | 0.644 | 0.575 |
> | 256, original | 85.4 | 92.2 | 0.401 | 0.375 | 65.2 | 67.8 | 0.642 | 0.577 |
> | 512 | 85.1 | 92.0 | 0.410 | 0.377 | 64.9 | 67.7 | 0.645 | 0.582 |
>
> As shown in Table 11, the performance with a 512 batch size is slightly worse than the original model. This is likely due to underfitting caused by fewer training steps. We keep the number of training epochs the same as the original experiment (256 batch size, epoch 3000), therefore the larger batch size results in fewer gradient updates, leading to reduced model performance. Other than this observation, using the 128 batch size does not lead to a significant difference than the original model.
>
> Thank you for your suggestion. We have revised Appendix B.2 of our submission to include these experiments, and they have significantly improved our presentation.

---

> ### Author Response · Authors · 2024-11-21
> **Response to Reviewer ZnjA (Part 3/4)**
>
> > **W3:** Limited Exploration of Alternative Architectures: The use of RMHA and the specific structure of DMT are well-motivated but not directly compared to alternative architectures. A comparative study could clarify if these design choices are optimal for all molecular generation tasks.
> >
> > **Q4:** Could you elaborate on why RMHA was chosen over other potential attention mechanisms? A comparison or justification of this design choice could strengthen the architectural motivation.
> >
>
> **Response:** Thank you for the suggestion. We agree that including a comparison with alternative architectures would strengthen the presentation. Unfortunately, due to the time limitation in rebuttal and the significant efforts required for re-implementation and hyperparameter tuning, we regret to note that having this comparison is difficult. Below, we answer the other question regarding the elaboration of RMHA’s advantages over other attention mechanisms.
>
> We first revisit the original multi-head self-attention (MHA) and elaborate on how RMHA enhances MHA to incorporate pair information of graphs. Further, we present the limitations of three representative graph attention methods to justify our design choice.
>
> **Revisit MHA.**  Given a input sequence of node representations $\mathbf{H}\in \mathbb{R}^{N\times d}$, the original MHA can be written as follows:
>
> $$
> \\begin{align}
> [\mathbf{Q};\mathbf{K};\mathbf{V}]&=[\mathbf{W}\_q;\mathbf{W}\_k;\mathbf{W}\_v]\mathbf{H}^\top, \quad (1)\\\\
> a\_{ij} &= \text{softmax}\_j\left(\frac{\mathbf{Q}\_i\mathbf{K}\_j^\top}{\sqrt{d}}\right), \quad (2) \\\\
> \mathbf{O}\_i & = \sum\_{j=1}^N{a\_{ij}}\mathbf{V}\_j, \quad (3)
> \\end{align}
> $$
>
> where $\mathbf{O}_i\in \mathbb{R}^d$ is the output for the i-th input node. However, MHA cannot directly incorporate node pair’s information, such as bond existence or type, limiting its suitability for graph-based tasks.
>
> **Revisit RMHA.** RMHA addresses the limitation by taking a pair representation $\mathbf{E}\in \mathbb{R}^{N\times N\times d}$ as input, alongside the node representations $\mathbf{H}\in \mathbb{R}^{N\times d}$. $\mathbf{E}$ includes the edges of the fully connected structure of the corresponding graph. Specifically, $\mathbf{E}_{ij}\in \mathbb{R}^d$ encodes pair information like, whether $i$ and $j$ is connected by a bond, bond type (if applicable), and the Euclidean distance. RMHA modifies the MHA as follows:
>
> $$
> \\begin{align}
> [\mathbf{Q}; \mathbf{K}; \mathbf{V}] & = [\mathbf{W}\_{q}; \mathbf{W}\_{k}; \mathbf{W}\_{v}] \mathbf{H}^\top, \quad (4)\\\\
> [\mathbf{Q}^E;\mathbf{V}^E] & = \tanh([\mathbf{W}\_{eq};\mathbf{W}\_{ev}] \mathbf{E}^{\top}), \quad (5)\\\\
> a\_{i,j}&=\text{softmax}\_{j}(\frac{(\mathbf{Q}^E\_{i,j}\odot \mathbf{Q}\_i){\mathbf{K}}\_j^{\top}}{\sqrt{d}}), \quad (6)\\\\
> \mathbf{O}\_i &= \sum\_{j=1}^{N} a\_{i,j}(\mathbf{V}^E\_{i,j}\odot \mathbf{V}\_j). \quad (7)
> \\end{align}
> $$
>
> Here, the pair representation $\mathbf{E}$ modifies the node-level queries ($\mathbf{Q}_i$) and values ($\mathbf{V}_j$) through element-wise multiplication ($\odot$), enabling RMHA to fully incorporate pairwise information:
>
> - Pair information affects attention scores via $(\mathbf{Q}^E_{ij} \odot \mathbf{Q}_i) \mathbf{K}_j^\top$.
> - Pair information influences aggregated attention values via $\mathbf{V}^E_{ij} \odot \mathbf{V}_j$.
>
> By incorporating the rich pair information directly into the attention mechanism, RMHA significantly enhances the modeling of molecular graphs, addressing the limitations of MHA.
>
> **Limitations of Previous Works.**
>
> - **Limitations of MCF [9].** MCF is the previous state-of-the-art method for 3D conformer prediction.
>     - MCF uses the standard MHA, and incorporates graph structures by including graph eigenvectors as node features. However, graph eigenvectors lack bond type information, which can help discriminate bond lengths.
>     - MCF’s implementation uses a fixed subset of all the eigenvectors to handle variable-length molecules. This causes a lossy representation of graph structures, and might demonstrate insufficient when encountering large molecules.
> - **Limitations of Graphormer [10] and UniMol [11].**  Both methods enhance the original MHA by modifying attention scores to include pair representations. However:
>     - **Their pair representations are constrained to scalars, significantly limiting their representation capacity of the multi-dimensional pair features.**
>     - Their pair representations cannot influence the aggregated values in attention (see Equation 3 and Equation 7 above).
>     - In contrast, RMHA uses vector-based pairwise representations, which are more expressive and influence both attention scores and aggregated values, enabling richer modeling of molecular graphs.
>
> Thank you very much for the suggestion. We have revised Section 3.2 to incorporate the insights from this elaboration.

---

> ### Author Response · Authors · 2024-11-21
> **Response to Reviewer ZnjA (Part 4/4)**
>
> > **W4:** Lack of Discussion on Model Limitations and Future Extensions: Although the model shows improvements, potential challenges such as memory overhead in larger molecules or limitations in certain chemical property predictions are not thoroughly discussed.
> >
> > **Q3:** Are there specific molecular properties or types of molecules where NEXT-Mol struggles to perform as well? Identifying any limitations or edge cases where the model’s performance drops would clarify its practical scope.
> >
>
> **Response:** Thank you for the valuable suggestion. We identify the following limitations of the existing NEXT-Mol model:
>
> - **Limited Exploration on Diffusion Guidance.** Our DMT model utilizes i.i.d. sampling, without exploring advanced sampling methods like classifier guidance [12] and particle guidance [13]. However, particle guidance [13] demonstrates that a well-tuned guidance method can improve the conformer prediction by 10% precision. This is because the 3D molecular conformational space is large, and a guidance method with appropriate chemical inductive bias can improve the sampling efficiency. We leave this exploration as a future work.
> - **Computational Cost when Incorporating MoLlama for 3D Conformer Prediction.** Incorporating MoLlama, a large LM with 960M parameters, increases training time. For example, training DMT-B alone (55M parameters) takes 52 seconds per epoch on an A100 GPU, while DMT-B *with* MoLlama takes 210 seconds. We mitigated this problem by using a pretrained DMT-B, instead of training it from scratch, to reduce the training epochs when incorporating MoLlama. Yet, we will need improvement when transferring 1D representations from a large LM.
> - **Quadratic Memory Complexity of DMT’s Pair Representation.** This pair representation incurs an additional $O(N^2)$ GPU memory cost than the standard transformer, compared to the standard transformer’s $O(N)$ memory complexity when using FlashAttention, where N is the node number of molecular graphs. While we encountered no memory issues on the GEOM-DRUGS dataset (molecules with hundreds of nodes), this could be a bottleneck for molecules with thousands of nodes. Potential solutions include smaller batch sizes and model parallelism.
>
> Thank you for the valuable suggestion, we have incorporated this new discussion into our limitation section (Appendix A).
>
> **Reference:**
>
> [1] STRATEGIES FOR PRE-TRAINING GRAPH NEURAL NETWORKS. In ICLR 2020
>
> [2] PRE-TRAINING MOLECULAR GRAPH REPRESENTATION WITH 3D GEOMETRY. In ICLR 2022.
>
> [3] GraphMAE: Self-Supervised Masked Graph Autoencoders. In KDD 2022.
>
> [4] Llama 2: Open Foundation and Fine-Tuned Chat Models. In Arxiv 2023.
>
> [5] BERT: Pre-training of Deep Bidirectional Transformers for Language Understanding. In NAACL 2019.
>
> [6] Improved denoising diffusion probabilistic models. In ICLR 2021.
>
> [7] Denoising diffusion probabilistic models. In NeurIPS 2020.
>
> [8] Equivariant Diffusion for Molecule Generation in 3D. In ICML 2022.
>
> [9] Swallowing the Bitter Pill: Simplified Scalable Conformer Generation. In ICML 2024.
>
> [10] Do Transformers Really Perform Bad for Graph Representation? In NeurIPS 2021.
>
> [11] Uni-Mol: A Universal 3D Molecular Representation Learning Framework. In ICLR 2023.
>
> [12] Diffusion Models Beat GANs on Image Synthesis. In NeurIPS 2021.
>
> [13] PARTICLE GUIDANCE: NON-I.I.D. DIVERSE SAMPLING WITH DIFFUSION MODELS. In ICLR 2024.

---

### Official Review · Reviewer_boXt · 2024-11-02

**Soundness:** 2
**Presentation:** 2
**Contribution:** 2
**Rating:** 3
**Confidence:** 4

**Summary:**

The paper introduces the NEXT-Mol method, which first generates 1D molecular character representations using the Mol-LLAMA model and then generates the 3D structures of the molecules. The pre-trained molecular generation model’s atomic representations improve the performance of 3D structure generation.

**Strengths:**

1. This strategy has advantages in terms of the stability and effectiveness of molecule generation because it completely ignores the influence of 3D structures during the generation process.
2. The pre-trained molecular representations and the improved structural diffusion model achieve a new state-of-the-art (SOTA) in small molecule conformation generation, although the improvement is very slight.

**Weaknesses:**

1. The comparison results of 3D molecular generation in the paper are unfair because the authors completely ignore the influence of 3D structures on molecular representation during the generation process. The results of molecular generation should be compared with generation models based on 1D molecular representations.
2. The improvement in molecular conformation generation results mentioned in the paper is actually very small. In Table 3(A), DMT-B on GEOM-Drugs only improves by 1.4% compared to MCF-B, COV-R. Compared to Par. Guid, COV-P decreases by 3.7%, but the model parameters increase by 30 times. The significance of this method is limited.

**Questions:**

I think the authors should provide new evidence of the advantages of this method as a 3D model for molecular generation.

---

> ### Author Response · Authors · 2024-11-21
> **Response to Reviewer boXt (Part 1/3)**
>
> Thank you very much for your thoughtful feedbacks and detailed review comments. Below, we provide point-by-point responses to address your concerns.
>
> > **W1:** The comparison results of 3D molecular generation in the paper are unfair because the authors completely ignore the influence of 3D structures on molecular representation during the generation process. The results of molecular generation should be compared with generation models based on 1D molecular representations.
> >
>
> **Response:** Thank you for the thoughtful feedback. In response, we (1) **add new baselines using 1D language models (LMs) for 1D/2D molecule generation,** (2) clarify that **our comparisons are fair since we focus on baselines that can generate 3D molecules, which previous 1D LMs cannot,** and (3) clarify that **our model considers 3D structures through the 3D conformer generation process.**
>
> **Table 2:** Performances for ***de novo*** 3D molecule generation. We highlight the results of new 1D LM baselines in $\textcolor{blue}{\text{blue}}$. * denotes our reproduced baseline results using their source codes.
>
> **(a)** Performances on the GEOM-DRUGS dataset. New baselines are evaluated for 2D-metric only.
>
> | **2D-Metric** | **FCD**$\downarrow$ | **AtomStable** | **MolStable** | **V&C** | **V&U** | **V&U&N** | **SNN** | **Frag** | **Scaf** |
> | --- | --- | --- | --- | --- | --- | --- | --- | --- | --- |
> | Train | 0.251 | 1.000 | 1.000 | 1.000 | 1.000 | 0.000 | 0.585 | 0.999 | 0.584 |
> | $\textcolor{blue}{\text{MolGPT}}*$ | $\textcolor{blue}{0.888}$ | $\textcolor{blue}{0.979}$ | $\textcolor{blue}{0.977}$ | $\textcolor{blue}{0.975}$ | $\textcolor{blue}{0.955}$ | $\textcolor{blue}{0.918}$ | $\textcolor{blue}{0.520}$ | $\textcolor{blue}{0.991}$ | $\textcolor{blue}{0.539}$ |
> | $\textcolor{blue}{\text{MolGen}}*$ | $\textcolor{blue}{0.655}$ | $\textcolor{blue}{\mathbf{1.000}}$ | $\textcolor{blue}{0.995}$ | $\textcolor{blue}{\mathbf{1.000}}$ | $\textcolor{blue}{0.993}$ | $\textcolor{blue}{0.759}$ | $\textcolor{blue}{0.513}$ | $\textcolor{blue}{0.993}$ | $\textcolor{blue}{0.549}$ |
> | CDGS | 22.051 | 0.991 | 0.706 | 0.285 | 0.285 | 0.285 | 0.262 | 0.789 | 0.022 |
> | JODO | 2.523 | **1.000** | 0.981 | 0.874 | 0.905 | 0.902 | 0.417 | 0.993 | 0.483 |
> | MiDi* | 7.054 | 0.968 | 0.818 | 0.633 | 0.654 | 0.652 | 0.392 | 0.951 | 0.196 |
> | EQGAT* | 6.310 | 0.999 | 0.998 | 0.959 | 0.993 | 0.702 | 0.368 | 0.986 | 0.147 |
> | NEXT-Mol, ours | **0.334** | **1.000** | **0.999** | **1.000** | **0.999** | **0.945** | **0.529** | **0.999** | **0.552** |
>
> **(b)** Performances on the QM9-2014 dataset. New baselines are evaluated for 2D-metric only.
>
> | **2D-Metric** | **FCD** $\downarrow$ | **AtomStable** | **MolStable** | **V&C** | **V&U** | **V&U&N** | **SNN** | **Frag** | **Scaf** |
> | --- | --- | --- | --- | --- | --- | --- | --- | --- | --- |
> | Train | 0.063 | 0.999 | 0.988 | 0.989 | 0.989 | 0.000 | 0.490 | 0.992 | 0.946 |
> | $\textcolor{blue}{\text{MolGPT}}$* | $\textcolor{blue}{0.461}$ | $\textcolor{blue}{0.982}$ | $\textcolor{blue}{0.976}$ | $\textcolor{blue}{0.977}$ | $\textcolor{blue}{0.937}$ | $\textcolor{blue}{0.763}$ | $\textcolor{blue}{0.523}$ | $\textcolor{blue}{0.958}$ | $\textcolor{blue}{0.923}$ |
> | $\textcolor{blue}{\text{MolGen}}$* | $\textcolor{blue}{0.085}$ | $\textcolor{blue}{\textbf{1.000}}$ | $\textcolor{blue}{0.988}$ | $\textcolor{blue}{\textbf{1.000}}$ | $\textcolor{blue}{0.955}$ | $\textcolor{blue}{0.479}$ | $\textcolor{blue}{0.500}$ | $\textcolor{blue}{0.988}$ | $\textcolor{blue}{0.934}$ |
> | CDGS | 0.798 | 0.997 | 0.951 | 0.951 | 0.936 | **0.860*** | 0.493 | 0.973 | 0.784 |
> | JODO | 0.138 | 0.999 | 0.988 | 0.990 | 0.960 | 0.780* | 0.522 | 0.986 | 0.934 |
> | MiDi* | 0.187 | 0.998 | 0.976 | 0.980 | 0.954 | 0.769 | 0.501 | 0.979 | 0.882 |
> | EQGAT* | 2.157 | **1.000** | 0.972 | **1.000** | **0.996** | 0.695 | 0.479 | 0.949 | 0.707 |
> | NEXT-Mol, ours | **0.070** | **1.000** | **0.989** | **1.000** | 0.967 | 0.802 | **0.530** | **0.992** | **0.945** |
>
> **New baselines of 1D molecule LMs.** To address your concern, we include MolGPT [1] and MolGen [2] as new baselines for de novo 1D/2D molecule generation on the GEOM-DRUGS and QM9-2014 datasets. MolGPT is a decoder-only transformer trained on SMILES sequences, and MolGen is an encoder-decoder transformer trained on SELFIES sequences. For MolGen, we fine-tune its public checkpoint on the two datasets; for MolGPT, we first reproduce its pretrained model using its source code, and then fine-tune it on the two datasets. Note that, we do not directly compare with MolGen’s reported performances, because MolGen employs a different experimental setting to fill in corrupted spans in SELFIES sequences from the test set, which is unusual. We report only 2D-metric for these two baselines because they cannot generate 3D molecules.

---

> > ### Author Response · Authors · 2024-11-23
> > **Updated Table 2(a) Results**
> >
> > Dear Reviewer boXt,
> >
> > Thank you again for your valuable suggestions and comments on our submission. This is a gentle reminder that we have updated the MolGPT’s results in Table 2. Following the suggestion of Reviewer iSmq, we have revised the evaluation pipeline to remove the kekulization and sanitization steps before feeding molecules for evaluation. This is to ensure that the generated molecules’ structures are not affected by pre-processing. Other models are minimally influenced by this modification, while MolGPT’s results have slightly increased.
> >
> > If you have read Table 2 already, you can rest assure that the update is minor, only MolGPT's results are slightly different, which does not influence the main conclusion. If you have not read Table 2, the current Table 2 should reflect the latest results.
> >
> > Thank you again for your insightful feedback. We would be happy to further discuss and address any additional questions or concerns you may have.

---

> ### Author Response · Authors · 2024-11-21
> **Response to Reviewer boXt (Part 2/3)**
>
> As shown in Table 2(a) and 2(b) above, NEXT-Mol consistently outperforms the new baselines. Specifically, **NEXT-Mol achieves relative FCD score improvement by 49% over MolGen and 62% over MolGPT on the GEOM-DRUGS datasets.** These significant improvements are attributed to NEXT-Mol’s benefits from the scaling law:  it is pretrained on a large dataset of 1.8 billion molecules, and features a large model size of 960M parameters. Additionally, NEXT-Mol will be released to support open-source research.
>
> We have revised our submission to include these new results.
>
> **Our existing benchmark is fair because it focuses on baselines capable of generating 3D molecules.** We initially excluded 1D LMs as baselines since they cannot generate 3D molecules. However, following your recommendation, we have now included them to enhance the presentation. Additionally, in the de novo molecule generation task, we report both 2D metrics, which evaluate the generated 2D structures, and 3D metrics, which evaluate the generated 3D structures. This ensures fairness by comparing 2D generation methods to NEXT-Mol’s 2D outputs and 3D generation methods to NEXT-Mol’s 3D outputs.
>
> **Our model considers 3D structures during 3D generation through the 3D conformer generation process.** As elaborated in the introduction, we explore a two-step solution for 3D molecule generation: initially generating a 1D molecule (a subset of a 3D molecule) using an LM and subsequently predicting its 3D conformer using a diffusion model. Importantly, the generation of a 1D molecule is not the final step in 3D molecule generation. This is because a molecule can have many possible 3D conformers, therefore accurately predicting the desired 3D conformer is crucial for completing the 3D molecule generation.
>
> Thank you for your feedback. We have revised the introduction accordingly to improve the clarity.
>
> > **W2:** The improvement in molecular conformation generation results mentioned in the paper is actually very small. In Table 3(A), DMT-B on GEOM-Drugs only improves by 1.4% compared to MCF-B, COV-R. Compared to Par. Guid, COV-P decreases by 3.7%, but the model parameters increase by 30 times. The significance of this method is limited.
> >
>
> **Response:** Thank you for the feedback. We address your concerns by (1) highlighting that DMT achieves better performance **with fewer parameters** than MCF, and (2) revising our comparison with particle guidance (TD w/ PG) [3] to include reproduced baseline performance and fix a typo. The updated results for 3D conformer prediction on the GEOM-DRUGS dataset are shown in Table 3(a) below.
>
> **Table 3(a):** 3D conformer prediction results. * denotes reproduction using their codes. -R←Recall and -P←Precision. GEOM-DRUGS dataset. $\text{\color{blue}\small TD w/ PG denotes torsional diffusion with particle guidance.}$ We omit some baselines in the original submission for brevity.
>
> |  |  | COV-R $\uparrow$ |  | AMR-R $\downarrow$ |  | COV-P $\uparrow$ |  | AMR-P $\downarrow$ |  |
> | --- | --- | --- | --- | --- | --- | --- | --- | --- | --- |
> | Method | Model Size | Mean | Median | Mean | Median | Mean | Median | Mean | Median |
> | **Model size $\leq$  100M** |  |  |  |  |  |  |  |  |  |
> | Torsional Diffusion | 1.6M | 72.7 | 80.0 | 0.582 | 0.565 | 55.2 | 56.9 | 0.778 | 0.729 |
> | $\color{gray}{\text{\small TD w/ PD}}$ | $\color{gray}{1.6\text{M}}$ | $\color{gray}{77.0}$ | $\color{gray}{82.6}$ | $\color{gray}{0.543}$ | $\color{gray}{0.520}$ | $\color{gray}{68.9}$ | $\color{gray}{78.1}$ | $\color{gray}{0.656}$ | $\color{gray}{0.594}$ |
> | $\color{blue}{\text{\small TD w/ PD*}}$ | $\color{blue}{1.6\text{M}}$ | $\color{blue}{73.8}$ | $\color{blue}{79.3}$ | $\color{blue}{0.566}$ | $\color{blue}{0.539}$ | $\color{blue}{\mathbf{65.2}}$ | $\color{blue}{\mathbf{70.8}}$ | $\color{blue}{0.680}$ | $\color{blue}{0.615}$ |
> | MCF-S | 13M | 79.4 | 87.5 | 0.512 | 0.492 | 57.4 | 57.6 | 0.761 | 0.715 |
> | MCF-B | 64M | 84.0 | 91.5 | 0.427 | 0.402 | 64.0 | 66.2 | 0.667 | 0.605 |
> | DMT-B, ours | 55M | **85.4** | **92.2** | **0.401** | **0.375** | **65.2** | 67.8 | **0.642** | **0.577** |
> | **Model size > 100M** |  |  |  |  |  |  |  |  |  |
> | MCF-L | 242M | 84.7 | 92.2 | 0.390 | **0.247** | 66.8 | 71.3 | 0.618 | 0.530 |
> | DMT-L, ours | 150M | **85.8** | **92.3** | **0.375** | 0.346 | **67.9** | **72.7** | **0.598** | **0.527** |
>
> **DMT outperforms MCF with fewer parameters.** The comparison between DMT and MCF should consider both our better performances and our smaller model sizes. Specifically, DMT-B outperforms MCF-B by 1.4% in COV-R while using only 85% of its parameters, and DMT-L outperforms MCF-L by 1.1% in COV-R with just 61% of its parameters. This demonstrates that DMT achieves better performances with more efficient parameter utilization than MCF, which is a significant contribution to the field.

---

> ### Author Response · Authors · 2024-11-21
> **Response to Reviewer boXt (Part 3/3)**
>
> **New comparison between TD w/ PD and DMT.**
>
> - **Reproduced results for TD w/ PD.** Following your feedback, we reproduce the TD w/ PG baseline (denoted as TD w/ PG*) and include the results in Table 3(a). The reproduced results differ from the originally reported values. We include both, graying out the original results as we could not replicate them. We assure you that we used their source code and public checkpoint. For transparency, we upload the generated 3D conformers and the evaluation log into our anonymous share link [OSF](https://osf.io/gqy39/?view_only=5905ef8957f9444a8808fd49933b35c7), under the “onedrive/TD with PG Results” folder.
> - __New comparison to TD w/ PD*__. Using the reproduced results, we can observe that DMT-B achieves comparable COV-P to TD w/ PD*, while outperforming it by **11%** **in COV-R**. Additionally, DML-L outperforms TD w/ PD* across all metrics, demonstrating our model’s scalability. We hope these new results highlight the effectiveness of our DMT model, and address your concerns.
> - **Particle Guidance (TD w/ PD) [3] is an orthogonal contribution to our method.** Particle guidance is a diffusion guidance method like the classifier guidance [4]: it modifies torsional diffusion’s original i.i.d. sampling process. Our DMT model uses i.i.d sampling, and focuses on improving neural architectures and transfer learning from 1D data. Therefore, our contributions are orthogonal.
> - **Typo in original submission.** Thank you for noticing the typo: we have incorrectly bolded the best COV-P performance in Table 3(a) in the original submission, which is now fixed.
>
> Thank you for your thoughtful observations. We have updated Table 3(a) and Section 4.3 in our submission to address your suggestions, which we believe have significantly improved the presentation.
>
> > **Q1:** I think the authors should provide new evidence of the advantages of this method as a 3D model for molecular generation.
> >
>
> **Response:** Thank you for the suggestion. Following your suggestions, we have provided the following new evidence to support NEXT-Mol’s advantage for 3D molecule generation.
>
> - **New results for 1D LM.** In our previous response to weakness 1 (**W1**), we include new baselines of MolGPT [1] and MolGen [2], and demonstrate NEXT-Mol outperforms these 1D LM-based molecule generation methods.
> - __New comparison with TD w/ PG*.__ In our previous response to weakness 2 (**W2**), we include a new comparison between DMT and our reproduced TD w/ PG*. We show that DMT-B outperforms TD w/ PG* by 11% in COV-R, and achieves comparable COV-P scores. Additionally, DMT-L outperforms TD w/ PG* across all metrics.
>
> All these results are incorporated in our revised submission, which contributes to improving the empirical evidence of our methodology.
>
> **Reference:**
>
> [1] MolGPT: Molecular Generation Using a Transformer-Decoder Model. In Journal of Chemical Information and Modeling 2021.
>
> [2] Domain-Agnostic Molecular Generation with Chemical Feedback. In ICLR 2024.
>
> [3] PARTICLE GUIDANCE: NON-I.I.D. DIVERSE  SAMPLING WITH DIFFUSION MODELS. In ICLR 2024.
>
> [4] Diffusion Models Beat GANs on Image Synthesis. In NeurIPS 2021.

---

> ### Author Response · Authors · 2024-11-24
> **Follow-up Discussion**
>
> Dear Reviewer boXt,
>
> Thank you for your insightful feedbacks on our submission, especially for advising us to 1) **adding more comparisons to 1D molecular LMs**, and 2) **demonstrating the improvements compared to previous 3D conformer prediction methods**. These valuable suggestions have improved the significance and clarity of our work. We hope that these improvements will be taken into consideration.
>
> Beyond these updates above, we have also conducted **new experiments on molecule property prediction**, detailed below, to provide further evidence of the advantages of our approach.
>
> If our responses have adequately addressed your concerns, we would be grateful if you could reconsider the rating of our paper. Of course, we are more than happy to engage in further discussions if you have additional questions or comments.
>
> **Molecule Property Prediction Results.** To evaluate MoLlama's capabilities beyond 1D molecule generation, we apply it to molecular property prediction tasks, highlighting the quality of its molecular representations. Following the setup in [1], we fine-tune MoLlama on four MoleculeNet datasets: FreeSolv, ESOL, Lipo, and QM7. We adopt the same experimental settings and dataset splits as [1], reporting mean performance and standard deviation over 10 random seeds. For each run, MoLlama is trained for 100 epochs, with test performance selected based on the validation dataset. We use a fixed learning rate of 1e-4 with the AdamW optimizer, and fine-tune MoLlama using LoRA (LoRA r=8 and $\alpha=32$) applied to all linear layers of the model. Following Section 3.3, we attach a single-layer bi-directional self-attention layer after MoLlama to improve its encoding ability. After that, we apply a linear layer on the mean embedding of all molecule tokens for property prediction.
>
> **Table 10:** Molecule property regression results on four MoleculeNet datasets. Baseline results are from [1]. Lower $\downarrow$ is better.
>
> | Method | FreeSolv | ESOL | Lipo | QM7 |
> | --- | --- | --- | --- | --- |
> | **Supervised Learning Methods** |  |  |  |  |
> | RF | 2.03±0.22 | 1.07±0.19 | 0.88±0.04 | 122.7±4.2 |
> | SVM | 3.14±0.00 | 1.50±0.00 | 0.82±0.00 | 156.9±0.0 |
> | **Supervised GNN-based Methods** |  |  |  |  |
> | GCN | 2.87±0.14 | 1.43±0.05 | 0.85±0.08 | 122.9±2.2 |
> | GATv2 | 3.14±0.00 | 1.41±0.00 | 0.89±0.00 | 113.3±0.0 |
> | GIN | 2.76±0.18 | 1.45±0.02 | 0.85±0.07 | 124.8±0.7 |
> | SchNet | 3.22±0.76 | 1.05±0.06 | 0.91±0.10 | 74.2±6.0 |
> | 3D Infomax | 2.23±0.26 | 0.947±0.04 | 0.739±0.01 | - |
> | MGCN | 3.35±0.01 | 1.27±0.15 | 1.11±0.04 | 77.6±4.7 |
> | D-MPNN | 2.18±0.91 | 0.98±0.26 | 0.65±0.05 | 105.8±13.2 |
> | **Pretrained GNN-based Methods** |  |  |  |  |
> | Pretrain-GNN | 2.83±0.12 | 1.22±0.02 | 0.74±0.00 | 110.2±6.4 |
> | MolCLR | 2.20±0.20 | 1.11±0.01 | 0.65±0.08 | 87.2±2.0 |
> | **LM-based methods** |  |  |  |  |
> | ChemBERTa-2 | 2.047±0.00 | 0.889±0.00 | 0.798±0.00 | 172.8±0.00 |
> | MolPROP | 1.70±0.09 | 0.777±0.02 | 0.733±0.02 | 151.8±10.0 |
> | MoLlama, ours | **1.59±0.04** | **0.740±0.01** | **0.627±0.01** | **63.5±1.6** |
>
> **Observation.** As shown in Table 10, MoLlama outperforms baseline methods, achieving relative improvements of 6.5%, 4.7%, 3.5%, and 16.9% on the FreeSolv, ESOL, Lipo, and QM7 datasets, respectively. Notably, our baselines include LM-based, GNN-based, and pretrained GNN-based methods, and MoLlama's better performance demonstrates its advantages derived from the extensive pretraining.
>
> **Reference:**
>
> [1] MolPROP: Molecular Property prediction with multimodal language and graph fusion. In Journal of Cheminformatics 2024.

---

> > ### Author Response · Authors · 2024-11-27
> > **Awaiting Final Feedback on Revised Manuscript**
> >
> > Dear Reviewer boXt,
> >
> > As the reviewer-author discussion period nears its conclusion (November 27 at 11:59 pm AoE), we kindly remind you that we are eagerly awaiting your feedback on our revised manuscript.
> >
> > We have carefully addressed your valuables comments in the updated version. If our revisions have resolved your concerns, we would greatly appreciate it if you could re-evaluate our paper for a higher rating. If there are any remaining questions or clarifications needed, we are happy to address them promptly.
> >
> > Thank you for your time and thoughtful input. We deeply value your feedback and look forward to your response.
> >
> > Best regards,
> > Authors

---

> > > ### Author Response · Authors · 2024-12-02
> > > **Follow-Up: Have We Addressed Your Concerns?**
> > >
> > > Dear Reviewer **boXt**,
> > >
> > > Thank you for your thoughtful feedback on our submission. We appreciate your valuable suggestions and acknowledge the mixed reviews, with two accept ratings (8) and two reject ratings (3).
> > >
> > > As the reviewer discussion window is set to close in 1 day, we kindly ask if the new experimental results we provided—specifically on 1D molecule LM baselines and molecule property prediction—have addressed your concerns regarding the effectiveness of our method.
> > >
> > > If our response has resolved your concerns, we kindly invite you to re-evaluate our submission. Should you have any additional questions or require further clarification, we would be happy to address them promptly.
> > >
> > > Thank you for your time and consideration.
> > >
> > > Authors

---

> ### Author Response · Authors · 2024-12-03
> **More Experimental Results on New Sampler**
>
> |  |  | COV-R $\uparrow$ |  | AMR-R $\downarrow$ |  | COV-P $\uparrow$ |  | AMR-P $\downarrow$ |  |
> | --- | --- | --- | --- | --- | --- | --- | --- | --- | --- |
> | Method | Model Size | Mean | Median | Mean | Median | Mean | Median | Mean | Median |
> | **Model size $\leq$  100M** |  |  |  |  |  |  |  |  |  |
> | RDKit | - | 38.4 | 28.6 | 1.058 | 1.002 | 40.9 | 30.8 | 0.995 | 0.895 |
> | OMEGA | - | 53.4 | 54.6 | 0.841 | 0.762 | 40.5 | 33.3 | 0.946 | 0.854 |
> | GeoMol | 0.3M | 44.6 | 41.4 | 0.875 | 0.834 | 43.0 | 36.4 | 0.928 | 0.841 |
> | GeoDiff | 1.6M | 42.1 | 37.8 | 0.835 | 0.809 | 24.9 | 14.5 | 1.136 | 1.090 |
> | Torsional Diffusion | 1.6M | 72.7 | 80.0 | 0.582 | 0.565 | 55.2 | 56.9 | 0.778 | 0.729 |
> | $\text{\textcolor{gray}{TD w/ PD}}$ | $\text{\textcolor{gray}{1.6M}}$ | $\text{\textcolor{gray}{77.0}}$ | $\text{\textcolor{gray}{82.6}}$ | $\text{\textcolor{gray}{0.543}}$ | $\text{\textcolor{gray}{0.520}}$ | $\text{\textcolor{gray}{68.9}}$ | $\text{\textcolor{gray}{78.1}}$ | $\text{\textcolor{gray}{0.656}}$ | $\text{\textcolor{gray}{0.594}}$ |
> | $\text{\textcolor{blue}{TD w/ PD*}}$ | $\text{\textcolor{blue}{1.6M}}$ | $\text{\textcolor{blue}{73.7}}$ | $\text{\textcolor{blue}{79.3}}$ | $\text{\textcolor{blue}{0.566}}$ | $\text{\textcolor{blue}{0.539}}$ | $\text{\textcolor{blue}{65.2}}$ | $\text{\textcolor{blue}{70.7}}$ | $\text{\textcolor{blue}{0.680}}$ | $\text{\textcolor{blue}{0.615}}$ |
> | MCF-S | 13M | 79.4 | 87.5 | 0.512 | 0.492 | 57.4 | 57.6 | 0.761 | 0.715 |
> | MCF-B | 64M | 84.0 | 91.5 | 0.427 | 0.402 | 64.0 | 66.2 | 0.667 | 0.605 |
> | DMT-B, i.i.d. sampler, ours | 55M | 85.4 | **92.2** | 0.401 | 0.375 | 65.2 | 67.8 | 0.642 | 0.577 |
> | $\text{\textcolor{blue}{DMT-B, PC samp., snr=0.2, ours}}$ | $\text{\textcolor{blue}{55M}}$ | $\text{\textcolor{blue}{85.3}}$ | $\text{\textcolor{blue}{91.5}}$ | $\text{\textcolor{blue}{0.398}}$ | $\text{\textcolor{blue}{0.372}}$ | $\text{\textcolor{blue}{66.5}}$ | $\text{\textcolor{blue}{69.2}}$ | $\text{\textcolor{blue}{0.633}}$ | $\text{\textcolor{blue}{0.560}}$ |
> | $\text{\textcolor{blue}{DMT-B, PC samp., snr=0.3, ours}}$ | $\text{\textcolor{blue}{55M}}$ | $\textbf{\textcolor{blue}{85.5}}$ | $\text{\textcolor{blue}{91.2}}$ | $\textbf{\textcolor{blue}{0.396}}$ | $\textbf{\textcolor{blue}{0.370}}$ | $\text{\textcolor{blue}{67.6}}$ | $\text{\textcolor{blue}{71.5}}$ | $\text{\textcolor{blue}{0.623}}$ | $\textbf{\textcolor{blue}{0.546}}$ |
> | $\text{\textcolor{blue}{DMT-B, PC samp., snr=0.4, ours}}$ | $\text{\textcolor{blue}{55M}}$ | $\text{\textcolor{blue}{73.8}}$ | $\text{\textcolor{blue}{79.9}}$ | $\text{\textcolor{blue}{0.535}}$ | $\text{\textcolor{blue}{0.501}}$ | $\textbf{\textcolor{blue}{68.0}}$ | $\textbf{\textcolor{blue}{72.1}}$ | $\textbf{\textcolor{blue}{0.621}}$ | $\text{\textcolor{blue}{0.548}}$ |
>
> **Settings.** We implemented the Predictor-Corrector (PC) sampler [1] to further enhance DMT’s performance in 3D conformer prediction. Specifically, we used the original checkpoint without retraining and modified only the inference code. By tuning the `snr` hyperparameter, the PC sampler enables a trade-off between diversity (recall) and precision. We experimented with snr values of {0.2, 0.3, 0.4}, observing that higher snr values lead to greater precision but reduced diversity.
>
> **Observations.** The PC sampler can improve DMT-B’s sampling precision with minimal impact on recall. For example, DMT-B with the PC sampler (snr=0.3) significantly outperforms our original vanilla i.i.d. sampler and the TD *w/* PD* baseline across all metrics, achieving a 2.4% increase in COV-P and an 11.8% increase in COV-R. However, increasing snr to 0.4 further boosts precision but causes a noticeable decline in recall. These results highlight the utility of hyperparameter tuning and may guide future research.
>
> **Why We Did Not Implement Particle Guidance?** This experiment demonstrates that DMT’s performance can be further improved using an advanced sampler, similar to how Particle Guidance [2] enhances torsional diffusion. However, we did not use Particle Guidance because its original implementation is designed for diffusion on torsion angles. Adapting it to our method, which operates in the full Euclidean space, is non-trivial and requires significant theoretical and engineering effforts.
>
> **Further Discussion.** We hope these new results underscore the effectiveness of our methods and provide valuable insights for future work. We look forward to continued discussion and collaboration.
>
> **Reference:**
>
> [1] SCORE-BASED GENERATIVE MODELING THROUGH  STOCHASTIC DIFFERENTIAL EQUATIONS. In ICLR 2021
>
> [2] PARTICLE GUIDANCE: NON-I.I.D. DIVERSE  SAMPLING WITH DIFFUSION MODELS. In ICLR 2024.

---

### Author Response · Authors · 2024-11-23
**General Response to Reviewers**

We sincerely appreciate all reviewers’ time and effort in evaluating our submission. Our paper received mixed reviews, including two acceptances (scores of 8) and two rejections (scores of 3).

We thank all the reviewers for their valuable suggestions, which have significantly contributed to improving our work. In response to your concerns and questions, we have revised our submission to include new experimental results and enhance the overall presentation. Below, we summarize the key revisions:

- **[Reviewer boXt and iSmq]:** We included 1D molecular language models as new baselines for 1D molecule generation. The updated results are presented in Table 2.
- **[Reviewer boXt]:** We demonstrated the improvements over Particle Guidance. These results are now included in the revised Table 3.
- **[Reviewer  pyyj and ZnjA]:** We elaborated on the rationale behind 1D-to-3D transfer learning. This discussion can be found in Appendix C.3.
- **[Reviewer pyyj and iSmq]:** We discussed the advantages of achieving 100% validity beyond validity itself in the context of 1D and 3D molecule generation. This discussion is provided in Appendix C.1.
- **[Reviewer ZnjA and iSmq]:** We compared the computational time of our proposed 3D conformer prediction method with baselines. The comparison is included in Figure 5 in Appendix B.4.
- **[Reviewer iSmq]:**  We conducted new experiments for molecule property prediction using MoLlama to demonstrate its representation quality. The results are included in Table 10 in Appendix B.2.
- **[Reviewer iSmq]:** We clarified and updated the evaluation metrics for 3D generation. Specifically, Table 2 was revised to remove the 3D molecule stable metric, while the 2D atom stable and 2D molecule stable metrics were updated. Additionally, the FCD metric has been highlighted by moving it to the leftmost column of every comparison. Details of the evaluation metrics are provided in Appendix D.3.
- **[Reviewer ZnjA]:** We performed a hyperparameter analysis and ablation study on different noise schedulers and batch sizes. The results are included in Tables 11 and 12 in Appendix B.3.
- **[Reviewer ZnjA]:** We added a more detailed discussion of the limitations and future directions of our work. These updates are included in the revised Limitations Section in Appendix A.

We hope these revisions address the reviewers’ concerns and provide clarity on the contributions of our work. Please do not hesitate to reach out with further questions or comments. We look forward to continuing the discussion.

---

### Meta-Review · Area_Chair_iuK6 · 2024-12-20

**Metareview:**

The reviews for this submission were highly biased, consisting of four reviewer opinions. Two reviewers with a confidence level of 4 scored 3 and recommended rejection; however, this was their first review, and they did not respond. One reviewer with a confidence level of 3 scored 8 and recommended acceptance. Another reviewer with a confidence level of 4 increased the score from 6 to 8, though it was also the first time review.

From my perspective, the authors addressed all the reviewers' concerns very diligently in their rebuttal, but the method's innovation seems limited, especially in transferring the 1D sequence information to 3D generation (also mentioned by Reviewer pyyj). Therefore, my recommendation is to accept (poster) even though the average score is below 6. However, if the Senior Area Chair (SAC) decides to reject, I also find this reasonable as the paper is borderline."

**Additional Comments On Reviewer Discussion:**

Two reviewers (Reviewer boXt and Reviewer pyyj) with a confidence level of 4 scored 3 and recommended rejection; however, this was their first review, and they did not respond to the rebuttal.

One reviewer with a confidence level of 3 scored 8 and recommended acceptance. Another reviewer with a confidence level of 4 increased the score from 6 to 8, though it was also the first time review.

---

### Decision · Program_Chairs · 2025-01-22

Accept (Poster)